# Acute stress induces long-term metabolic, functional, and structural remodeling of the heart

Thulaciga Yoganathan [1] ✉, Mailyn Perez-Liva [1,2], Daniel Balvay [1,3], Morgane Le Gall [4], Alice Lallemand [1], Anais Certain[1], Gwennhael Autret [1,3], Yasmine Mokrani [1], François Guillonneau [5], Johanna Bruce [4], Vincent Nguyen [6], Umit Gencer[7], Alain Schmitt [8], Franck Lager [9], Thomas Guilbert [10], Patrick Bruneval[1], Jose Vilar [1], Nawal Maissa [1], Elie Mousseaux [7], Thomas Viel [1,3], Gilles Renault [9], Nadjia Kachenoura [6] & Bertrand Tavitian [1,3,7] ✉

Takotsubo cardiomyopathy is a stress-induced cardiovascular disease with symptoms comparable to those of an acute coronary syndrome but without coronary obstruction. Takotsubo was initially considered spontaneously reversible, but epidemiological studies revealed significant long-term morbidity and mortality, the reason for which is unknown. Here, we show in a female rodent model that a single pharmacological challenge creates a stress-induced cardiomyopathy similar to Takotsubo. The acute response involves changes in blood and tissue biomarkers and in cardiac in vivo imaging acquired with ultrasound, magnetic resonance and positron emission tomography. Longitudinal follow up using in vivo imaging, histochemistry, protein and proteomics analyses evidences a continued metabolic reprogramming of the heart towards metabolic malfunction, eventually leading to irreversible damage in cardiac function and structure. The results combat the supposed reversibility of Takotsubo, point to dysregulation of glucose metabolic pathways as a main cause of long-term cardiac disease and support early therapeutic management of Takotsubo.

Stress is an independent risk factor for cardiovascular diseases in patients with pre-existing cardiac diseases or cardiovascular risk factors, as well as in persons without a known cardiac condition[1,2]. A spectacular consequence of acute stress on the heart is Takotsubo cardiomyopathy (TTC), a condition mimicking an acute coronary syndrome but in which coronarography appears normal[3]. In the absence of other signs of associated or underlying cardiac conditions, TTC patients are dismissed from hospitals because the acute cardiac symptoms (pain, electrocardiography, heart rate, blood pressure…) are reversible and because there is no evidence of coronary

[1]Université Paris Cité, Inserm, PARCC, F-75015 Paris, France. [2]Nuclear Physics Group and IPARCOS, Department of Structure of Matter, Thermal Physics and Electronics, CEI Moncloa, Universidad Complutense de Madrid, 28040 Madrid, Spain. [3]Université Paris Cité, Plateforme d'Imageries du Vivant, PARCC, F-75015 Paris, France. [4]Université Paris Cité, P53 proteom'IC facility, Institut Cochin, INSERM, CNRS, F-75014 Paris, France. [5]Institut de Cancérologie de l'Ouest, CNRS UMR6075 INSERM U1307, 15 rue André Boquel, F-49055 Angers, France. [6]Sorbonne Université, Laboratoire d'Imagerie Biomédicale, Inserm, CNRS, F-75006 Paris, France. [7]Service de Radiologie, AP-HP, hôpital européen Georges Pompidou, F-75015 Paris, France. [8]Université Paris Cité, Cochin Imaging, Electron microscopy, Institut Cochin, INSERM, CNRS, F-75014 Paris, France. [9]Université Paris Cité, Plateforme d'Imageries du Vivant, Institut Cochin, Inserm-CNRS, F-75014 Paris, France. [10]Université Paris Cité, Cochin Imaging Photonic, IMAG'IC, Institut Cochin, Inserm, CNRS, F-75014 Paris, France. ✉e-mail: thulacigayoganathan@calicolabs.com; bertrand.tavitian@inserm.fr

obstruction. However, recent long-term follow-up studies have reported similar cardiovascular annual death rates for TTC and for proven acute coronary obstruction[4,5], suggesting that TTC often induces severe cardiac sequelae.

Although there is converging evidence that the stress-induced catecholamine rush is the cause of Takotsubo[6], the mechanism by which a single acute stressing event can have long-term deleterious effects on the heart remains mysterious. Catecholamines activate adrenoreceptors, in particular the beta-1 adrenoreceptors that are predominant in the heart. The activation of beta1 and beta2 receptors has positive inotropic and chronotropic effects, increases cardiac output, myocardial oxygen consumption and coronary flow[7], while the activation of beta 3 receptors has a negative inotropic effect. Studies have reported that at high catecholamine concentrations, such as those found during acute stress, the activation of beta 3 receptors counteracts the activation of beta 1 and 2 receptors[8], supporting the view that, mechanistically, the heart is well adapted to stress. Another role of stress-induced catecholamine release is to secure energy substrates for the "fight or flight" reaction, by opposing the action of insulin, namely by inducing hyperglycemia and glycogenolysis. The mammalian heart is omnivorous: in basal conditions, it feeds essentially on fatty acids (ca. 80%) and modestly on glucose (ca. 20%) and other substrates[9]. In conditions of high-energy demand, e.g., exercise, the supplementary energy requirements are provided by an increase of aerobic and anaerobic glucose breakdown. This is also the case under high glycaemia in diabetes, or in conditions with reduced cardiac muscle microperfusion such as under anti-angiogenic treatment[10,11]. The adaptation of the cardiac energy balance through the regulation of glucose utilization is advantageous for the challenged heart because it can be switched on very rapidly and can maintain ATP production in low oxygen conditions. The mechanisms involve increased intracellular transport of glucose through Glut1, the universal glucose membrane transporter, and of Glut4 that is expressed in the heart, skeletal muscle, and insulino-dependant tissues. Glut4 translocation to the plasmatic membrane is under the control of insulin growth factor (IGF) and other effectors, including the stress hormones catecholamines and cortisol.

Although glucose metabolism is a major factor of regulation of cardiac activity in health and disease[12], molecular imaging of metabolism is not an indication in patients with Takotsubo. One of the reasons may be that the metabolic responses of the heart to a sudden and dramatic rise of circulating catecholamines are not easily predictable and depend on the respective affinities and sensitivities of different adrenergic receptors and on their cardiac densities, as well as on the physiological and metabolic state of the heart, which is highly variable among individuals[13]. Moreover, such studies are difficult to conduct in human patients because the neurohumoral response to stress is highly variable and unpredictable among individuals, and because cardiac and metabolic comorbidities are frequent in Takotsubo patients[14]. In contrast, studies of TTC in animal models are easy to run in homogeneous populations using uniform stress triggers. Several Takotsubo-like models have been proposed in rodents, the more robust and reproducible being based on the administration of catecholamines in different routes, frequency and dose[15,16].

Considering that little is known about the regulation of glucose metabolism of the myocardium during the early and late phases of Takotsubo, and, more importantly, that even less is known on the influence of metabolism on the functional and structural integrity of the myocardium in the long term after Takotsubo, we performed a comprehensive set of non-invasive explorations for close longitudinal follow-up in the same animals before stress (baseline), during acute stress (2 h) and at early (7 days) and late (1 and 3 months) time points after stress. Our objective was to decipher the role of energy metabolism on long-term tissue and vascular remodeling[17,18] in a preclinical model recapitulating the clinical signs of TTC over the whole course of the disease. Reasoning that acute stress was the cause of Takotsubo, at least

in its initial description[6], and that there is a large consensus that stress-induced catecholamine release is the main causal factor of Takotsubo, we mimicked in rats the catecholamine rush described in patients[19], with a single administration of the adrenergic agonist isoproterenol, a drug used to increase heart rate in case of bradyarrhythmias[15] that can provoke Takotsubo[20]. Here, we show that the increased glucose uptake in the myocardium induced by an isoprenaline stress is not used to increase energy production, but is diverted into alternative anabolic pathways of glucose. We provide evidence that this glucose diversion induces immediate and long-term tissue, metabolic and functional changes that may explain the increased risk of heart failure in Takotsubo patients.

## Results

### Acute pharmacological challenge recapitulates Takotsubo

In young adult female rats, one single intraperitoneal injection of the catecholaminergic drug isoprenaline (ISO; 50 mg/kg, a non-lethal dose) induced typical signs of acute stress. Glycaemia, cardiac and respiratory rates were significantly increased 2 h post-ISO (Supplementary Fig. 1). As expected, biomarkers of the acute catecholamine surge were present[13,21], including increased plasmatic concentrations of glycerol, triglyceride, non-esterified fatty acids and lactate (Supplementary Fig. 1). Glycogen deposits significantly decreased in both apical (−64%) and basal regions (−57%) (Supplementary Fig. 2), while lipid deposits significantly increased in the apex (threefold) and in the base (11-fold) of the heart (Supplementary Fig. 2). The plasmatic concentrations of Alanine-Amino Transferase (ALAT), Aspartate-Amino Transferase (ASAT) and creatinine kinase (CK) all increased 2 h post-ISO, reflecting an acute myocardial damage. The plasmatic concentration of brain natriuretic peptide (BNP) was unchanged during the acute stress phase (Supplementary Fig. 1).

During the acute phase, electrocardiograms (ECG) of the heart were characteristic of impaired ventricular contraction and myocardial dysfunction: significant shortening of the RR interval concomitant with an increased heart rate, a substantial prolongation of the QRS interval, and a highly variable S wave amplitude with prolongation of the ST segment (Supplementary Fig. 1). High-resolution ultrasound showed increased left ventricular (LV) ejection fraction and fractional shortening along with notably reduced end-diastolic and end-systolic LV volumes. Systolic arterial blood pressure and cardiac output were low, and systemic arterial resistances were high, although not statistically different from pre-stress values (Supplementary Fig. 3).

The original description of TTC in patients 30 years ago[3] reported a spectacular apical ballooning of the LV shaping the heart into a Japanese octopus fishing pot (called a *tako-tsubo* in Japanese). Nowadays it is recognized that not all TTC hearts show apical ballooning and that some patients may present medial, basal or global cardiac ballooning, or even no visible ballooning of the LV[14]. In rodents with fast-beating hearts, LV ballooning is difficult to evidence. However, in our rat model a quantitative analysis of cardiac magnetic resonance (CMR) cine sequences enabled global and segmental measurement of strains in the three myocardial fiber orientations (Fig. 1). Two hours post-ISO, longitudinal strain (LS) increased in the LV and decreased in the right ventricle (RV). As both ventricles share the same interventricular septum, these observations suggest hypercontractility of the LV wall and especially a reduction of the deformation of the RV free wall[22] (Fig. 1). In the right atrium (RA), the strain decreased, reflecting RA dysfunction (Fig. 1). Interestingly, in the LV both the longitudinal strain (LS) and the circumferential strain (CS) varied regionally: both significantly increased in the basal segment of the LV whereas they remained unchanged or decreased in the apical segment (Fig. 1). Taken together, myocardial deformation during the acute phase reflects simultaneous hypercontractility of the basal/mid segments of LV and hypocontractility of the apex, in line with the transitory LV apical ballooning observed in TTC patients[23,24].

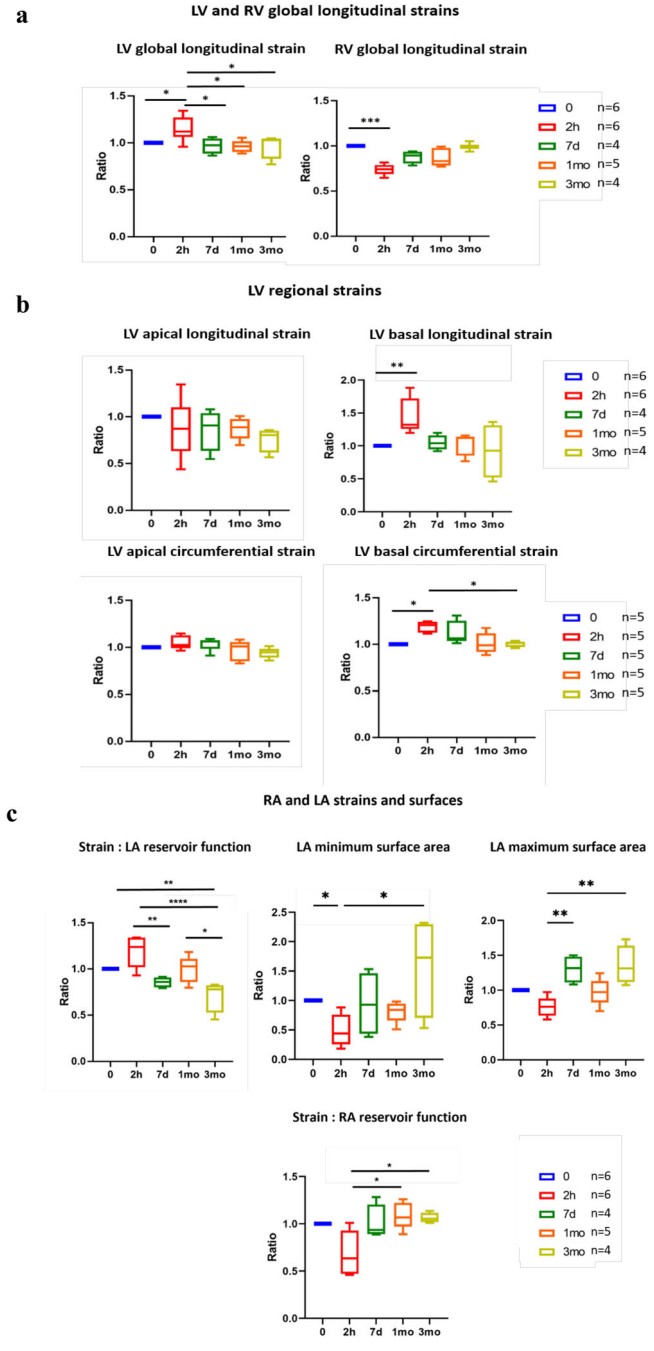

**a** LV and RV global longitudinal strains

**b** LV regional strains

**c** RA and LA strains and surfaces

**Fig. 1 | Cardiac strains in LV, RV, LA, and RA at baseline and 2 h, 7 d, 1 mo, and 3 mo post ISO.** Boxplots showing median, 25 and 75 percentiles, and extremes of values of $n = 4$ to 6 animals per time point. **a** Longitudinal strain is increased in the LV at 2 h and returns to normal in the remaining time-points (0 vs. 2 h: $p = 0.0428$, 2 h vs. 7d: $p = 0.0379$, 2 h vs. 1mo: $p = 0.0175$, and 2 h vs. 3mo: $p = 0.0385$). In contrast, longitudinal strain in the RV decreases at 2 h and then returns progressively to baseline values (0 vs. 2 h: $p = 0.0002$). **b** Longitudinal and circumferential strains in the apex and in the basal region. Hyperkinesia, the normal reaction to a catecholaminergic surge is observed at 2 h in the basal region highlighted by the increase in the basal longitudinal (0 vs. 2 h: $p = 0.0081$) and circumferential (0 vs. 2 h: $p = 0.0107$, and 2 h vs. 3mo: $p = 0.0185$) strains, while at the same time no substantial changes were found in the longitudinal and circumferential strains in the apex. **c** LA and RA strains and LA surface area. LA strain decreases progressively after the initial rise at 2 h, suggesting an irreversible LA dysfunction (0 vs. 3mo: $p = 0.0098$, 2 h vs. 7d: $p = 0.0038$, 2 h vs. 3mo: $p < 0.0001$, and 1mo vs. 3mo: $p = 0.0203$). It goes along with the increase of LA minimum (0 vs. 2 h: $p = 0.0447$, and 2 h vs. 3mo: $p = 0.0357$) and maximum (2 h vs. 7d: $p = 0.0038$ and 2 h vs. 3mo: $p = 0.0031$) surface areas over time. The strain that corresponds to the reservoir function decreases in the RA at 2 h (2 h vs. 1mo: $p = 0.0310$, and 2 h vs. 3mo: $p = 0.0376$). Unpaired comparison tests: $*p < 0.05$, $**p < 0.01$ and $***p < 0.001$. All values are individually normalized to the baseline value for each animal. Statistical significance ($p < 0.05$) for each variable was estimated by one-way or two-way ANOVA when group variances were equal (Bartlett test); if not the non-parametric Kruskall–Wallis test, and the Holm multiple comparisons test was used to execute simultaneous t-tests. Source data are provided as a « SourceData_Figure1 » file.

The main sign for differential diagnosis between TTC and acute coronary syndrome is the absence of coronary obstruction visible on coronarography in TTC patients. In rodents, coronary obstruction can be assessed by measuring the difference in the coronary flow (CF) under basal and hyperemic conditions (CF during hyperemia – CF pre-hyperemia)/ CF pre-hyperemia, which corresponds to the coronary flow reserve (CFR, Supplementary Fig. 4). Two hours post-ISO, the peak velocity of coronary flow was increased with respect to pre-ISO (baseline). However, at that time, it was no further increased by the inhalation of 5% isoflurane (hyperemic enhancement) in contrast to the pre-stress response to hyperemia (Supplementary Fig. 4). The CFR was lower during the acute post-stress phase and recovered its baseline level at 7 days post-ISO. Therefore, the vasodilatation induced by isoprenaline was already maximal at 2 h post-ISO, which confirms the absence of coronary obstruction during the acute post-stress phase.

Ultra-structurally, transmission electron microscopy revealed discrete signs of LV tissue remodeling at 2 h post-ISO reflecting the mechanical stress induced by the hypercontractility of the LV wall: mainly signs of disruption of intercalated discs and loss of cell–cell adhesion (Supplementary Fig. 5).

## The Takotsubo cardiomyopathy is partly reversible

All the signs of acute cardiac stress observed at 2 h post-ISO were re-analyzed at 7d post-ISO. Glycaemia, cardiac and respiratory rates, blood pressure, global and segmental cardiac strains, and strain rates of the LV and RV returned to pre-ISO (baseline) levels (Supplementary Figs. 1 and 3, Fig. 1). The ECG and the plasmatic levels of ALAT, ASAT and CK returned to baseline; however, the plasmatic concentration of brain natriuretic peptide (BNP) was normal at 2 h post-ISO and increased at 7d post-ISO, suggesting a delayed LV dysfunction (Supplementary Fig. 1). Parameters derived from cardiac ultrasound images were normal at 7d post-ISO (Supplementary Figs. 3 and 4). In short, the major parameters of cardiac function and physiology had reversed to normal values, as typically described in TTC patients. Ultrastructural signs reflecting mechanical stress were not found at 7 days post-ISO, however fibrosis appeared in the LV apex (Fig. 2, Supplementary Figs. 5 and 6) but not in the LV base or in atria (Fig. 2). Taken together, these results on animal model strongly corroborate the clinical signs of Takotsubo, including the reversibility of acute signs of cardiac stress, although it cannot be excluded that diffuse myocardial fibrosis is a potential primer of cardiac dysfunction.

## Acute stress induces immediate metabolic remodeling

Two hours post the acute administration of ISO, positron emission tomography (PET) imaging showed a 47% increase in myocardial uptake of the glucose analogue 2′-[[18]F]-fluoro-2′-deoxy-D-glucose (FDG) relatively to baseline (Fig. 3). A two-tissue compartmental analysis of FDG uptake kinetics[25] showed that K1, the rate constant for FDG passage from blood to tissue, increased by 70% at 2 h post-ISO. However, k3, the constant that reflects phosphorylation of FDG by hexokinase, was reduced. Staining for Glut1 significantly increased 2 h post-ISO, with a slight predominance in the apical over the basal region of the heart, while staining for Glut4 was unchanged (Fig. 4).

This increase of the entry of glucose into myocardial cells occurred simultaneously with a decrease in the rate of glucose phosphorylation, with lipid deposits and with low glycogenolysis demonstrated using oil-red and periodic acid Shiff stainings, respectively. Similar observations

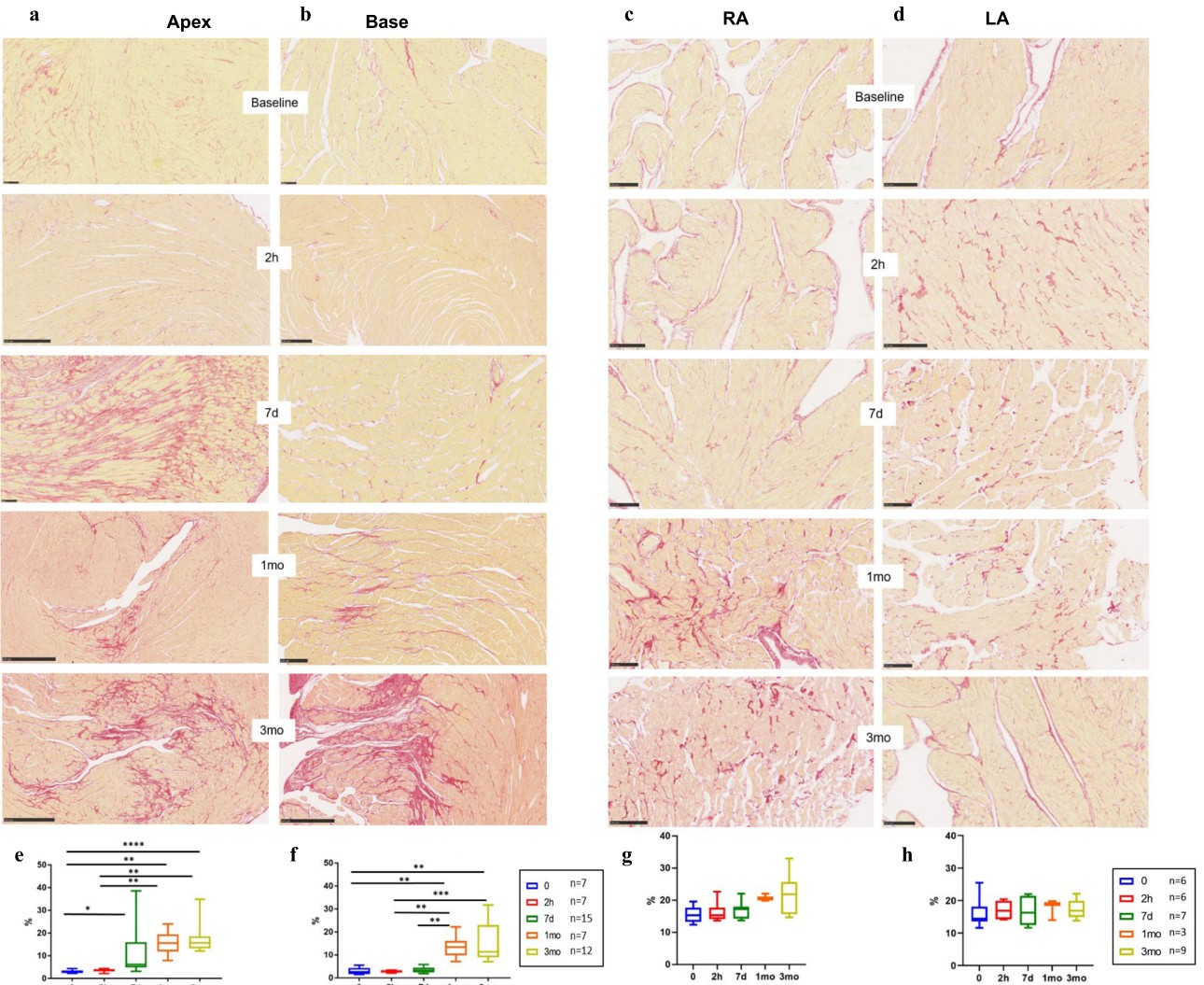

**Fig. 2 | Progression of cardiac fibrosis after ISO stress. a–d** Representative section of Sirius red staining in the LV apex and base, right atrium and left atrium of the heart, at baseline (scale bars of a length of 100 µm) and 2 h (scale bars of a length of 500 µm in LV images and of 100 µm in LA and RA images), 7d (scale bars of a length of 100 µm) and 1-month (scale bars of a length of 500 µm in LV apical and 250 µm in LV basal images, and of 100 µm in LA and RA images,) and 3-months (scale bars of a length of 500 µm in LV images and of 100 µm in LA and RA images) post-ISO. **e–h** Quantitative analysis of staining using FIBER-ML[71] in the LV apex and base, right atrium and left atrium of the heart at the corresponding post-ISO time points for the indicated number of animals represented as boxplots showing median, 25 and 75 percentiles, and extremes of values. The diffuse fibrosis that first appears in the

LV apex at 7d post-ISO augments and extends into the LV base at 1 and 3 months (in the apex, 0 vs. 7d: $p = 0.0353$, 0 vs. 1mo: $p = 0.0011$, 0 vs. 3mo: $p = 0.0001$, 2 h vs. 1mo: $p = 0.0073$, and 2 h vs. 3mo: $p = 0.0012$; in the base, 0 vs. 1mo: $p = 0.0059$, 0 vs. 3mo: $p = 0.0015$, 2 h vs. 1mo: $p = 0.0038$, 2 h vs. 3mo: $p = 0.0009$, 7d vs. 1mo: $p = 0.0043$, and 7d vs. 3mo: $p = 0.0005$); unpaired comparison tests: *$p < 0.05$, **$p < 0.01$ and ***$p < 0.001$. Statistical significance ($p < 0.05$) for each variable was estimated by one-way or two-way ANOVA when group variances were equal (Bartlett test); if not the non-parametric Kruskall–Wallis test, and the Holm multiple comparisons test was used to execute simultaneous $t$-tests. Diffuse fibrosis is also apparent in the LA at 1- and 3-months post-ISO: ANOVA test, $p = 0.0294$. Source data are provided as a « SourceData_Figure5 » file.

were reported[26,27] and correspond to an immediate myocardial metabolic response[28] that goes along with the catecholamine-induced heart transient dysfunction[15].

## A metabolic paradox: more glucose-6-phosphate with lower glycolysis

During the recovery stage (7d post -ISO), Glut1 staining was high in the apex but returned to baseline in the basal region, as well as Glut4 staining. However, a potential increase in Glut4 expression level was observed in the LV base at 7d compared to the baseline (+107%). FDG uptake and K1 remained at their acute phase levels, but remarkably, k3, which was below baseline at 2 h, rose to higher than baseline levels at 7d, in the apex: +34%, and in the base +44% (Fig. 3). In line with this observation, the expression of hexokinase 2 (Hk2) increased in the apex at 7d post-ISO (Fig. 5). Of note, the levels of the inflammatory cell

biomarker Cd68 at 7d post-ISO were modest (0.1% in the apex and 0.04% in the base), and not significantly different from the 2-hour acute phase value, and Glut1 and 4 stained essentially cardiomyocytes (Fig. 4), which is not in favor of a major contribution of inflammatory cells to cardiac FDG uptake (Fig. 4). Taken together, these results indicate a switch in glucose metabolism during recovery from stress, that leads to myocardial accumulation of glucose and of its phosphorylated form, glucose-6-phosphate (G6P), mainly in the LV apex.

At 7d post-ISO, proteomic analyses indicate inactivation of glycolysis and oxidative phosphorylation and over-activation of glucose alternative pathways, namely the hexosamines biosynthetic (HBP) and polyol pathways, implying a diversion of glucose to anabolic pathways that involves only 3–5% of glucose under normal physiological conditions[29]. Indeed, the main proteins involved in glycolysis and oxidative phosphorylation were downregulated and large amounts of

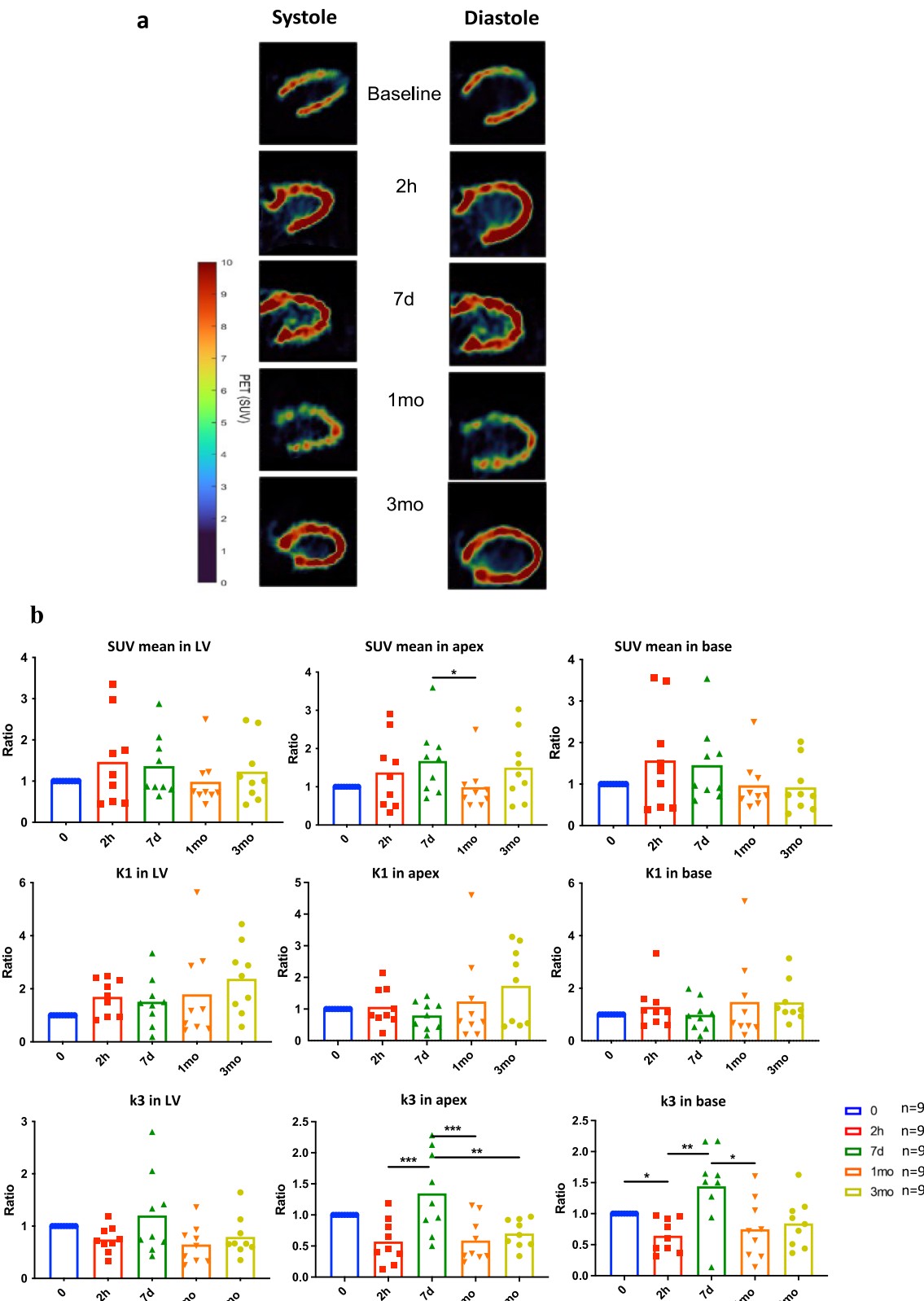

myocardial G6P were diverted to alternative pathways (Fig. 5, Supplementary Table 1). Contrasting with increased glucose phosphorylation (increased HK2 and k3), proteomic analysis at 2 h post-ISO revealed reduced expression of Gapdh, Pfkm, and Pgam2 (Fig. 5). At the same time, there was an activation of the HBP (overexpression of Pgm3, Uap1l1, Ogt, and Oga), and of the polyol pathway (overexpression of the rate-limiting enzyme aldose reductase, and of

sorbitol dehydrogenase) (Fig. 5). Interestingly, the expression of Gfat1, the limiting enzyme of the HBP, also increased (+74%) at 7d post-ISO in the apical region, as compared to baseline (Fig. 5). Gfat2 has recently been reported to mediate cardiac hypertrophy in mice subjected to chronic ISO stress (one-week infusion at 15 mg.kg$^{-1}$.day$^{-1}$)[30]. In rats, Gfat1 is the primary cardiomyocyte isoform responsible for stress-induced protein O-GlcNAcylation while Gfat2 is only present in cardiac

**Fig. 3 | Longitudinal FDG PET before, during and after ISO stress.**
**a** Representative images of FDG PET registered UUI (PETRUS) at the indicated time points post-ISO from diastolic phase to systolic phase. Images acquired 30 min after FDG injection of one section along the long axis of the LV. Color scale depicts the Standard Uptake Value (SUV) from 0 to 10. Note the increase in FDG uptake at 2 h and 7d respective to baseline. **b** Data are presented as mean values +/− SD of $n = 9$ animals per time point. Quantitative analysis of FDG uptake in the LV: SUV mean in whole LV, apex (7d vs. 1mo: $p = 0.0337$) and basal LV, normalized to baseline values; rate constant K1 reflecting the exchange of FDG from blood to tissue; rate constant k3 reflecting FDG phosphorylation, calculated using two-compartmental analysis. Note the global increase of SUV at 2 h and 7d post-ISO, the modest increase of K1 that is not statistically significant from baseline at any time point, and the significant decrease in k3 at 2 h followed by an increase at 7d post-ISO (in the apex, 2 h vs. 7d: $p = 0.0004$, 7d vs. 1mo: $p = 0.0005$, and 7d vs. 3mo: $p = 0.0029$; in the base, 0 vs. 2 h: $p = 0.0312$, 2 h vs. 7d: $p = 0.0096$, and 7d vs. 1mo: $p = 0.0312$). Paired comparison tests: $*p < 0.05$, $**p < 0.01$ and $***p < 0.001$. Statistical significance ($p < 0.05$) for each variable was estimated by one-way or two-way ANOVA when group variances were equal (Bartlett test); if not the non-parametric Kruskall−Wallis test, and the Holm multiple comparisons test was used to execute simultaneous $t$-tests. Source data are provided as a « SourceData_Figure2 » file.

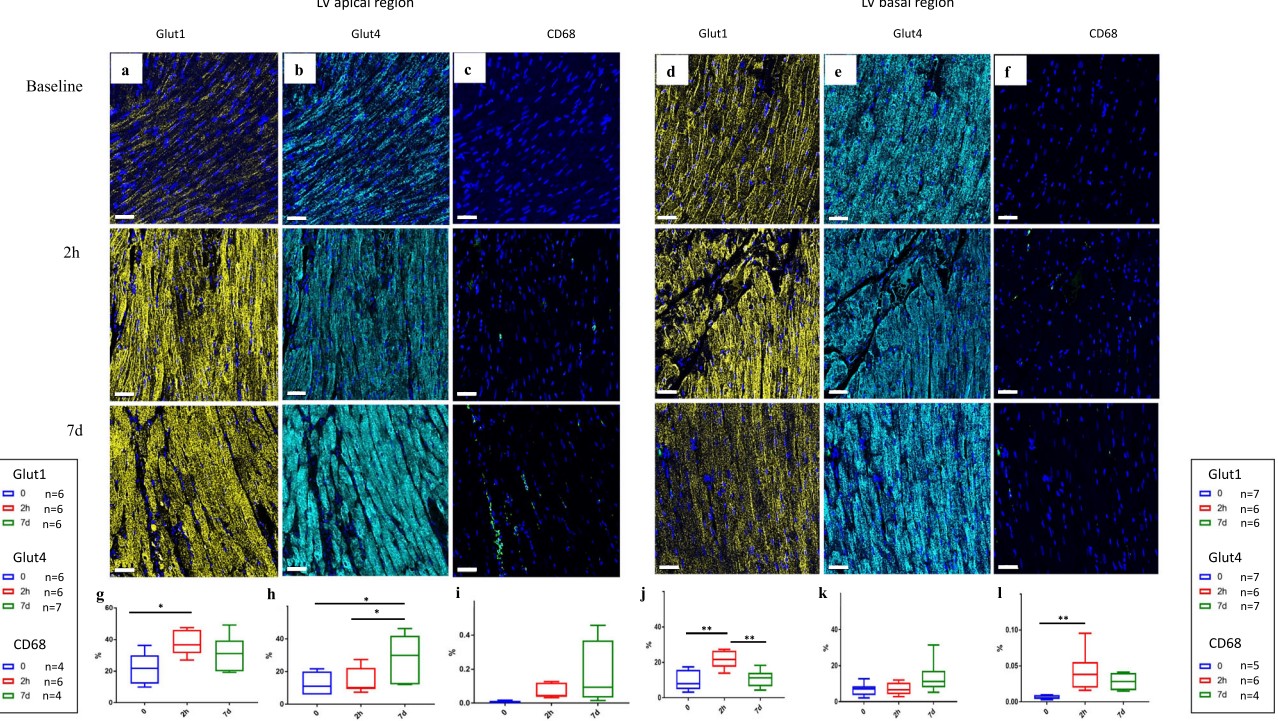

**Fig. 4 | Immunohistochemistry of Glut1, Glut4 and Cd68 expression at baseline, 2 h and 7d post-ISO.** **a–f** Representative fields of view of 4 µm cardiac sections stained for Glut1, Glut4 and Cd68. **a–c** apex, **d–f** base; the three stainings were performed in each section (OPAL® technology). **g–l** Quantification of staining densities (medians, 25 and 75 percentiles, and extremes of values of the percentage of section surface staining for each protein) of 4 to 7 animals for Glut1, Glut4 and Cd68. **g**, **h** apex, **j-l**: base. Note the significant increase in Glut1 expression at 2 h post-ISO followed by a return to baseline values in the apex (0 vs. 2 h: $p = 0.0307$) and the base (0 vs. 2 h: $p = 0.0020$, and 2 h vs. 7d: $p = 0.0050$). In contrast, Glut4 expression is unchanged from baseline at 2 h post-ISO but increases at 7d in the apex (0 vs. 7d: $p = 0.0196$, and 2 h vs. 7d: $p = 0.0362$). Disperse Cd68+ inflammatory cells are observed at 2 h (apex + basal, in the base: 0 vs. 2 h: $p = 0.0063$) and 7d (apex). Note that Glut1 and Glut4 stain essentially myocardial cells. Unpaired comparison tests: $*p < 0.05$, $**p < 0.01$ and $***p < 0.001$. Statistical significance ($p < 0.05$) for each variable was estimated by one-way or two-way ANOVA when group variances were equal (Bartlett test); if not the non-parametric Kruskall−Wallis test, and the Holm multiple comparisons test was used to execute simultaneous $t$-tests. Indicated scale bars in the images correspond to a length of 40 µm. Source data are provided as a « SourceData_Figure3 » file.

fibroblasts[31]. These observations were confirmed by western blot quantifications of the rate-limiting enzymes of the HBP, Gfat1 and Gfat2, which were highly expressed (respectively +74% and +469%) at 7d post-ISO in the apex compared to the control group and by increased O-GlcNAc levels (Fig. 5b), suggesting an overactivation of the HBP in both cardiomyocytes and myofibroblasts. The switch in myocardial glucose metabolism may have important consequences for myocardial structure and function. In parallel, proteomic analyses point to the overactivation of certain signaling pathways involved in myocardial tissue and vascular remodeling. Accordingly, we observed dysregulation of regulatory cascades known to be induced by O-GlcNAcylation[32–35], such as phosphatidylinositol 3-kinase (Pi3k)/Akt, Pkc, Ampk, Mapk, p38, Nf-kB, and insulin receptor substrate (Irs-1, Irs-2) (Supplementary Table 1) with, in parallel, activation or inactivation of canonical pathways of tissue remodeling, in particular the inactivation of Hippo signaling (Dlg1, Ppp1cb, Ppp1cc, Ppp2r3a, Pp2r5e,

Skp1, Ywhae, Ywhaq) (Supplementary Table 2). Hippo signaling induces the transition of cardiomyocytes into myofibroblasts, is essential in organ size control and tissue homeostasis and is directly regulated by HBP in response to high glucose uptake[36,37]. In addition, Hippo signaling maintains cardiac fibroblasts in a resting state, and its inactivation switches fibroblasts to active myofibroblasts that trigger fibrosis[38]. We also observed increased Rho signaling, a regulator of actin-based motility, increased sphingosine-1-phosphate signaling that modulates vascular tone, endothelial function and integrity, as well as lymphocyte trafficking; and increased Vegf signaling that triggers neo-angiogenesis (Supplementary Table 3). Accordingly, immuno-fluorescent staining for the endothelial proliferation marker Cd31 was significantly increased, as well as that of the smooth muscle cell activation marker alpha Sma. In the LV, Cd31 and alpha-Sma remained in proximity as the capillary density increased, suggesting coordinated vascular growth (Fig. 6). Taken together, these results support that

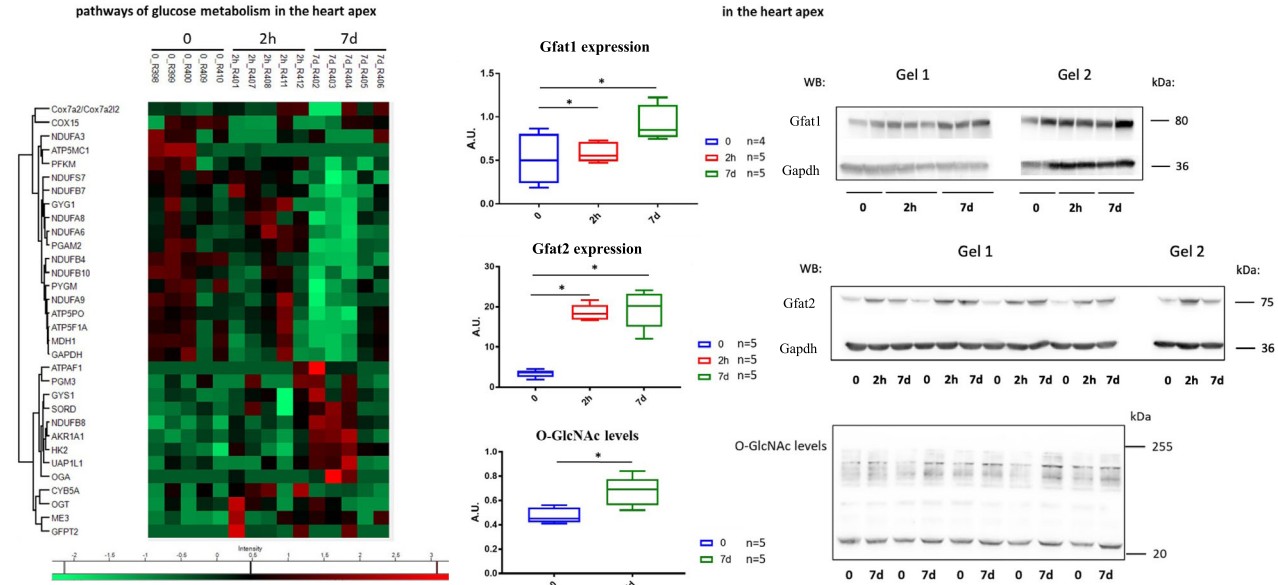

**a** Hierarchical clustering with all proteins of interest involved in different pathways of glucose metabolism in the heart apex

**b** Immunoblots quantification of proteins of interest involved in Hexosamines Biosynthetic pathway in the heart apex

**Fig. 5 | Expression of major proteins of the different pathways of glucose metabolism in the heart apex. a** Hierarchical clustering with all proteins of interest involved in different pathways of glucose metabolism in the heart apex, based on Pearson correlation on z-scored quantification values. Heatmap indicating proteins involved in glycolysis, glycogenesis, gluconeogenesis, oxidative phosphorylation, hexosamine biosynthetic pathway, and polyol pathway in the LV apical segment. Colors depict the z-scored changes in protein expression at 2 h and 7d post-ISO respective to baseline expression levels: red, overexpression higher than 1.3-fold, green: under-expression lower than −1.3-fold. All values are statistically significant with $p < 0.05$ using two-tailed Student's $t$-test, $n = 5$ per time point. **b** Western blot analysis of Gfat1 (0 vs. 7d: $p = 0.0168$, and 2 h vs. 7d: $p = 0.0238$),

Gfat2 (0 vs. 2 h: $p = 0.0486$, and 0 vs. 7d: $p = 0.0112$) represented in boxplots showing median, 25 and 75 percentiles, and extremes of values (a.u.: arbitrary units) depicted a significant increase of their expression at 2 h and 7d post ISO. The levels of O-GlcNAcylated proteins (i.e., the effect of the HBP overactivation), were significantly increased at 7d post-ISO compared to the control groups ($p = 0.0118$); unpaired comparison tests: *$p < 0.05$, **$p < 0.01$ and ***$p < 0.001$. Statistical significance ($p < 0.05$) for each variable was estimated by one-way or two-way ANOVA when group variances were equal (Bartlett test); if not the non-parametric Kruskall–Wallis test, and the Holm multiple comparisons test was used to execute simultaneous $t$-tests. Source data are provided as a « SourceData_Figure4_WB » file.

HBP hyperactivation induced by an acute stress is a driver of cardiac fibrosis, dysfunction, and angiogenesis.

In parallel with hyperactivation of the polyol pathway in the apex at 7d post-ISO, proteomic analysis showed a decrease in the antioxidant glutathione pathways and an increase in the generation of nitrite oxide and reactive oxygen species (Fig. 5, Supplementary Table 2). The activation of these two pathways has been regarded as a consequence of the hyperactivation of the polyol pathway, generating fructose whose accumulation leads to the production of polyols and harmful metabolites such as advanced-glycation end products (AGEs)[39,40].

**Long-term TTC heart remodeling: metabolism, structure, and function**

Long-term observations at 1- and 3-months post-ISO confirmed the observations at 7d post-ISO (Supplementary Fig. 3). The most striking finding was the reinforcement of fibrosis and its extension to other regions of the heart. At 7d post-ISO, diffuse Sirius red staining concerned only the LV apex, but at 1mo it was found in the basal LV and the LA, where it dramatically increased at 3mo post-ISO (Fig. 2). This was in line with the gradual decrease of the longitudinal and circumferential strains in the apex, and with decreased LA strain and increased minimum and maximum surfaces of the LA (Fig. 1). Taken together, these results indicate a continuous tissue remodeling accompanying a gradual extension of fibrosis in the LA during the delayed post-stress phase, leading to irreversible degradation of the LA reservoir function and to LA enlargement, underlying biomarkers of diastolic dysfunction of the heart[41,42]. Similarly, PAS staining and FDG PET imaging indicate persistent metabolic remodeling in the long term (Fig. 3 and Supplementary Fig. 2a). Considering that

isoprenaline has an elimination half-life of 3–7 h, its plasmatic levels are negligible after one day. Thus, acute effects of isoprenaline inducing the metabolic, functional, vascular and tissue remodeling during the acute and early phases (2 h and 7d post-ISO) are crucial for later aggravation of the disease.

## Discussion

Takotsubo cardiomyopathy, induced by a variety of stress factors, is multiform in its clinical manifestations and cardiac localizations, and mostly concerns patients in the second half of their life, at a time when known or unknown comorbidities are often present[14]. While resorting to a Takotsubo-like small animal model is a reductionist approach, it offers fast and easily reproducible explorations in a homogeneous population for repeated longitudinal examinations in the same individual of the disease's natural history. This opens a unique window on the time course of progression.

In the acute phase (2 h post-ISO), all the animals showed typical signs of stress-induced cardiomyopathy, with ECG abnormalities and decreased blood pressure, heterogeneity in regional deformations of the LV, and glucose metabolism remodeling. Interestingly, quantitative compartmental analysis of FDG cardiac kinetics suggested that, during the acute phase, the global increase in cardiac uptake of FDG was the consequence of increased FDG entry, with increased rate constant K1 and increased Glut1 expression. While enhanced myocardial glucose uptake is expected in ISO-induced hyperglycemic conditions, surprisingly myocardial insulin-dependent Glut4 presented no changes in terms of its expression rate and translocation to the surface membrane. This observation goes along with the decreased level of PAS staining for glycogens, absence of fatty acids transporters, Fabp, expression's changes, and lipid toxicity shown by

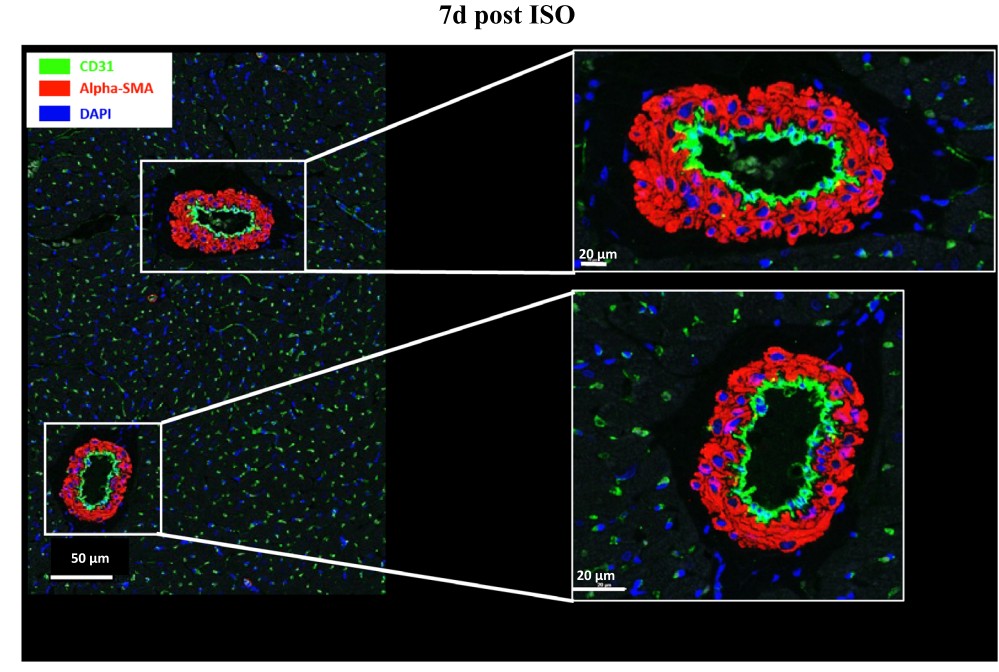

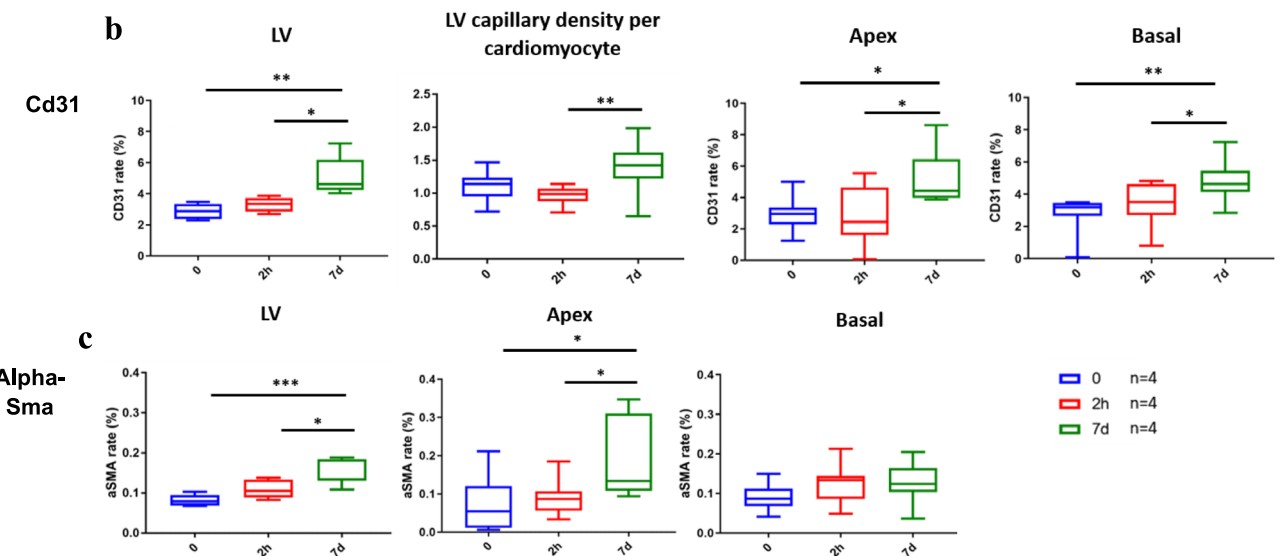

**Fig. 6 | Post-stress vascular remodeling of the LV. a** Representative 4 μm-section of the LV apex co-immunostained for Cd31 and alpha-Sma. The scale bar in the main image represents a length of 50 μm, while the scale bar in the two zoom-in images corresponds to a length of 20 μm. Quantitative analysis of Cd31 and alpha-Sma expression's rate represented as boxplots showing median, 25 and 75 percentiles, and extremes of values. No dissociation between the Cd31-stained endothelial layer and the alpha-Sma-stained smooth muscle cell layer was found. **b** Increase in Cd31 expression (in the LV, 0 vs. 7d: $p = 0.0013$, and 2 h vs. 7d: $p = 0.0318$; in the apex, 0 vs. 7d: $p = 0.0318$, and 2 h vs. 7d: $p = 0.0300$; in the base, 0 vs. 7d: $p = 0.0011$, and 2 h vs. 7d: $p = 0.0162$) and in the ratio of capillaries per cardiomyocyte (2 h vs. 7d: $p = 0.0003$) and **c** in alpha-Sma (in the LV, 0 vs. 7d: $p = 0.0004$, and 2 h vs. 7d: $p = 0.0209$; in the apex, 0 vs. 7d: $p = 0.0221$, and 2 h vs. 7d: $p = 0.0369$) at 7d post-ISO indicates endothelial proliferation and angiogenesis. Unpaired comparison tests: *$p < 0.05$, **$p < 0.01$ and ***$p < 0.001$. Statistical significance ($p < 0.05$) for each variable was estimated by one-way or two-way ANOVA when group variances were equal (Bartlett test); if not the non-parametric Kruskall–Wallis test, and the Holm multiple comparisons test was used to execute simultaneous $t$-tests. Source data are provided as a « SourceData_Figure6 » file.

the oil-red staining, indicating that compensation of energetic needs by fatty acid beta-oxidation, essentially a mitochondrial process, is highly unlikely. After acute ISO challenge, the energetic status of the heart resembles myocardial glucose intolerance, reported in diabetic hearts by Stratmann et al., in which lipid toxicity is caused by the myocardial decrease of glucose uptake and causes insulin resistance with decreased Glut4 activity[28].

Therefore, the heart under acute stress is in a situation of high glucose plasmatic concentrations that do not increase the capacity to use glucose as energy source through glycolysis. This paradoxical unbalanced situation leads to consider whether G6P could be diverted into alternative pathways of glucose metabolism.

Most of the signs and symptoms typical of myocardial stress present during the acute phase revert during the recovery phase 7d post-ISO. Functional signs and, except BNP, plasmatic biomarkers of heart suffering return to normal at 7d post-ISO, and plasmatic concentrations of energetic substrates are not different from those prior to stress. Overall, the general state of the animals returns to normal, as in patients during the recovery phase[23], although minor functional signs such as strain of the LA remain abnormal. It is plausible that in a

clinical setting, these parameters would be disregarded as being minor sequelae of the transient Takotsubo cardiomyopathy.

Cardiac metabolism has rarely been explored in TTC patients. Here, we document a high uptake of FDG, which has been considered a sign of cardiac suffering[11], high expression of Glut1 and Glut4 and increased glucose phosphorylation (k3) in the apex, together with paradoxically reduced energetic breakdown of glucose in glycolysis and oxidative phosphorylation. Overall, isoprenaline acting on ß1 adrenoreceptors creates a situation in which glycolysis is uncoupled from myocardial glucose levels[43]. In a rat model of TTC, Godsman et al. have described a dysregulation of glucose and lipids metabolic pathways, as well as inflammation and upregulation of remodeling pathways and fibrosis[43]. Our present results show the deviation of excess glucose and phosphorylated glucose to alternative and anabolic pathways, which under normal physiological conditions would contribute but marginally to glucose metabolism[29,44–48]. Indeed, the cardiac apex shows a dramatic activation of the alternative pathways of glucose utilization: sorbitol dehydrogenase (SDH), a NADH-dependent enzyme of the polyol pathway is overexpressed at 2 h and 7d post-ISO and aldose reductase (AR), the rate-limiting enzyme of this pathway overexpressed at 7d post-ISO; the expression of Gfat1 and Gfat2, the rate limiting enzymes of the HBP in cardiomyocytes and myofibroblasts, respectively, increase dramatically at 2 h post-ISO and remains high at 7d post-ISO. The O-GlcNAc levels increased significantly at 7d post-ISO in the LV apex. In rats, Gfat1 is the primary cardiomyocyte isoform responsible for stress-induced protein O-GlcNAcylation while Gfat2 is found only in cardiac fibroblasts[31]. This increase of both isoforms may regulate cardiac myofilaments, induce cardiomyocyte dysfunction, and generate fibrosis[31,46–48].

Following an acute ISO surge in mice, Liao et al. observed increased cardiac infiltration of pro-inflammatory monocytes and a significant reduction of anti-inflammatory (Tim4+/Lyve1+) cardiac resident macrophages[49]. In their interesting study, the blockade of monocyte infiltration or pro-inflammatory activation alleviated ISO-induced cardiac dysfunction. Hence, they suggested that monocyte infiltration was a major mechanism of TTC pathogenesis, although the contribution of other cells such as neutrophils, dendritic cells, lymphocytes, etc., was not excluded[49]. In the present study, we observed an immediate inflammatory response with the presence of Cd68+ macrophages early during the acute phase at 2 h post-ISO, i.e., at an earlier time point than those explored by Liao et al.[49], and also at 7d post-ISO with the activation of proinflammatory cytokines. Our results confirm that ISO induces an inflammatory response, the question being whether this is the main causal mechanism inducing Takotsubo-like ISO-induced pathology, or one of several mechanisms with contribution from other cells, including the cardiomyocytes themselves. In favor of a direct myocardial reaction to ISO, we show here that the metabolic remodeling concerns cardiomyocytes and does not colocalize with macrophages during the acute and early recovery phases post-ISO (see the cardiac section co-stained for Glut1, Glut4 and Cd68 in Fig. 4). On the other hand, we found as early as 2 h an increase in the expression of Gfat2, which has an anti-inflammatory role in macrophages[50].

In addition to Liao et al.'s study[49], several studies have explored other avenues of treatment in preclinical TTC-like models. Tsikas et al. suggested that GAA, a non-protein guanidino amino acid acting as analogue of Lys, could induce drastic changes in the heart and kidneys compensating energy requirements in case of insufficient creatine supply[51]. According to Anwar et al., Entresto®, an approved drug that prevents natriuretic peptide degradation, increases BNP levels, and decreases mortality in heart failure patients with reduced EF, decreased cardiac sympathetic activity, attenuated ISO-induced myocardial hypoperfusion, decreasing mortality[52]. On the other hand, a study by Ellison et al. focusing on the effects of β-adrenergic overload on cardiomyocytes and cardiac stem cells (CSCs), proposed that CSC activation observed at 3 and 6 d post-ISO contributed significantly to the rapid clinical recovery of the TTC heart[53]. Although they differ in terms of species, sex and time points, these unrelated observations, including ours, strongly suggest that a combination of myocardial metabolic remodeling, inflammation, and other mechanisms involving interactions between cardiac muscle, vessels and multiple resident and infiltrating cell types is responsible for late sequelae of Takotsubo. Further studies are needed to integrate these various mechanisms into a global operational scheme to drive treatment.

Simultaneously with the metabolic and inflammatory abnormalities, we observed extensive tissue and vascular remodeling in the apex of the LV already at 7d post-ISO: diffuse fibrosis and increase of endothelial and smooth muscle cells' vascular biomarkers. It thus appears that a single acute ISO stress is responsible for long term functional, structural, vascular, and molecular changes. The coincidence between tissue and vascular remodeling, and the massive glucose entry and phosphorylation, as well as the hyperactivation of the hexosamine biosynthetic and polyol pathways suggest an intricate link between these events, as reported in other studies. Indeed, overactivation of the HBP promotes cardiac hypertrophy in vitro, and significantly increases the size of cardiac cells, protein synthesis, and the expression of hypertrophy markers[46]. It was also showed that overactivation of Gfat, the rate-limiting enzyme of the HBP, resulted in increased heart size and fibrosis in Gfat1 transgenic mice[50]. These results and those from other studies support the view that this overactivation is responsible for the remodeling of the cytoskeleton, the ECM, and the cardiac vessels[50,54,55]. On another hand, the HBP generates uridine-diphosphate-N-acetylglucosamine (UDP-GlcNAc), the limiting substrate for O-GlcNAcylation, a post-translational modification playing an important regulatory role for O-linked β-Nacetylglucosamine (O-GlcNAc) proteins[54–59]. A significant increase in O-GlcNAcylation was notably observed in the hypertensive[60,61], diabetic[32,62], chronically hypertrophied heart[63], and in heart failure[60] and thought to contribute to contractile and mitochondrial dysfunction[63]. Fülöp et al. have shown that O-glycosylation of specific proteins contributes to the impairment of cardiomyocyte function in diabetes[64]. Accordingly, here myocardial O-Glycosylated proteins were significantly more abundant at 7d post-ISO in the apex (Fig. 5b), reflecting a high level of O-GlcNAcylation, in line with the overactivation of the HBP observed in proteomic and western blot analyses of Gfat1/2 (Fig. 5b).

The fibrotic conversion of the cardiac tissue that we have shown here definitively excludes the reversibility of stress-induced cardiac damage: fibrosis was present in the LV apex at 7d post-ISO and spread to the entire LV and to the LA at 1 and 3mo post-ISO. The change in the LA reservoir strain, along with the concomitant increase of the LA end-diastolic and end-systolic surfaces, indicate irreversible LA dysfunction and volume enlargement, which are known predictors of diastolic dysfunction[41]. This supports a long-term cardiac embrittlement following Takotsubo, and an evolution towards diastolic dysfunction predictive of a serious cardiac condition with reserved prognosis of Takotsubo patients[41,42].

Overall, longitudinal observations in this animal model Takotsubo-type cardiomyopathy triggered by a single injection of ISO, clearly reflect irreversible tissue, vascular, and metabolic sequelae that are prone to weaken the heart and render it susceptible to recurrent cardiac disease[5]. The present work points to alternative pathways of glucose metabolism as the critical mechanism by which the one-time stressed heart engages in a continuous degradation of structure and function. These pathways may represent attractive targets to prevent the progression of cardiac damage following an acute stress. The rapid initial changes in metabolic control and early appearance of diffuse apical fibrosis suggest that fibrosis progression, vascular network alteration, and metabolic remodeling are initiated early after stress. Therefore, preventing deleterious evolution of TTC may require treatment to be administered during or immediately after the acute phase of TTC.

This calls for careful attention to the management of TTC patients, whose incidence has been reported to increase recently[5,23,65,66]. Our study also highlights the importance of myocardial strain measurements and FDG PET imaging to assess and detect both regional and global functional, tissue, and metabolic remodeling in the TTC heart. The outcomes reported here in an animal model are in favor of clinical trials using these two in vivo and non-invasive imaging modalities is crucial for a better management of TTC patients, and the optimization of the diagnosis and prognosis of Takotsubo.

Clinically, Takotsubo is multifactorial and complex and often associated with different comorbidities, The present study used a simple animal model to investigate the longitudinal functional, anatomical, metabolic, tissue, and vascular modifications of the heart in a reproducible Takotsubo-type animal model, without the various comorbidities encountered in patients, such as the cardiovascular diseases associated with aging, hypertension, diabetes, obesity, etc. A unique injection of beta-adrenergic catecholamines led to a cascade of reproducible activation and inactivation of different metabolic and anabolic pathways of glucose and to short- and long-term tissue, fibrotic and vascular sequelae of the heart. In short, the stress induced a transient ventricular dysfunction during the acute phase, and an irreversible tissue and functional impairment of the left atrium in the long term. The proteomic analysis at 2 h and 7 days were based on 5 replicates per condition, which may lead some samples to yield less protein raw identification than the average across all samples. However, in the present study we used the proteomics data to build hypotheses regarding energetic pathway changes and interpreted these in the light of the in vivo and ex vivo imaging, and of the physiological and biochemical analysis. This reasonable use of proteomics-derived data does suggest clues/leads about pathophysiological mechanisms in TTC, which naturally remain to be examined in the light of other proposed mechanisms such as, e.g., direct lipidotoxicity[27], impairment of the mitochondrial respiratory chain[67] innate immune reactions[46], modifications of calcium signaling[62,68], and others. Importantly, as with any reductionist approach, transposition into the clinical realm should be weighted for its relevance with respect to clinical translation. Among the limitations of the model used here, we used young adult female rats, while TTC is most frequently observed in post-menopausal female patients. However, TTC, has also been described in young females and in males, therefore our study may benefit from its extension to male and to older animals. Secondly, the animal model used here showcases the consequences of a single triggering factor of TTC, a sudden catecholaminergic rush, while Takotsubo has been associated with mental and physical stress, surgical stress, neurological disorders, pheochromocytoma, etc.[14]. No single model can encompass the whole TTC spectrum, the variable severity of the disease and of its progression, and the role of inducing stressors[5,14]. Comparing the results from the present animal model, with different forms of TTC in individual patients and with the original description of emotional stress-induced TTC[14,65,66,69], is certainly a challenge of future TTC studies.

## Methods

### Animals

Experiments were approved by the French Animal Ethics Committee (agreement number 19064). One hundred and twenty 12-week-old female Wistar rats were obtained from Janvier (Le Genest-St-Isle, France) and housed in our facility on a 12:12 light-dark cycle, controlled temperature, 25 °C, 60% air humidity, and free access to food (M-bricks normal chow from TAPVEI, Kiili Fajumaa, Estonia) and water. Our longitudinal study of myocardial remodeling focused on the female TTC-like animal model for several reasons. The main reason is that TTC predominantly affects female patients (about 90% of TTC cases)[14]. In addition, male rodents depict more variability in functional parameters following ISO injection than female rodents[70]. Kneale et al.

speculate that this is the consequence of a different sensitivity to adrenergic stimulation between males and females[70,71]. The purpose of our study was to explore metabolic, functional, tissue, and vascular remodeling cascades following a viable dose of ISO in small groups of animals, therefore we prioritized females for a better homogeneity between individuals and for the closest match with the clinical incidence of TTC.

### Induction of Takotsubo-like syndrome in animals

Imaging, physiological parameters, blood chemistry, histology, western blots and proteomics were obtained before stress induction (day 0, baseline). A unique intraperitoneal injection of 50 mg/kg isoprenaline (ISO, Isoproterenol hydrochloride, Sigma-Aldrich, Germany) was performed on day 1. Imaging, physiological parameters, blood chemistry, histology, western blots and proteomics were obtained 2 h and 7 days post-ISO and then repeated at 1 month and 3 months post-ISO in the same animals. Two rats died during MRI imaging sessions, one before ISO injection from anesthesia, and one rat died 2 h post-ISO (1% mortality rate).

### FDG positron emission tomography (PET, Supplementary Fig. 7)

On days 0, 1, 7 and at 1 and 3 months, non-fasted rats were anesthetized (isoflurane 4% induction and 2% maintenance in air), weighted and glycemia and animal temperature were recorded. The animal was placed supine in a nanoScan PET-CT scanner (Mediso Medical Imaging Systems, Hungary) with respiratory and cardiac monitoring. A commercial ultrasound probe (SuperLinear™ SLH20-6, Supersonic Imagine, France, central frequency 15 MHz) connected to a small animal ultrasound device (Aixplorer, Supersonic Imagine, France) was positioned on the depilated chest of the animal to obtain a view of the full long-axis of the beating heart. The animal was then moved in the PET gantry and a whole-body X-ray tomodensitometry (CT) was acquired using the following acquisition parameters: semi-circular mode, 70 kV tension, 720 projections full scan, 300 ms per projection, binning 1:4. Images were reconstructed by filtered retro-projection (filter: Cosine; Cutoff: 100%) using Nucline version 3.00.010.0000 (Mediso Medical Imaging Systems, Hungary). Immediately after CT acquisition, a 30 min dynamic PET scan and 30 s after starting the acquisition, $32.9 \pm 1.4$ MBq of 2′-deoxy-2′-[$^{18}$F]fluoro-D-glucose (FDG; Advanced Applied Applications, France) in 0.4 mL saline were injected in the lateral tail vein. At the end of the first dynamic PET scan, a static 30 min PET scan was acquired with ECG and respiratory gating. PET data was collected in list mode and binned using a 5 ns time window, with a 400-600 keV energy window and a 1:5 coincidence mode. Data was reconstructed using the Tera-Tomo reconstruction software (3D-OSEM based manufactured customized algorithm) with expectation maximization iterations, scatter, and attenuation correction. The 30 min dynamic PET exam was reconstructed in 23 frames as follows: 30 s; $6 \times 5$ s; $4 \times 10$ s; $6 \times 30$ s; $3 \times 120$ s; $4 \times 300$ s. The ECG-gated cardiac PET was reconstructed in a single frame of 15 min, 45–60 min post FDG injection. Using the PET/CT fusion slices, volumes-of-interest (VOI) were delineated for the left ventricle (LV) using PMOD software (PMOD Technologies Ltd, version 3.8, Zürich, Switzerland). FDG uptake was quantified as Standard Uptake Value. The Peak SUV was calculated as the maximum average SUV within a 1-cm$^3$ spherical VOI, and the LV volume was automatically segmented at 40% of this value. Compartmental kinetic assessment of FDG uptake was based on the 2-tissue compartment model of the PMOD kinetics package with a lump constant set to 1.

### Positron emission tomography registered ultrafast sonography

The high temporal sampling of ultrafast ultrasound imaging (UUI) and its unique spatial resolution (~0.1 mm) were used as a priori anatomical information conducive to correct movement and partial volume effect of cardiac PET images using Super-Resolution (SR) methods[72,73].

B-mode images were acquired during the acquisition of the dynamic PET scan, over 1 s, using Aixplorer® (SuperSonic Imagine, France) in order to sample the whole cardiac cycle. Respiratory and cardiac motions were monitored during the acquisitions.

## Cardiac magnetic resonance imaging

Cardiac magnetic resonance (CMR) imaging acquisitions were performed in a preclinical 4.7 T MRI system (Bruker BioSpec 47/40 USR, Ettlingen, Germany) with a 7 cm inner diameter resonator for emission and a phase array surface coil for reception under isoflurane anaesthesia as above. Respiration, heart rate, rectal temperature and isoflurane delivery were monitored constantly and maintained stable during the acquisitions. LV function was measured by cine T1-weighted cine sequences including short axis stack of 6–7 slices covering the left and right ventricles from base to apex, along with conventional 2-chamber and a 4-chamber views, using the following scan parameters: TR = 7.2 ms; TE = 2.7 ms; flip angle=18.0°; field of view (FOV) = 60 × 60 mm; matrix size = 256 × 256, planar resolution = 235 μm, oversampling = 100, number of frames = 16 per heart cycle, slice thickness = 2 mm. Total scan time was 1min32s per slice. The cine intra-gate sequence was triggered using averaged heart and respiration rates in a retrospective reconstruction. Acquisitions were performed using Paravision software 6.0.1 (Bruker, Ettlingen, Germany) and T1 cine images were analyzed manually using Circle civ42 software (version 5.13.5, Circle Cardiovascular Imaging Inc., Canada) to estimate global heart chamber volumes.

## Cardiac strain and strain rate analysis

Cardiac strain defines the level of the wall deformation of a cardiac cavity along the cardiac cycle. It is a non-volumetric parameter, complementary to the conventional functional parameters, allowing to study different spatial components of the myocardial contraction function. Strain can be measured by the speckle tracking method using ultrasound[70], or by MRI that offers the advantage of an observer-independent complete coverage of the heart[74]. In our study, we opted for strain measurements based on T1 cine sequences using a semi-automatic homemade software, CardioTrack (Sorbonne University, Paris, France)[75,76], based on feature tracking[77] and written in Matlab® (The MathWorks, Natick, MA, USA). The strain measurements of the left and right ventricles and of the atria were performed using the 4 chamber views for reliability and repeatability reasons. We also determined on this view the minimum and maximum atrial surfaces, as surrogates of the atrial volumes. Missing data correspond only to the strain values that were incalculable for technical reasons, e.g., incorrect orientation or improper ECG gating, but no data was removed to ameliorate statistical significance.

## Sample preparation for histology

The LV and atria were excised and fixed during 24 h in 4% formaldehyde, then transferred to 70% EtOH and embedded in paraffin. For oil-red staining, they were fixed in an isopentane bath placed in liquid nitrogen. Hearts were cut in 4 μm thick section with a microtome (RM2145, Leica, Germany) for paraffin embedded tissue, or at 6 μm in a cryostat (CM3050 S, Leica, Germany) for frozen tissue. For each heart, 6 sections were taken at 5 positions spaced 100 μm apart in the apex, 6 sections at 3 positions spaced 200 μm apart in the basal region, and 6 sections at 4 positions spaced 50 μm apart in the atria.

## Immunohistochemistry (IHC)

Each cardiac section was stained in triplicate for red Sirius. Sirius red staining was performed in an automate (ST5020, Leica, Germany). Stained sections were scanned in their entirety with a NanoZoomer HT 2.0 (Hamamatsu) at a magnification of x20.

## Immunohistofluorescence (IHF)

Sections were deparaffinized, rehydrated and incubated in blocking buffer (5% Bovine Serum Albumine (BSA) in Tris buffered saline-tween® 20, Sigma-Aldrich, USA) during 30 min. Sections were incubated with the primary antibody in 3% BSA overnight before adding the fluorophore coupled-secondary antibody in 3% BSA during 2 h. Immuno-staining for Glut1 (1:200, rabbit monoclonal, #ab115730, Abcam), Glut4 (1:500, rabbit polyclonal, #ab33780, Abcam) and Cd68 (1:400, mouse monoclonal, #MCA341R, Bio-rad) were performed using Opal™ Multiplex immunohistochemistry (IHC) kits (AKOYA Biosciences, Marlborough, MA and Menlo Park, CA, USA) that allow simultaneous IHC detection of the three antigens. Immuno-staining for Cd31 (1:200, goat polyclonal, #AF3628, R&D systems) and cyanine 3 coupled-alpha SMA (mouse monoclonal, #C6198, Sigma-Aldrich) were performed simultaneously. The immune-stained sections were digitalized using Vectra® Polaris™ (AKOYA Biosciences, Marlborough, MA and Menlo Park, CA, USA) with the entire section in a single field of view.

## Staining quantification and analysis

A homemade machine learning based software, Fiber-ML, written in Matlab® (The MathWorks, Natick, MA, USA)[77], was used to quantify IHC and IHF staining in the entirety of each section.

The capillary density in the LV was calculated as the number of Cd31 stained capillaries divided by the number of cardiomyocytes in a given field of the IHF sections using an ImageJ® macro.

## Western blot

LV samples were divided into apex and basal segments and frozen. Two protocols were performed in order to detect and quantify the immunoblots of interest.

Protocol 1: samples lysed in 1X SDS sample buffer and sonicated for 10–15 s. Protein concentrations were determined using the Pierce™ BCA Protein Assay kit (#23225 and #23227, Thermoscientific). A volume of 20 μL of each lysate was loaded onto a SDS-PAGE gel (10 cm × 10 cm, mini-protein TGX gel, #5678024, BioRad), run at 100 volts for 1 h under constant amperage at 30 mA, and the gel was electro-transferred to nitrocellulose membranes (Trans-Blot Turbo Midi 0.2 μm, #1704159, BioRad). The membranes were blocked in skimmed milk and incubated overnight with the following primary antibodies: anti-Gfat1 (1:1000, #D12F4, Cell Signaling Technology) and anti-Gapdh (1:1000, Abcam). Blots were then incubated during 2 h with HRP-linked anti-rabbit IgG (1:5000, goat, #4030-05, SouthernBiotech) and anti-mouse IgG (1:5000, light chain binding protein, #sc-516102, Santa Cruz Biotechnology).

Protocol 2: Ripa buffer (150 mM NaCl, 1 mM EDTA, 1% Triton X-100, 1% Sodium Deoxycholate, 0.1% SDS, 50mMTrisHCl pH7.4,) with protease cocktail inhibitor (cOmplete Mini, Roche) was added to the samples and they were first mechanically homogenized (30 s; Ultra-Turrax IKA) and then stored on ice for 30 min to complete the lysis. Samples were centrifuged (4500 g 5 min 4 °C) and the pellet discarded. Protein concentrations were determined using the DC™ Protein Assay kit (Bio-Rad). 30 μg total proteins of each lysate were loaded onto a SDS-PAGE gel and then electro-transferred to nitrocellulose membrane (Nitrocellulose 0.2 μm, #1620112, BioRad). The membranes were blocked 1 h in Blotto 3% (TBS buffer with 0.05% Tween20 and 3% skimmed milk) and incubated 1 h with the following primary antibodies: anti-Gfat2 (1:1000, #ab190966, Abcam) or O-linked N-acetylglucosamine (1:2000, #RL2, Invitrogen) and anti-Gapdh (1:1000, #ab8245, Abcam). Blots were then incubated 1 h with HRP-linked donkey anti-rabbit IgG (1:5000, Jackson ImmunoReasearch) or HRP-linked donkey anti-mouse IgG (1:5000, Jackson ImmunoReasearch).

Antibody binding was revealed using the chemiluminescent substrate kit (Clarity Western ECL substrate, BioRad) and SuperSignal West Femto Maximum sensitivity substrate (#34095, ThermoFisher).

## Proteomic analysis

Frozen hearts samples were individually ground under liquid nitrogen to yield a fine powder using a pestle and mortar. The tissue powder was quickly weighted and solubilized in 95 °C lysis buffer (4% SDS, 100 mM Tris-HCl, pH 8.0). Protein extracts were clarified by centrifugation at $21,000 \times g$ for 1 h at 4 °C. Protein concentration of the supernatant was estimated and normalized using image intensity integration of the Coomassie blue G250-colored SDS PAGE, loaded with the same volume of each lysate. Peptides were prepared by the Strap technique (ProtiFi, NY, USA) and desalted on $C_{18}$ StageTips[78]. After speed-vacuum drying, peptides were solubilized in 2% trifluoroacetic acid (TFA) and fractionated by strong cationic exchange (SCX) StageTips. Each SCX fraction of each sample was matched with the same fraction and the two adjacent fractions (1 with 1 and 2; 2 with 1, 2 and 3; 3 with 2, 3 and 4…). LC-MS analyses were performed on an U3000 RSLC nano-LC (Dionex) system coupled to a TIMS-TOF Pro mass spectrometer (Bruker Daltonik GmbH, Germany). After drying, peptides from SCX StageTip, the fractions were solubilized in 10 μL of 0.1% TFA containing 2% acetonitrile (ACN). one μL was loaded, concentrated, and washed for 3 min on a $C_{18}$ reverse phase precolumn (3 μm particle size, 100 Å pore size, 75 μm inner diameter, 2 cm length, from Thermo Fisher Scientific). Peptides were separated on an Aurora $C_{18}$ reverse phase resin (1.6 μm particle size, 100 Å pore size, 75 μm inner diameter, 25 cm length mounted to the CSI module, from IonOpticks; Australia) with a 120 min run time and a gradient ranging from 99% of solvent A containing 0.1% formic acid in milliQ-grade $H_2O$, to 40% of solvent B containing 80% acetonitrile, 0.085% formic acid in $mQH_2O$. The mass spectrometer acquired data throughout the elution process and operated in DDA PASEF mode with a 1.89 second/cycle, with Timed Ion Mobility Spectrometry (TIMS) mode enabled and a data-dependent scheme with full MS scans in PASEF mode. This enabled a recurrent loop analysis of a maximum of the 120 most intense nLC-eluting peptides which were CID-fragmented between each full scan every 1.89 s. Ion accumulation and ramp time in the dual TIMS analyzer were set to 166 ms each and the ion mobility range was set from $1/K0 = 0.6$ Vs cm$^{-2}$ to 1.6 Vs cm$^{-2}$. Precursor ions for MS/MS analysis were isolated in positive polarity in the $100-1.700$ $m/z$ range by synchronizing quadrupole switching events with the precursor elution profile from the TIMS device. The cycle duty time was set to 100%, accommodating as many MSMS in the PASEF frame as possible. Singly charged precursor ions were excluded from the TIMS stage by tuning the TIMS using the Otof control software (Bruker Daltonik GmbH). Precursors for MS/MS were picked from an intensity threshold of 1000 arbitrary units (a.u.) and resequenced until reaching a 'target value' of 20,000 a.u, considering a dynamic exclusion gap of 0.40 min. The mass spectrometry data were analyzed using MaxQuant version 1.6.17 (Max Planck institute of Biochemistry, Germany). The database used was a concatenation of *Rattus norvegicus* sequences from the Swissprot database (release 2020-10) and a list of contaminant sequences from MaxQuant. The enzyme specificity was trypsin. The precursor and fragment mass tolerance were set to 20ppm. Carbamido-methylation of cysteins was set as permanent modification and acetylation of protein N-terminus and oxidation of methionines were set as variable modifications. Second peptide search was allowed, and minimal length of peptides was set at 7 amino acids. False discovery rate (FDR) was kept below 1% on both peptides and proteins. Label-free protein quantification (LFQ) was done using both unique and razor peptides. At least 2 such peptides were required for LFQ. The "match between runs" (MBR) option was allowed with a match time 0.7 min window and an alignment time window of 20 min. For differential analysis, LFQ results from MaxQuant were imported into the Perseus software (version 1.6.14).

After excluding reverse and contaminant proteins from analysis, 4257 proteins were identified, among which 3203 proteins had at least one LFQ value across all samples. Before performing statistics, the data was transformed to a logarithmic scale (log2) and further filtered: only 2618 proteins with at least 3 out of 5 valid LFQ values in at least one experimental group were used for comparison between groups. Of these, 971 proteins were identified with a value in each sample (Supplementary Fig. 8a).

Five samples/group post-ISO were performed. In the five 7d post-ISO LV apical group, two samples were outliers. However, in order to maintain homogeneous sample comparison between groups, we did not eliminate these 2 outlier samples since (i) they showed no series effect and (ii) their inclusion did not flaw the statistical confidence of this group's comparisons (Supplementary Fig. 8b).

Significant differential proteins were identified with a two-tailed Student's $t$-test. Only proteins with $p$-value < 0.05 and |fold change|>1.3 were kept for over-representation analysis using Ingenuity Pathway Analysis (QIAGEN Inc, version 60467501). Proteins with missing values in all samples of one group (0/5) and at least 3 values in another group (3/5) were considered as appearance / disappearance and included in the dataset of 2618 proteins (supplementary fig. 8a). Statistical analysis of six group-to-group comparisons, i.e., in the apex: 0 vs. 2 h, 7d vs. 0 and 7d vs. 2 h; in the basal region: 0 vs. 2 h, 7d vs. 0 and 7d vs. 2 h, showed significant differences in 1110 proteins (743 proteins in the apex and 614 proteins in the basal region). Significantly overrepresented terms (canonical pathways, functions, upstream regulators) were identified with a right-tailed Fisher's Exact Test that calculates an overlap p-value. The z-score was also calculated to assess the activation (positive z-score) or repression (negative one) of each term.

## Statistical analysis

For each experimental series, data are presented as means ± standard deviation. For data from longitudinal follow-up of in vivo imaging, each animal is its own control, and values were normalized to the animal's baseline values. Statistical analysis was performed with GraphPad Prism 9.2.0 (GraphPad Software, San Diego, CA, USA). Statistical significance ($p < 0.05$) for each variable was estimated by one-way or two-way ANOVA when group variances were equal (Bartlett test); if not the non-parametric Kruskall–Wallis test, and the Holm multiple comparisons test was used to execute simultaneous $t$-tests.

## Reporting summary

Further information on research design is available in the Nature Portfolio Reporting Summary linked to this article.

# Data availability

Source data are provided with this paper. The mass spectrometry proteomics data that support the findings of this study are available in the ProteomeXchange Consortium via the PRIDE[79] partner repository with the dataset identifier PXD032667. Source data are provided with this paper.

# Code availability

The software developed for this study has been archived in open repositories: Fiber-ML is available at https://gitlab.com/balvayda/fiber-ml. CardioTrack is available at https://gitlab.com/LIB_ICV/cardiotrack.

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

## Acknowledgements

We are grateful to Corinne Lesaffre from the PARCC Histology Core Facility, Florence Marliot from the HEGP Histology Core Facility, Dr Hana Manceau and Nicolas Sorhaindo from the Bichat Biochemistry Core Facility, to Maryline Favier and Rachel Onifarasoaniaina from the Cochin Histology Core Facility, and to Omar Zenteno for Supplementary Fig. 7a. In vivo imaging was performed at the Imaging Facility (Plateforme d'Imageries du Vivant) of the University Paris Cité supported by France Life Imaging (ANR-11INBS-0006), by Infrastructures Biologie-Santé (IBiSa), by Aviesan grants #ASC20001SSP, ASC16025KSA and ASC20031KSA, and by the Région Ile-de-France SESAME funding program. V.N. was funded by H2020 MAESTRIA #965286. We warmly thank Philippe Chafey for his contribution to sample preparation, proteomics data processing, and for his long career of contributions to protein analysis for the scientific community. Many thanks to Tony Lefebvre from University of Lille for his expert advice on O- and N- acetylglycosylation. This work was supported by the DIM Thérapie Génique Paris Ile-de-France Region, by IBiSA, by the Labex GR-Ex, and by a grant from ANR PACIFIC (ANR-18-CE14-0032). T.Y. was supported by the French Ministry of Research and Higher Education, and by the ANR PACIFIC grant. M.P.-L. received funding from the European Union's Horizon 2020 research and innovation Program under the Marie Sklodowska-Curie Grant Agreement no.101030046, and by the Programme Ramón y Cajal RYC2021-032739-I, funded by MCIN/AEI/10.13039/501100011033 and the European Union "NextGenerationEU"/PRTR. Part of the technology developed for this study was supported by SIRIC CARPEM grants to B.T.

## Author contributions

T.Y. carried out all the experimental work and data analysis. T.Y. and B.T. conceived the experiments, discussed the results, and wrote the manuscript. M.P.L. developed cardiac PETRUS technology. D.B. created the Fiber-ML software. M.L.G., F.G., and J.B. conducted and analyzed the proteomics data. A.C. performed the OPAL experiments and histological analysis using Fiber-ML, supervised by D.B. and B.T. P.B. provided expert assistance for the optimization and interpretation of histological results. Y.M. analyzed all the OPAL experiment under the supervision of T.Y. G.A. optimized cardiac MRI sequences and trained T.Y. in their usage. U.G.,

V.N., and N.K. developed the Cardiotrack software. A.L. analyzed the strain measurements results under the supervision of T.Y. and B.T. A.S. performed the electron microscopy. T.G. supervised the SHG imaging and analysis performed by T.Y. F.L. and G.R. performed the cardiac ultrasound imaging, G.R. overseeing the analysis conducted by T.Y. J.V. performed and analyzed western blot with N.M. E.M. provided expert assistance in myocardial MRI and ultrasound imaging. T.V. supervised PET FDG acquisitions made by T.Y. All authors reviewed and approved the final manuscript.

## Competing interests

The authors declare no competing interests.
