## [Peer Review File · Nature Communications]

Acute stress induces long-term metabolic, functional and structural remodeling of the heartREVIEWER COMMENTS

Reviewer #1 (Remarks to the Author):

In this study, Yoganathan and colleagues investigate the mechanism of Takotsobu syndrome using a rodent (rat) model. Given the severity of the acute and long-term effects of TTS on heart function, this is worthy of investigation.

The authors characterize the model according to detection of known biomarkers, ECG measurements and myocardial deformation according to altered strain rates in myocardium. Further, the animal model shared key characteristics such as reversibility and lack of coronary obstruction. Key markers for glucose uptake (Glut1 and Glut4) were analysed by immunohistochemistry, suggesting (subtle) differences in the response in apical and basal regions, in particular the return of Glut1 to baseline levels in the LV basal region, but sustained higher level in the apical region. Is there an explanation for the higher baseline level of Glut1 in the basal region?

The authors then used a proteomic approach to further investigate difference in protein abundance. Generally the methods are well reported and the bioinformatic analysis is robust. However there are a few concerns about the approach used to filter significant proteins and there is room for improvement regarding the methods used to represent the data.

What was the total number of proteins identified and how many were significant according to the selected criteria? While it is common in large-scale datasets with many variables to adjust for multiple hypotheses testing, the authors use raw p-values as the threshold to determine significant proteins. Were the reported p-values significant after applying a false discovery rate?

It would be helpful to represent the data in full to visualise the relationships between the samples, for example by PCA plot. Regarding the pairwise comparison it would be helpful to see the context of key proteins (ie those reported in the table) highlighted, for example on a volcano plot. In addition to the table in Fig 4, it would also be helpful to see sample-level data (ie how consistent the LFQ values were across the full set of samples) for example using a heat map. Are the n values reported in Fig 4B applicable to the whole proteomic dataset, ie 4, 5 and 5 animals for 0h, 2hr and 7d time points?

Regarding the protein identification methods, the authors use SCX fractionation, presumably to increase the number of peptides identified. How many fractions per sample were analyzed? Given that SCX is not particularly precise (ie the same peptide may elute in multiple fractions, or different fractions from one sample to another), how were such peptide features matched between LC/MS runs in MaxQuant?

Did the use of concatenated database comprising UniProt and SwissProt sequences increase the number of redundant entries? Did this affect the 'grouping' of proteins (ie were UniProt and SwissProt entries reported as separate protein groups due to sequence variations?)

The authors uploaded their raw data to PRIDE, however, I was unable to review the search results (ie ProteinGroups.txt file) as reviewer login credentials were not provided.

Reviewer #2 (Remarks to the Author):

The authors should be commended for implementing a murine model of TTS and analyzing anatomical and functional myocardial changes in such a multiparametric and comprehensive way. However, I would like to draw the authors' attention to the following points:

1. In a recent study, Godsman et al. already described a dysregulation of glucose and lipids metabolic pathways, as well as inflammation and upregulation of remodeling pathways and fibrosis in a rat model of TTS (Godsman N et al. *Cardiovasc Res.* 2022 doi: 10.1093/cvr/cvab081). A close comparison with this study highlighting analogies and new elements is recommended. Furthermore, a

long-term heart failure phenotype due to impaired cardiac deformation indices, increase of T1 mapping values and anomalies of cardiac energetic status have been previously described in humans with TTS, challenging the concept of reversibility (Sclay C et al. *Circulation*. 2018 doi: 10.1161/CIRCULATIONAHA.117.031841; Schwarz K et al. *J Am Soc Echocardiogr*. 2017 doi: 10.1016/j.echo.2017.03.016). Please emphasize the new aspects emerging from your current study.

2. Left ventricular ballooning has been often described in previous TTS rat models (Shao Yet al. *Int J Cardiol*. 2013 doi: 10.1016/j.ijcard.2012.12.092; Ali A et al. *ESC Heart Fail*. 2021 doi: 10.1002/ehf2.13530). In the current model mainly LV hypercontractility and decrease of LV apical strain was reported, but not hypo- or akinesia of LV myocardial segments. How would you explain these differences? Might this have implications on short- and long-term metabolic and structural changes?

3. The reduction of left atrial reservoir function is in my opinion one of the most interesting findings of the study. It suggests that TTS could lead to diastolic dysfunction with important prognostic implications at long-term. Please expand this topic since it is one of the strengths of the study.

4. In the Introduction the authors states: "The results combat the notion of irreversibility of stress-induced cardiomyopathy, point to dysregulation of metabolic pathways as the main cause of delayed consequences and support early therapeutic management of Takotsubo". Considering the study results, which kind of treatment could prevent metabolic dysregulation and myocardial fibrosis? You report that overactivation of Hexosamine Biosynthetic pathway may lead to fibrosis and vascular proliferation. Could this be a therapeutical target?

Reviewer #3 (Remarks to the Author):

I read with interest the work by Yoganathan and colleagues on the aspects of adverse cardiac remodeling following acute stress in animal model, with the overarching aim to mimic the mechanisms of Takotsubo cardiomyopathy in human. The authors studied the metabolic, functional and structural changes in rodents at baseline and at 2 hours, 7 days, 1-month and 3-month intervals following a single administration of isoproterenol (ISO), which leads to catecholamine surge.

The study is ambitious. It comprehensively touched upon multiple aspects of physiological and structural changes of the rodent heart using multi-modality imaging techniques, namely FDG positron emission tomography, high resolution echocardiography and cardiac magnetic resonance imaging, as well as electrocardiogram, clinical parameters such as blood pressure measurement, serum biochemistry, histology and proteomic analysis.

The study was systematically carried out with the results clearly presented. The findings were intriguing and challenges our current understanding of Takotsubo cardiomyopathy being a reversible and rather benign entity. Most acute stress changes observed at the 2nd hour had recovered at the 7th day. However, there was high FDG uptake, reduced glucose breakdown and alternative pathways of glucose utilisation in the apex. This is coupled with tissue and vascular remodeling at the apex. In addition, fibrosis was already present in the apex at the 7th day post-ISO and persisted at 3-month, suggesting permanent structural damages to the heart following acute stress.

I have the following questions and some suggestions:

1. The study protocol was well designed and provided new insight into the pathophysiology of TTS. Basically, I agree with the conclusion. Nevertheless, methods of assessment of cardiac function must be addressed.

a. Please describe the conditions in CMR imaging of the RV, RA, and LA.

b. Number of frames of LV was 16 per heart cycle in CMR. When the left and right atria are imaged under the same conditions, the number of frames is insufficient to calculate early and late diastolic strain rates. In fact, the LA late diastolic strain rate (pump function) in Figure 1C increased nearly 50-fold over the control during the chronic period, and the late diastolic strain rate of the RA was negative relative to the control. It is generally not possible for atrial function to increase 50-fold or become negative (reverse pumping function?) relative to controls, either in humans or in rats. All cited papers 68-70 are studies on humans and not on rats having heart rate greater than 300 beats/min.

c. Line 105-107 “Furthermore, end-diastolic atrial contraction wave (A) was predominant over the early LV filling wave (E), resulting in a reduced E/A ratio, revealing an abnormal LV relaxation and a compensatory left atrial (LA) contraction.” When rats with high resting heart rates further accelerate their heart rate during acute stress, it is difficult to separate the E and A waves in the mitral inflow. In this case, the apparent high A wave may reflect the high E wave (“Am J Physiol Heart Circ Physiol 283: H346–H352, 2002: “Because the heart rate was greater than 300 beats/min in most rats, it was difficult to clearly identify the A wave on transmitral flow velocity spectra. For this reason, measurement of the peak velocity of the A wave and DT was much less feasible than E wave measurement in all three groups”).

d. In addition, in figure 1C, there was no evidence of compensatory LA contraction (LA booster pump function), while it is possible that the strain rate measurements were incorrect.

e. During the chronic phase of the disease, atrial volume is an excellent indicator of atrial load. Please display left and right atrial volume data as atrial volume can be measured by CMR.

f. “Eur Heart J Cardiovasc Imaging 2020;21:1184-1207”. Multimodality imaging in Takotsubo Syndrome: a joint consensus document of the European Association of Cardiovascular Imaging (EACVI) and the Japanese Society of Echocardiography (JSE) “RV free wall longitudinal strain is therefore recommended to detect even small areas of myocardial dysfunction hardly recognizable by visual assessment and/or by other traditional echocardiographic parameters. Conversely, RV global longitudinal strain should not be assessed because involvement of the interventricular septum can be misleading.” In this study (figure 1A), the RV longitudinal strain was decreased in the acute phase and the LV longitudinal strain was increased. Since both ventricles share the ventricular septum, these results suggest a reduction in RV free wall strain. “Recent studies have further demonstrated RV involvement in TTS as an important finding associated with worse outcome.” (Eur Heart J Cardiovasc Imaging 2020;21:1184-1207). The finding that LA remodeling occurs in the chronic phase of TTS with RV damage in the acute phase has important implications for the pathophysiology of TTS.

g. Based on a human study, “TTS patients demonstrated a significantly decreased LA function during the acute/subacute phase of the disease. However, impairment of LA performance seems to be transient in TTS with recovery during follow-up.” (Journal of Cardiovascular Magnetic Resonance (2017) 19:15). As observed in the current study, LA function in the chronic phase was reduced and complicated by LA fibrosis, suggesting that this is an animal model which recapitulates a slightly more severe form of TTS.

h. Although the term GLS (global longitudinal strain) was used in this study, it is preferable to use the simple term “longitudinal strain”, since this study did not use the average value of longitudinal strain in multiple cross sections (2-, 3-, and 4-chamber view). If the average of longitudinal strain of multiple cross sections was used, please state how many cross-sectional averages were used for the LV, RV, LA, and RA.

i. Line 149 “Parameters derived from cardiac ultrasound images were normal at 7d post-ISO (figure 6; extended data 2 and 4).” There was no data of ultrasound images in figure 6.

2. The plasma concentrations of AST, ALT and CK were measured. These are non-specific and could be raised in various conditions, which are non-cardiac. If the aim was to measure myocardial damage, would the authors explain why cardiac-specific markers, e.g., CK-MB, troponins (T or I) were not measured?

3. Overall, in this study, a rat model of stress-induced cardiomyopathy is used to observe acute and long-term myocardial damage via high-end imaging and microscopic imaging. The article is well written and highly interesting to read. However, despite a huge amount of work, the article eventually offers as only claim to provide early treatment to patients with Takotsubo, which is intuitive. Please expand to what other of the current patient management the authors may refer...

4. The whole study is rationalized by the fact that long-term cardiovascular annual death rates for Takotsubo Syndrome and acute coronary syndrome are the same. Supportive references are: 1). a review article with expert panel from Lyon et al. that did not report survival rates from Takotsubo syndrome; and 2). an observational study that reported a large heterogeneity in long-term outcomes from Takotsubo syndrome, depending on the identified cause. Whilst those caused by physical activities or neurological disorders have poorer or equal outcomes than patients with acute coronary syndrome, those with emotional trigger (by far the most!) had a much better outcomes than the patients with acute coronary syndrome. This points to a significant heterogeneity of all sub-syndromes included in the umbrella of Takotsubo, and probably of the subsequent long-term damage. As such, it would be reductive to consider that a single model would help decipher the whole spectrum. It is thus important to refocus the work and the writing on the subtype which is addressed by the proposed model.

5. The claim that overactivation of the hexosamine and polyol biosynthetic pathways explain tissue and vascular remodeling and induction of reactive oxygen species is proven by the current work. However, in my opinion, at least another hypothesis has been neglected based on the study findings: the cell-cell junction alteration (that is presumably one of the results of hyperdynamism). In a different, but close disease model, extreme athletes may end having acute myocardial damage and even long-term fibrosis via hyper dynamism without any alteration of the glucose metabolism. In general, the hypothesis-generating part of this work should be enriched to include other potential pathways for myocardial damage.

6. The demonstration of the absence of coronary obstruction part seems superfluous and could be moved to the additional data.

7. Page 2, line 37. Could the authors mean 'reversibility', rather than 'irreversibility'?

8. Please only capitalise proper nouns. As such, 'Humans', 'Ultrasound' etc. should be in small letters.

Reviewer #4 (Remarks to the Author):

The overall goal of this study was to examine the effects of an acute pharmacological stress on the heart designed to mimic Takotsubo syndrome and to determine the contributions of acute and chronic metabolic remodeling to the long term damage to the myocardium.

There are however some serious limitations their evaluation and interpretation of the data.

1) In the text the authors state that 2h post-ISO PET imaging showed a 50% increase in the uptake of FDG into the heart (lines 169-170). In Figure 2A the PET images demonstrate marked increase in FDG uptake at both 2 hours and 7days; however, this is not reflected in any of the quantified data presented in Fig 2B.

2) While the data presented in Fig 2B do not appear to show any substantial changes at 2h post-ISO, it is stated that FDG uptake was increased by 70% whereas the index of glucose phosphorylation was reduced. They refer to the apparent disconnect between changes in glucose uptake and phosphorylation as "metabolic stunning". They suggest that this low capacity to use glucose as an energy source is contributes to cardiac dysfunction; however, this conclusion is flawed. It is widely recognized that glucose is not the predominant fuel for mitochondrial ATP production, rather fatty acids are the largest source with contributions from lactate, ketone bodies amino acids (For a recent review see *Circulation Research*. 2021; 128:1487–1513). Thus, in the absence of any other measures of substrate utilization it is not possible to conclude that potential changes in glucose metabolism have any meaningful effects. In addition, in Extended Data 3C, the circulating plasma concentrations of lactate and fatty acids increase at least 2-fold 2h post-ISO both of which can be readily oxidized by the heart and likely contribute to the apparent decrease in glucose metabolism. Indeed, high

circulating lactate levels have been associated with reduced cardiac glucose uptake (e.g., J Physiol. 2002;542: 403-12).

3) Lines 180-181: "During the recovery stage (7d post -ISO), GLUT1 staining was high in the apex but returned to baseline in the basal region, while an increase of GLUT4 staining was observed in both regions". However, in Fig 3 there are no changes in GLUT4 levels in the basal region at any time point. Moreover, measurements of total GLUT1 and GLUT4 protein levels provide no information regarding their contributions to glucose uptake, as only membrane levels are of relevance to glucose metabolism.

4) Lines 193-194 "A high myocardial uptake of glucose increases collagen synthesis, myocardial myofibroblast proliferation, and the expression of fibronectin and transforming growth factor (TGF)- β 1". This is a misinterpretation of the studies cited, which show that high extracellular glucose levels stimulate fibrotic changes in cultured cardiac myoblasts. I am unaware of any evidence that increased myocardial uptake of glucose induces fibrotic like changes.

5) Understanding the proteomic data presented in Fig. 4 is complicated by the fact that the legend states that "green, overexpression higher than 1.3-fold, red: under-expression lower than -1.3-fold". However, in the figure the green numbers are all preceded by a negative sign and the red numbers have no sign. In the text, which is mostly focused on the HBP and Polyol pathways it is stated that proteins in these pathways are overexpressed, so despite the legend I have assumed that the red colors indicate overexpressed proteins. The conclusion for this specific section is "notion that HBP hyperactivation induced by an acute stress is a driver of cardiac fibrosis and angiogenesis" which as stated in the Discussion this is due to increased O-GlcNAcylation. However, it is not possible to reach this conclusion based on the data presented. It is true that key elements of the HBP and O-GlcNAc regulation are increased at 7 days post-ISO. While the authors have highlighted changes in GFAT expression they have overlooked the 8-fold increase in O-GlcNAcase (OGA) expression, which removes O-GlcNAc from proteins. Therefore, assuming that protein expression equates to activity, then OGA activity is increased more 5-times that of OGT, which if true would result in reduced O-GlcNAc levels not an increase. It is not possible to draw any conclusions regarding changes in O-GlcNAcylation, without a measurement of O-GlcNAc levels at 0 and 7d post-ISO. Immunoblots of some of these key proteins to support the changes observed in the proteomic studies would also be helpful.

6) Lines 304-306: "Finally, overactivation of O-GlcNAcylation increases synthesis of extracellular matrix proteins, as observed here at 7d post-ISO in the apex." Apart from the fact that it cannot be concluded that there are any changes in O-GlcNAc levels, the references cited to support this statement indicate that increased HBP flux in mesangial cells is linked to increased synthesis of extracellular matrix proteins. Neither study includes measurements of O-GlcNAc levels.

Important changes have been made to address the comments of the four reviewers. In particular,
we have performed a hierarchical clustering analysis of the proteins involved in glucose pathways
(Figure 4A), an analysis of GFAT2 and of O-GlcNAcylated proteins (Figure 4B), and added an
analysis of the canonical metabolic, vascular and tissue pathways (Extended data 8). We have also
followed Reviewers' suggestion to remove the graph representing the E/A ratio from the
ultrasound analysis in extended data Figure 4, as well as those of the strain rate measurements
from figure 1.

The point-by-point answers to reviewers' comments are color-coded:

- - Comments/Answers to reviewer #1 and corresponding changes in the revised manuscript are
highlighted in yellow.
 - - Comments/Answers to reviewer #2 and corresponding changes in the revised manuscript are
highlighted in green.
 - - Comments/Answers to reviewer #3 and corresponding changes in the revised manuscript are
highlighted in gray.
 - - Comments/Answers to reviewer #4 and corresponding changes in the revised manuscript are
highlighted in blue.

**1. Reviewer #1 (Remarks to the Author):**

In this study, Yoganathan and colleagues investigate the mechanism of Takotsubo syndrome using a
rodent (rat) model. Given the severity of the acute and long-term effects of TTS on heart function, this
is worthy of investigation.

**The authors characterize the model according to detection of known biomarkers, ECG**
**measurements and myocardial deformation according to altered strain rates in myocardium.**
**Further, the animal model shared key characteristics such as reversibility and lack of coronary**
**obstruction. Key markers for glucose uptake (Glut1 and Glut4) were analysed by**
**immunohistochemistry, suggesting (subtle) differences in the response in apical and basal regions,**
**in particular the return of Glut1 to baseline levels in the LV basal region, but sustained higher**
**level in the apical region. Is there an explanation for the higher baseline level of Glut1 in the basal**
**region?**

In fact, the baseline level of GLUT1 expression is slightly *lower* in the basal than in the apical region,
although the difference is not statistically significant. The use of the same scales for the graphs of panels

G-H-I (apical region) and J-K-L (basal region) in Figure 3 may have been misleading. Therefore, we
have now changed the scales of panels J-K-L in the revised **Figure 3 page 25**. Please note that, as shown
in the accompanying graph hereunder, GLUT1 in the apical region is significantly higher than in the
basal region, both during the acute (2h post-ISO) and late phases (7d post-ISO), indicating that the apex
is the major region of GLUT1 changes.

The authors then used a proteomic approach to further investigate difference in protein abundance. Generally, the methods are well reported and the bioinformatic analysis is robust. However, there are a few concerns about the approach used to filter significant proteins and there is room for improvement regarding the methods used to represent the data.

What was the total number of proteins identified and how many were significant according to the selected criteria?

We identified 4257 proteins among which 3203 were quantified revealing a statistically significant differences between 1110 proteins in the 6 comparison groups (in the apex: 0 vs. 2h, 7d vs. 0 and 7d vs. 2h; in the basal region: 0 vs. 2h, 7d vs. 0 and 7d vs. 2h). This information has now been added to the revised manuscript (**page 22, lines 570-572**).

While it is common in large-scale datasets with many variables to adjust for multiple hypotheses testing, the authors use raw p-values as the threshold to determine significant proteins. Were the reported p-values significant after applying a false discovery rate?

Unlike RNA sequencing, in proteomics, as for micro-arrays, we generally do not consider adjusted p-values (q-values) as we often have few samples (usually 3 samples per group and in our case 5 samples). Moreover, on/off (appearances/disappearances) proteins are excluded from the statistical test as we perform statistics on log₂ LFQ intensities. In our case, as shown in the table below, only a small number of proteins were found significantly different using their q-values, while many proteins passed the cut

off of the p-values and the on/off criteria. So we considered as proteins of interest (POI), on/off proteins
 and proteins with p-value less than 0.05 and fold change greater than 1.3 in absolute value.

	POI	POI p<0,05 & FC >1,3	POI q<0,05 & FC >1,3	on/off
C_a vs B_a	313	235	1	78
C_a vs A_a	508	395	15	113
B_a vs A_a	202	170	0	32
C_b vs B_b	170	125	0	45
C_b vs A_b	421	354	4	67
B_b vs A_b	217	166	0	51

 **It would be helpful to represent the data in full to visualize the relationships between the samples,**
 **for example by PCA plot.**

 Please find below the PCA representation (Figures **PCA 1** and **PCA2**) of proteins with 100% valid
 values, i.e., proteins that have LFQ intensity values for all samples in the apical or basal regions. Figure
 **PCA1** shows that the proteins in the 7d post-ISO group are well distinguished from those of the other
 groups (control and 2h post-ISO) in the apex along PCA component #2, suggesting significant and
 distinct late-phase remodeling in the apex (Figure **PCA1**). In contrast, distinction between these three
 time points is less straightforward in the basal region (Figure **PCA2**), following either PCA
 component#1 or component#2, indicating that metabolic, tissue, and vascular remodeling occurred
 primarily in the apex at 7d post-ISO.

**PCA 1:** Principal component analysis of proteins with 100% of valid values (proteins that have LFQ
 intensity value for all conditions) in the LV apex. According to component#2, proteins of the group
 7d post-ISO are distinct from proteins of groups baseline (0) and 2h post-ISO, suggesting a strong
 proteomic remodeling in the apex during the late phase of TTC.

Regarding the pairwise comparison it would be helpful to see the context of key proteins (ie those reported in the table) highlighted, for example on a volcano plot. In addition to the table in Fig 4, it would also be helpful to see sample-level data (ie how consistent the LFQ values were across the full set of samples) for example using a heat map.

Using adjusted p-values, a volcano plot will not integrate the appearances/disappearances of proteins. Therefore, given the low number of proteins with significant q-values in our study, the volcano plot cannot be represented. Please find below the heatmaps with hierarchical clustering of all proteins in the apical regions (Figure **HM_apex-all**) as well as those involved in glucose metabolic pathways (Figure **HM_apex-GM**) in the apical region.

Figure **HM_apex-all**. Hierarchical clustering of proteins of interest in the apical region showing in particular clustering at 7d post ISO. Based on Pearson Correlation of z-scored quantification values.

**HM_apex-GM.** Hierarchical clustering of proteins of interest involved in the glucose
 metabolism pathways in the cardiac apex, based on Pearson Correlation of z-scored
 quantification values.

We thank the reviewer for his suggestion and added the above Figure **HM_apex-GM** to **Figure 4** in the
 text **page 26** in order to improve readability of the proteomic data.

**Are the n values reported in Fig 4B applicable to the whole proteomic dataset, ie 4, 5 and 5 animals**
 **for 0h, 2hr and 7d time points?**

The samples used for the proteomic analysis and those used for the western blot are from the same
 hearts. One of the control samples of the western blot is missing due to a blotting issue (bubble).

Note however that the new western blots data included in the revised manuscript to answer Reviewer
4's comments come from a different series of animals performed after the initial submission. These
experiments were performed within the same conditions, i.e., same strain, sex, age of animals, same
protocol yielding same phenotype after ISO stress (**Figure 4B, page 20, lines 508-519**).

**Regarding the protein identification methods, the authors use SCX fractionation, presumably to**
**increase the number of peptides identified. How many fractions per sample were analyzed? Given**
**that SCX is not particularly precise (ie the same peptide may elute in multiple fractions, or**
**different fractions from one sample to another), how were such peptide features matched between**
**LC/MS runs in MaxQuant?**

Each SCX fraction of each sample was matched with the same fraction and the two adjacent fractions
(1 with 1 and 2; 2 with 1, 2 and 3; 3 with 2, 3 and 4...). This is now expressively mentioned in the revised
manuscript (**page 20, lines 532-534**).

**Did the use of concatenated database comprising UniProt and SwissProt sequences increase the**
**number of redundant entries? Did this affect the 'grouping' of proteins (ie were UniProt and**
**SwissProt entries reported as separate protein groups due to sequence variations?)**

We are obliged to the reviewer for pointing to an error in our initial manuscript. In fact, only SwissProt
was interrogated. We have corrected this point in the Methods section of the article (**page 21, line 557**).

**The authors uploaded their raw data to PRIDE, however, I was unable to review the search results**
**(ie ProteinGroups.txt file) as reviewer login credentials were not provided.**

We apologize for this broken link. Please find here the Reviewer login credentials:

**Project Name:** Acute stress induces long-term metabolic, functional and structural remodeling of the
heart

**Project accession:** PXD032667

**Username:** reviewer_pxd032667@ebi.ac.uk

**Password:** YrkkTwzx

**2. Reviewer #2 (Remarks to the Author):**

The authors should be commended for implementing a murine model of TTS and analyzing anatomical
and functional myocardial changes in such a multiparametric and comprehensive way. However, I
would like to draw the authors' attention to the following points:

**1. In a recent study, Godsman et al. already described a dysregulation of glucose and lipids**

**metabolic pathways, as well as inflammation and upregulation of remodeling pathways and**
**fibrosis in a rat model of TTS (Godsman N et al. *Cardiovasc Res.* 2022 doi: 10.1093/cvr/cvab081).**
**A close comparison with this study highlighting analogies and new elements is recommended.**

We thank the reviewer for pointing out to the study conducted by Godsman *et al.*, as their results are
highly complementary to ours and strengthen our findings. We have referenced their study in the
discussion section of the revised manuscript (**page 11, lines 274 - 280**). However, some important
differences between their study and ours should be highlighted.

Firstly, Godsman et al studied animals at 3 days and 7 days post ISO while we looked at the acute phase
(2 hours post ISO), and late phases (1- and 3-months post ISO, corresponding to several years of “human
life). Secondly, they used an ISO dose twice that of our study, which we avoided because of a significant
mortality rate. Thirdly, they attribute higher FDG uptake to an activation of macrophages, while our
histochemical data shown in Figure 3 clearly demonstrates that the increase in CD68+ stained
macrophages is limited and, more importantly, that GLUT 1 and GLUT4 are essentially expressed in
cardiomyocytes. Of note, we performed GLUT1, GLUT4 and CD68 staining on the same sections, and
show a complete absence of colocalization between the GLUTs and CD68. Fourthly, we showed that
the diversion of glucose metabolism towards alternative anabolic pathways responsible for tissue and
vascular remodeling, mainly the HBP and polyol pathways. Fifthly, we showed pathological
angiogenesis and remodeling of the cytoskeleton. Finally, our study clearly demonstrates the
regionalization of the metabolic, tissue and vascular and functional abnormalities, with the important
finding of a late (1 and 3 month) extension of anomalies such as glucose metabolism, diffuse fibrosis,
and the left atrial dysfunction. In addition, our study demonstrates the capacity of noninvasive *in vivo*
follow-up after TTC to highlight TTC abnormalities throughout the evolution of the disease.

**Furthermore, a long-term heart failure phenotype due to impaired cardiac deformation indices,**
**increase of T1 mapping values and anomalies of cardiac energetic status have been previously**
**described in humans with TTS, challenging the concept of reversibility (Scally C et al. *Circulation.***
**2018 doi: 10.1161/CIRCULATIONAHA.117.031841; Schwarz K et al. *J Am Soc Echocardiogr.***
**2017 doi: 10.1016/j.echo.2017.03.016). Please emphasize the new aspects emerging from your**
**current study.**

We totally agree with the reviewer that the notion of reversibility in TTC was questioned by clinical
studies, including some cited by the reviewer that have supported long term heart failure phenotypes
based on *in vivo* cardiac imaging^{1,2}. In fact, our study was inspired by clinical data suggesting an
unexplained irreversibility of TTC. The major advantages of using an animal model are to strengthen
hypotheses and explore in much deeper detail the underlying mechanisms. In particular, for obvious
reasons, clinical studies do not compare pre-stress and post-stress conditions in the same individuals,
and rely on matched “control” subjects, and neither the TTC or the control subjects are exempt of co-
morbidities in addition to cardiac and other treatments. Thus, even though our study is reductionist and

calls for confirmation in human patients, it stands on an objective cause-effect paradigm. Regarding
Schwarz *et al.*², their interesting observations did not report on the metabolic defects that our study
designates as a prominent cause of long-term cardiac disability. Scally *et al.*¹ showed changes in cardiac
³¹P-spectroscopy, typical of impaired cardiac energy metabolism associated with increased native
myocardial T1 without change in extracellular volume. They did not investigate further the metabolic
pathways that were involved. Their study did not report signs of fibrosis, while Schwarz *et al.*² suspected
fibrosis based on increased cardiac extracellular volume, which is an indirect measurement in contrast
to the direct and quantitative tissue analysis that we report here.

In short, our original findings concern the mechanisms leading to long term post-stress cardiac
dysfunction. We show changes in glucose uptake and metabolism preceding a whole cascade of tissue
and vascular events, durably weakening the TTC heart. This remodeling corresponds first to a form of
glucose intolerance in the heart followed by inhibition of the energy metabolic pathway, glycolysis, and
overactivation of the non-energy anabolic pathways of glucose, namely the hexosamine biosynthetic
and polyol pathways. The latter two pathways are known to be responsible for myocardial abnormalities
and dysfunction^{3,4}. Therefore, our extensive exploration of a Takotsubo-like model demonstrates a close
connection between the pathological overactivation of alternative glucose pathways and structural and
functional long-term changes of the myocardium.

**2. Left ventricular ballooning has been often described in previous TTS rat models (Shao Yet al.**
**Int J Cardiol. 2013 doi: 10.1016/j.ijcard.2012.12.092; Ali A et al. ESC Heart Fail. 2021 doi:**
**10.1002/ehf2.13530). In the current model mainly LV hypercontractility and decrease of LV apical**
**strain was reported, but not hypo- or akinesia of LV myocardial segments. How would you explain**
**these differences?**

Because isoprenaline increases heart rate to 450 bpm (baseline 300 bpm) and respiratory rates were
around 60 cycles per min, conventional echocardiography and cardiac MRI imaging were limited by
temporal resolution and unable to directly observe the ballooning of the LV apex. This may have been
possible with superfast CT or MRI, which we didn't have access to. In some cases, we observed transient
apical ballooning using ultrafast ultrasound, however this was incidental, restricted to two-dimensional
observations and largely operator dependent. For this reason, we relied on MRI-based myocardial strain
measurements, that showed a basal hyperkinesia and an apical akinesia at 2h post-ISO, which
corresponds to the apical ballooning observed in patients with Takotsubo^{5,6} (**page 5, lines 119-124**).

**Might this have implications on short- and long-term metabolic and structural changes?**

Indeed, the transient basal hyperkinesia and irreversible apical akinesia coincided with the significant
presence of fibrosis, first in the apex and then in the whole LV, but also with the other tissue, vascular,
and metabolic remodeling observed at 7d, 1- and 3-months post-ISO.

3. The reduction of left atrial reservoir function is in my opinion one of the most interesting findings of
the study. It suggests that TTS could lead to diastolic dysfunction with important prognostic implications
at long-term. **Please expand this topic since it is one of the strengths of the study.**

We fully agree with the reviewer and have expanded on this important aspect of our study in the
Discussion section of the revised manuscript (**page 12, lines 312-317**).

4. In the Introduction the authors states: “The results combat the notion of irreversibility of
stress-induced cardiomyopathy, point to dysregulation of metabolic pathways as the main cause of
delayed consequences and support early therapeutic management of Takotsubo”. **Considering the
study results, which kind of treatment could prevent metabolic dysregulation and myocardial
fibrosis? You report that overactivation of Hexosamine Biosynthetic pathway may lead to fibrosis
and vascular proliferation. Could this be a therapeutical target?**

Thank you for this most pertinent question, which is presently the center of our investigations. Several
avenues of treatment may be considered based of the observations made in our study. First, as you
suggest, it would be interesting to counter the overactivation of the hexosamine biosynthetic pathway,
for example by blocking the activity of its limiting enzyme, GFAT. The compound 6-diazo-5-oxo-L-
norleucine (DON) is a synthetic analog of glutamine that inhibits GFAT and has already been used
experimentally. Studies have shown that muscle cells chronically exposed to high levels of glucose
develop insulin resistance which may be prevented when incubated with DON⁹. DON inhibition of the
hexosamine biosynthetic pathway results in a decrease in the expression of genes related to
cardiomyocyte hypertrophy induced by 48 h of incubation with phenylephrine, protein synthesis and
cardiomyocyte growth, thus preventing cardiac hypertrophy³.

Secondly, reducing the activity of the polyol pathway is also an interesting avenue to explore as it may
prevent cellular glucose from being diverted to non-oxidative pathways and redirected to glycolysis¹⁰.

Several authors have proposed the inhibition of AR by sorbinil or ranirestat^{11,12}. Third, antifibrotic
therapy may perhaps slow the progression of fibrosis in the LV and left atrium, although there is no
clear evidence as far as we know. The connective tissue growth factor CCN2 is a matricellular protein
that alters cell signaling pathways responsible for myofibroblast activation leading to fibrosis¹³. CCN2
antagonists may represent a novel strategy to limit and reverse cardiac fibrosis. In a preclinical study,
mice with pressure-overload-induced heart failure treated with anti-CCN2 monoclonal antibodies (such
as FG-3019) showed significant improvement in LV function compared with controls¹⁴. Another
candidate, Gal-3 is a soluble beta-galactoside-binding lectin that is considered a promising therapeutic
target in heart failure because it is involved in the proliferation of cardiac fibroblasts that cause fibrosis¹⁵.

Numerous studies have shown that Gal-3 expression is elevated in hypertrophied hearts¹⁶⁻¹⁸ in interstitial
lung disease¹⁹ and in the plasma of heart failure patients²⁰. One study showed that LV hypertrophy was
prevented in Gal-3 knockout mice and LV function was improved²¹. Also, among atypical potential ani-

fibrosing agents, some microRNAs (miRNAs) expressed by cardiac myocytes regulate signaling
pathways in fibroblasts. Silencing of miRNAs has been shown to prevent interstitial fibrosis and inhibit
cardiac function in mouse hearts under pressure overload²². Pirfenidone (PFD) is an antifibrotic drug
that reduces the synthesis and release of the pro-fibrotic cytokine TGF β (Tissue Growth Factor- β)²³,
thereby reducing collagen deposition in the lungs and kidneys²⁴. A preclinical study found that
pirfenidone slowed the progression to LV dysfunction, and limited ventricular arrhythmia²⁵. Similar
effects have been reported for cardiac stiffness²⁶, left ventricular hypertrophy²⁷, and cardiac fibrosis²⁸ in
animal models subjected to pressure overload. Relaxin is a vasodilator hormone that reduces collagen
synthesis in the extracellular matrix through various processes, including inhibition of profibrotic factors
such as TGF β ²⁹. Numerous preclinical studies have shown that it reduces myocardial fibrosis in
myocardial infarction³⁰, diabetes³¹ and hypertension³². Relaxin also reduces the incidence of atrial
fibrillation by limiting collagen deposition and reversing atrial fibrosis in rat hearts³³.
Thus, several signaling pathways can be targeted to mitigate the deleterious effects of metabolic and
tissue remodeling. This could confirm the direct link between the over activation of glucose anabolic
pathways and angiogenesis and the presence of fibrosis in the myocardium. Because of the complexity
of Takotsubo cardiomyopathy, which encompasses a broad spectrum of triggers and pathophysiological
changes following these triggers, we would like to insist on the importance of noninvasive in vivo
imaging to detect the characteristic imaging biomarkers of Takotsubo.

**3. Reviewer #3 (Remarks to the Author):**

I read with interest the work by Yoganathan and colleagues on the aspects of adverse cardiac remodeling
following acute stress in animal model, with the overarching aim to mimic the mechanisms of Takotsubo
cardiomyopathy in human. The authors studied the metabolic, functional and structural changes in
rodents at baseline and at 2 hours, 7 days, 1-month and 3-month intervals following a single
administration of isoproterenol (ISO), which leads to catecholamine surge.

The study is ambitious. It comprehensively touched upon multiple aspects of physiological and
structural changes of the rodent heart using multi-modality imaging techniques, namely FDG positron
emission tomography, high resolution echocardiography and cardiac magnetic resonance imaging, as
well as electrocardiogram, clinical parameters such as blood pressure measurement, serum
biochemistry, histology and proteomic analysis.

The study was systematically carried out with the results clearly presented. The findings were intriguing
and challenges our current understanding of Takotsubo cardiomyopathy being a reversible and rather
benign entity. Most acute stress changes observed at the 2nd hour had recovered at the 7th day. However,
there was high FDG uptake, reduced glucose breakdown and alternative pathways of glucose utilisation
in the apex. This is coupled with tissue and vascular remodeling at the apex. In addition, fibrosis was

already present in the apex at the 7th day post-ISO and persisted at 3-month, suggesting permanent
structural damages to the heart following acute stress.

I have the following questions and some suggestions:

**1. The study protocol was well designed and provided new insight into the pathophysiology of**
**TTS. Basically, I agree with the conclusion. Nevertheless, methods of assessment of cardiac**
**function must be addressed.**

**a. Please describe the conditions in CMR imaging of the RV, RA, and LA.**

MRI was performed under general anesthesia while keeping the heart rate, respiratory rate, animal
temperature and isoflurane flow constant during the exam. Acquisitions protocol comprised successive
T1-weighted cine sequences including short axis stack of 6 to 7 slices covering the left and right
ventricles from base to apex, along with conventional 2-chamber and a 4-chamber views (see in the
Methods section of the revised manuscript (page 16, lines 418-421).

**b. Number of frames of LV was 16 per heart cycle in CMR. When the left and right atria are**
**imaged under the same conditions, the number of frames is insufficient to calculate early and late**
**diastolic strain rates. In fact, the LA late diastolic strain rate (pump function) in Figure 1C**
**increased nearly 50-fold over the control during the chronic period, and the late diastolic strain**
**rate of the RA was negative relative to the control. It is generally not possible for atrial function**
**to increase 50-fold or become negative (reverse pumping function?) relative to controls, either in**
**humans or in rats. All cited papers 68-70 are studies on humans and not on rats having heart rate**
**greater than 300 beats/min.**

We thank you for this comment. Strain rate measurements being time derivatives of strains they are
subject to a high variability and low confidence in terms of magnitudes, especially in the acute phase
when contractility was highest, and the 16 frames cine sequences were certainly insufficient to capture
the rapid early filling and LA booster. Since the strain of the atria is by itself alone a good indicator of
the contractile abnormality of the atrial wall and of its tissue remodeling, we have now removed strain
rates from **figure 1C** in the revised version of the manuscript.

**c. Line 105-107 “Furthermore, end-diastolic atrial contraction wave (A) was predominant over**
**the early LV filling wave (E), resulting in a reduced E/A ratio, revealing an abnormal LV**
**relaxation and a compensatory left atrial (LA) contraction.” When rats with high resting heart**
**rates further accelerate their heart rate during acute stress, it is difficult to separate the E and A**
**waves in the mitral inflow. In this case, the apparent high A wave may reflect the high E wave**
**(“Am J Physiol Heart Circ Physiol 283: H346–H352, 2002: “Because the heart rate was greater**

**than 300 beats/min in most rats, it was difficult to clearly identify the A wave on transmitral flow**
**velocity spectra. For this reason, measurement of the peak velocity of the A wave and DT was**
**much less feasible than E wave measurement in all three groups”).**

We thank you for this very pertinent comment. In contrast to the article that you cite, we used here a
high-resolution ultrasound scanner (prf: 20kHz to 30kHz, frame rate: 100 frames/s) under general
anesthesia of the rats, in order to allow a good visualization of the E/A curves (see **Figure Curve 1**
hereunder). However, after verification of all E/A curves, we admit that a partial or total fusion of these
waves is present in some rats independently of the pre- and post-ISO times (see **Figure Curve 2**). This
is probably a consequence of the high heart rate in spite of the anesthesia, as observed in the study of
Fabrice Prunier *et al.*³⁴ that you mentioned. In order to avoid any misinterpretation, we therefore decided
to remove the graph representing the variations of the E/A ratio during the pre- and post-ISO times from
**the extended data 4.**

Curve 1: Visible E and A mitral waves

Curve 2: Non visible E and A mitral waves

**d. In addition, in figure 1C, there was no evidence of compensatory LA contraction (LA booster**
**pump function), while it is possible that the strain rate measurements were incorrect.**

We agree with the reviewer that the atrial strain rate measurements were not fully reliable and have
removed them from **figure 1C** in the revised version of the manuscript (see **answer to question b**).

**e. During the chronic phase of the disease, atrial volume is an excellent indicator of atrial load.**
**Please display left and right atrial volume data as atrial volume can be measured by CMR.**

The strain measurements for the atria were preferably performed on the 4 chamber views for reliability
and repeatability reasons. We were accordingly able to determine the minimum and maximum atrial
surfaces on this same view as a surrogate of the atrial size. We now show LA surfaces normalized to the
baseline value for each animal in **Figure 1C**. It shows a decrease of the atrial area at 2h post-ISO that
was coupled to ISO-mediated left ventricular hypercontractility. The atrial area tended to increase at the
late phases, at 7d and 3mo post-ISO, suggesting a long-term enlargement of the left atrium. This trend
is consistent with the significant decrease in atrial strain and the presence of fibrosis. It also shows that
strain measurements have a better sensitivity than atrial area/volume measurements.

We have added the atrial surfaces measurements in **Figure 1C** and modified the revised manuscript
accordingly (**pages 9-10, lines 241-245; page 12, lines 312-317; pages 17, lines 434-437**).

f. “Eur Heart J Cardiovasc Imaging 2020;21:1184-1207”. Multimodality imaging in Takotsubo Syndrome: a joint consensus document of the European Association of Cardiovascular Imaging (EACVI) and the Japanese Society of Echocardiography (JSE) “RV free wall longitudinal strain is therefore recommended to detect even small areas of myocardial dysfunction hardly recognizable by visual assessment and/or by other traditional echocardiographic parameters. Conversely, RV global longitudinal strain should not be assessed because involvement of the interventricular septum can be misleading.” In this study (figure 1A), the RV longitudinal strain was decreased in the acute phase and the LV longitudinal strain was increased. Since both ventricles share the ventricular septum, these results suggest a reduction in RV free wall strain. “Recent studies have further demonstrated RV involvement in TTS as an important finding associated with worse outcome.” (Eur Heart J Cardiovasc Imaging 2020;21:1184-1207). The finding that LA remodeling occurs in the chronic phase of TTS with RV damage in the acute phase has important implications for the pathophysiology of TTS.

We thank you the reviewer for highlighting this relevant point. We have now taken this comment into account and cited the corresponding references in the Results section (page 5, lines 115-118). We have also added a mention in "the study limitations" section of the revised manuscript (pages 13-14, lines 336-356).

g. Based on a human study, “TTS patients demonstrated a significantly decreased LA function during the acute/subacute phase of the disease. However, impairment of LA performance seems to be transient in TTS with recovery during follow-up.” (Journal of Cardiovascular Magnetic Resonance (2017) 19:15). As observed in the current study, LA function in the chronic phase was reduced and complicated by LA fibrosis, suggesting that this is an animal model which recapitulates a slightly more severe form of TTS.

The hypothesis expressed by Stiermayer T *et al.* concerning the reversibility of the LA dysfunction points to a difference between our study in rats and their clinical results. This could result from the different timelines between the two different species, our late, 3-month timepoint corresponding to perhaps a decade of human life³⁵. Moreover, contrary to our study, Stiermayer T *et al.* did not measure the strain and the fibrosis in the LA, which impedes the interpretations regarding the transient LA dysfunction.

h. Although the term GLS (global longitudinal strain) was used in this study, it is preferable to use the simple term “longitudinal strain”, since this study did not use the average value of longitudinal strain in multiple cross sections (2-, 3-, and 4-chamber view). If the average of longitudinal strain of multiple cross sections was used, please state how many cross-sectional averages were used for the LV, RV, LA, and RA.

We thank you for this suggestion. We did not measure the longitudinal strain in all incidences.
Therefore, we have made the corrections throughout the revised version of the manuscript.

**i. Line 149 “Parameters derived from cardiac ultrasound images were normal at 7d post-ISO**
**(figure 6; extended data 2 and 4).” There was no data of ultrasound images in figure 6.**

Thank you, we have now corrected this error (page 6, lines 146-147).

**2. The plasma concentrations of AST, ALT and CK were measured. These are non-specific and**
**could be raised in various conditions, which are non-cardiac. If the aim was to measure**
**myocardial damage, would the authors explain why cardiac-specific markers, e.g., CK-MB,**
**troponins (T or I) were not measured?**

As reported in numerous studies, the specific and validated plasma biomarker of TTC is primarily BNP
(brain natriuretic peptide), a late marker of heart failure that is found to be higher in TTC patients than
in patients with acute coronary syndrome³⁶⁻³⁹. Accordingly, we observed a trend towards a higher
plasma concentration of BNP at 7d post-ISO. Studies have shown that the evolutionary profiles of BNP
and troponin were opposite in TTC (lower troponin and higher BNP) compared with acute coronary
syndrome³⁸⁻⁴⁰, which motivated us to focus on the plasma BNP assay rather than on other cardiac
biomarkers.

**3. Overall, in this study, a rat model of stress-induced cardiomyopathy is used to observe acute**
**and long-term myocardial damage via high-end imaging and microscopic imaging. The article is**
**well written and highly interesting to read. However, despite a huge amount of work, the article**
**eventually offers as only claim to provide early treatment to patients with Takotsubo, which is**
**intuitive. Please expand to what other of the current patient management the authors may refer...**

As of today, all the treatments proposed for Takotsubo have been based on hypotheses, and indeed, a
recent review⁴¹ concluded with the Hippocratic recommendation “first do no harm”. Multicenter
randomized clinical trials are certainly needed to estimate the efficacy, reliability and long-term
prognosis of these treatments. A recent search on clinicaltrials.gov (October 17, 2022) recovered 16
trials, essentially based on physical exercise, or cognitive, behavioral or sex therapies. Only two drug
trials are mentioned, one using adenosine and one alpha-lipoic acid and L-acetylcarnitine. No study
results are reported for these latter studies. Recommendations were published in 2018 in an international
expert consensus document³⁷. When TTC patients present to the hospital with an ACS presentation, the
therapeutic management is organized for ACS and not for TTC, which means administration of aspirin
and heparin and, if necessary, morphine and oxygen. In TTC, the administration of beta-blockers has
been proposed because of the possible role of the acute catecholaminergic surge. Results from the

InterTAK registry showed that survival was comparable between patients receiving or not beta-blockers
at discharge⁶. Approximately 30% of TTC patients had a recurrence during beta-blocker treatment,
which raises questions about their cardioprotective effect in TTC⁶. Anticoagulation is recommended in
TTC patients with atrial fibrillation and thromboembolic complications, whereas prophylactic
administration of anticoagulants could be considered in patients with low LVEF or LV dysfunction
involving the apex until LV function improves. Thus, as far as we know, there are presently no
randomized trials of specific treatments for TTC, and we do not know what benefits could be expected
from any treatment.

In addition, the diagnosis of TTC can be difficult because of the similarity of clinical signs to those
encountered in AMI or acute coronary syndrome (ACS), particularly ECG abnormalities and myocardial
biomarkers⁵. Finally, there is a real lack of noninvasive approaches for rapid and reliable differential
diagnosis of TTC, as only LV coronography is considered a reliable diagnostic tool to distinguish
between these pathologies and make the diagnosis of TTC.

Our study shows that an overdose of catecholamines, in this case isoprenaline, leads to a metabolic
remodeling that persists over time. We show that there is a switch in myocardial metabolism with an
inhibition of oxidative phosphorylation and an overconsumption of glucose that is diverted to anabolic
pathways, in particular to the polyol and biosynthetic hexosamine pathways. We strongly suggest that
it is this bypass and overactivation of these two pathways that would be largely responsible for the
significant presence of fibrosis first in the apex and then throughout the LV and left atrium and that
would also be responsible for remodeling of the cardiac cytoskeleton and pathological angiogenesis. It
thus would be interesting to consider treatments that modulate these anabolic pathways of glucose,
especially if it can be demonstrated in parallel, that they attenuate fibrosis.

In our study, we highlighted the importance of myocardial strain measurements and FDG PET imaging
(or cardiac PETRUS) to assess and detect both regional and global functional, tissue, and metabolic
remodeling of the heart. We plead that the combination of these in vivo non-invasive imaging methods
will be crucial to improve the management of TTC patients, the diagnosis and ultimately the prognosis
of TTC. Since TTC is multifactorial and complex and often associates with comorbidity, the use of a
reductionist animal model has allowed us to detect imaging biomarkers characteristic of TTC that
deserve to be translated and explored in the clinic.

We have added part of the present comments in the Discussion section of the revised manuscript (**page**
**13, lines 328-333**).

**4. The whole study is rationalized by the fact that long-term cardiovascular annual death rates**
**for Takotsubo Syndrome and acute coronary syndrome are the same. Supportive references are:**
**1). a review article with expert panel from Lyon et al. that did not report survival rates from**
**Takotsubo syndrome; and 2). an observational study that reported a large heterogeneity in long-**
**term outcomes from Takotsubo syndrome, depending on the identified cause. Whilst those caused**
**by physical activities or neurological disorders have poorer or equal outcomes than patients with**
**acute coronary syndrome, those with emotional trigger (by far the most!) had a much better**

**outcomes than the patients with acute coronary syndrome. This points to a significant**
**heterogeneity of all sub-syndromes included in the umbrella of Takotsubo, and probably of the**
**subsequent long-term damage. As such, it would be reductive to consider that a single model**
**would help decipher the whole spectrum. It is thus important to refocus the work and the writing**
**on the subtype which is addressed by the proposed model.**

It is absolutely and entirely right to underline the complexity of this multifactorial and multifaceted
cardiomyopathy. Takotsubo is a pathology discovered and most frequently found in menopausal
women, but that also concerns men and younger women, and that has multi-cause apart from ISO stress
(genetic, physical, neurological disorders, pheochromocytoma etc). We have indeed limited our study
to a reductionist, reproducible and homogeneous animal model with a single acute onset of isoprenaline,
independently from aging, comorbidity, repetitive stress and differences in the stressors triggering the
disease. The interest of this approach was above all to understand the mechanisms involved and their
chronology during an excessive administration of beta-adrenergic agonists as it can happen in clinic⁴².
We have shown that a single injection of isoprenaline resulted in severe long-term deleterious effects in
young adult rats. We assume that these observed effects will be much more consequential in elderly
subjects and with comorbidity or with another stressor triggering the disease such as neurological
disorder⁴³⁻⁴⁵.

The choice of the reductionist model is a strength but also a limitation of our study. It would be reductive
to consider that a single model would help decipher the whole TTC spectrum. Following your comment,
we have added a paragraph " study limitations" to put our study in the global context of TTC (**pages 13-**
**14, lines 336-356**).

**5. The claim that overactivation of the hexosamine and polyol biosynthetic pathways explain tissue**
**and vascular remodeling and induction of reactive oxygen species is proven by the current work.**
**However, in my opinion, at least another hypothesis has been neglected based on the study**
**findings: the cell-cell junction alteration (that is presumably one of the results of hyperdynamism).**
**In a different, but close disease model, extreme athletes may end having acute myocardial damage**
**and even long-term fibrosis via hyper dynamism without any alteration of the glucose metabolism.**
**In general, the hypothesis-generating part of this work should be enriched to include other**
**potential pathways for myocardial damage.**

We thank you for this very interesting comment. We indeed observed a partial rupture of the intercalary
disc and thus of the cell-cell junction at 2h post-ISO by transmission electron microscopy (**cf. apical**
**images D, E and F and basal images L, M and N of extended data 6**). However, this rupture was
transient and resorbed from 7d post-ISO and was observed in both the apical and basal regions. In
contrast, at 7d we observed metabolic and tissue remodeling (significant presence of fibrosis) only in
the apex. In addition, myocardial strain measurements indicated transient hypercontractility
(hyperdynamism) only in the basal area at 2h post-ISO and apical akinesia. If there had been an effect

of this mechanical stress (hyperdynamism), we should have seen the appearance of fibrosis primarily in
the basal area, which was not the case. Therefore, we believe that the appearance of fibrosis is strongly
related to apical akinesia coinciding with an over-activation of glucose anabolic pathways.

**6. The demonstration of the absence of coronary obstruction part seems superfluous and could be**
**moved to the additional data.**

We report the results of coronary flow reserve in the "supplementary data" section (**extended data 2**),
as the increase in coronary flow reserve at 2 hours indicates normal coronary dilator function and is
therefore a good indicator of the absence of left coronary artery obstruction. We believe it is important
to mention this point since it discriminates TTC from acute coronary syndrome.

**7. Page 2, line 37. Could the authors mean ‘reversibility’, rather than ‘irreversibility’?**

Thank you for pointing out this typo, now corrected (**page 2, line 38**).

**8. Please only capitalise proper nouns. As such, ‘Humans’, ‘Ultrasound’ etc. should be in small**
**letters.**

We now made corrections in the revised manuscript accordingly.

**4. Reviewer #4 (Remarks to the Author):**

The overall goal of this study was to examine the effects of an acute pharmacological stress on the heart
designed to mimic Takotsubo syndrome and to determine the contributions of acute and chronic
metabolic remodeling to the long-term damage to the myocardium.

There are however some serious limitations their evaluation and interpretation of the data.

**1) In the text the authors state that 2h post-ISO PET imaging showed a 50% increase in the uptake**
**of FDG into the heart (lines 169-170). In Figure 2A the PET images demonstrate marked increase**
**in FDG uptake at both 2 hours and 7days; however, this is not reflected in any of the quantified**
**data presented in Fig 2B.**

We thank the reviewer for pointing to the unsuitable representation of the data in Figure 2B, in which
boxplots do not highlight the mean and standard deviation of FDG uptake and are misleading. The
revised version of the manuscript presents the same data using histograms based on the mean and
standard deviation values, which is all the more consistent with the statistical analyses. The new Figure

2B makes clear the 47% increase over baseline of the mean SUV in the myocardium at 2h post-ISO,
and the 36% increase over baseline at 7d post-ISO. It is true that, in this case as often, quantification of
PET imaging from small animal series yields statistically non-significant results, although studies cannot
be extended to large cohorts for practical and ethical reasons. This is the reason why we have performed
additional extensive histological and proteomic analyses (Figures 2 and 4 and extended data 8A) that
support the patterns of FDG uptake and kinetics in the myocardium. Amendments have now been made
accordingly in the revised manuscript (page 7, lines 167-169).

**2) While the data presented in Fig 2B do not appear to show any substantial changes at 2h post-**
**ISO, it is stated that FDG uptake was increased by 70% whereas the index of glucose**
**phosphorylation was reduced. They refer to the apparent disconnect between changes in glucose**
**uptake and phosphorylation as “metabolic stunning”. They suggest that this low capacity to use**
**glucose as an energy source is contributes to cardiac dysfunction; however, this conclusion is**
**flawed. It is widely recognized that glucose is not the predominant fuel for mitochondrial ATP**
**production, rather fatty acids are the largest source with contributions from lactate, ketone bodies**
**amino acids (For a recent review see Circulation Research. 2021; 128:1487–1513). Thus, in the**
**absence of any other measures of substrate utilization it is not possible to conclude that potential**
**changes in glucose metabolism have any meaningful effects. In addition, in Extended Data 3C, the**
**circulating plasma concentrations of lactate and fatty acids increase at least 2-fold 2h post-ISO**
**both of which can be readily oxidized by the heart and likely contribute to the apparent decrease**
**in glucose metabolism. Indeed, high circulating lactate levels have been associated with reduced**
**cardiac glucose uptake (e.g., J Physiol. 2002;542: 403-12).**

We observed at 2h post-ISO an average increase of 70% of the rate constant K_1 compared to its baseline
value. As mentioned in the text " K_1 , the rate constant for FDG passage from blood to tissue, increased
by 70%" (page 7, line 170), this rate constant reflects the passage of glucose from plasma to tissue and
is calculated using two-compartment model analysis equations derived from dynamic PET imaging. In
contrast, SUV values do not take into account the kinetics of FDG uptake. Thus, the entry of glucose
in the myocardium at 2h post-ISO is increased and coincides with the increase in GLUT1 expression in
cardiomyocytes (Figure 3).

The term "metabolic stunning"⁴¹ (page 7, lines 173-175) has been used to define the lack of metabolic
flexibility of the myocardium in conditions of increased energy demand, e.g., hypercontractility and
systolic dysfunction. Although high levels of glucose, fatty acids, or lactate, are available in the plasma,
the myocardial energy intake (the actual consumption of these substrates by the heart) does not adapt to
the demand.

In our study, we now used the term “glucose metabolic stunning” to define the mismatch between
hyperglycemia, hypercontractility of the LV, directly mediated by isoprenaline (extended data 3C),
and the low or unchanged use of glucose as an energetic substrate by the myocardium, (Figure 1A,
Figure 4, extended data 4). Fatty acids are indeed the major energy source in the heart, but glucose

remains the second preferred energy substrate, especially during rapid augmentations of contraction
because, in contrast to other energy sources, glycolysis can rapidly produce ATP without the activation
of mitochondrial oxidative phosphorylation. This is the reason why we have named “metabolic
stunning” the energetic situation of the myocardium at 2 h post-ISO⁴⁶ (**Figure 3**). We reckon with the
reviewer that the term may not be well chosen here because of the increased plasmatic concentrations
of the other energy substrates. However, we did show using oil-red staining an accumulation of lipid
droplets at 2 hours, indicating that compensation of energetic needs by fatty acid beta-oxidation,
essentially a mitochondrial process, is highly unlikely (**extended data 5B**). The energetic status of the
heart after ISO resembles the myocardial glucose intolerance, as shown in the diabetic heart by
Stratmann *et al.*, in which lipid toxicity is caused by the decrease of glucose uptake by the myocardium
and causes insulin resistance with decreased GLUT4 activity⁴⁷. At 2 h post-ISO, the heart is in a similar
situation as that described by Stratmann and co-authors: on the one hand, the activation of glucose
required for its entry into glycolysis (hexokinase activity reflected by the rate constant k_3) decreases in
the myocardium and, on the other hand, there is no change in the level of GLUT4 expression in
cardiomyocytes compared with the control group, whereas this glucose transporter is insulin-dependent
and should therefore be translocated to the membrane surface in response to catecholamine mediated
hyperglycemia⁴⁸.

**In addition, in Extended Data 3C, the circulating plasma concentrations of lactate and fatty acids**
**increase at least 2-fold 2h post-ISO both of which can be readily oxidized by the heart and likely**
**contribute to the apparent decrease in glucose metabolism. Indeed, high circulating lactate levels**
**have been associated with reduced cardiac glucose uptake (e.g., J Physiol. 2002;542: 403-12).**

Although there is a favoring of one energy substrate over another according to their plasma availability
and according to physiological conditions by the myocardium and although there is a significant plasma
concentration of fatty acids, lactate, and glucose at 2 h post-ISO, the hypothesis that you have formulated
concerning the decrease in glucose consumption by the myocardium at 2h post-ISO cannot be valid in
our study since it is true that the concentrations of fatty acids and lactate increase 2h post-ISO, and so
does glycemia. However, the plasma concentrations of these three potential energy substrates appear
decorrelated with their use as energy sources in the post-ISO myocardium. Firstly, as explicated in the
response to the previous comment of the reviewer, increased glucose entry is associated with decreased
or, at best, unchanged glycolysis. Secondly, regarding fatty acids, proteomics did not show an increase
in fatty acid beta-oxidation at 2h post-ISO, and there was no change in the expression of the fatty acid
transporter, FABP4 and FABP3, between 0 and 2h post-ISO, which suggests an absence of massive
entry of fatty acids for energy metabolism into the myocardium (**Table below and extended data 8A**).
The accumulation of lipid droplets at 2h post-ISO confirms that fatty acids do not act as supplementary
energy source in the post-ISO heart.

Regarding lactate, it also requires the activation of oxidative phosphorylation to be used as a source of
energy, which we have not observed. Myocardial lactate metabolism is favored over glucose metabolism

in the case of insufficient exogenous glucose supply⁴⁹, while the post-ISO heart is offered plenty of
glucose by the plasma and the increased transport rate constants.

Consequently, all of these data and observations indicate a deregulation of myocardial glucose
metabolism and intolerance at 2h post-ISO, a phenomenon that most certainly has consequences on the
progression of the pathology. Moreover, as shown by Scally C *et al.*¹ using cardiac ³¹P-spectroscopy,
there is a global deficit in ATP production that is typical of impaired cardiac energy metabolism.

We have completed the tables in extended data 8A (**supplementary information, page 8**).

Symbol	Gene Name	2h vs 0		7d vs 0	
		p-value	Fold Change	p-value	Fold Change
FABP3	fatty acid binding protein 3			7.08E-03	-1.554
FABP4	fatty acid binding protein 4			1.63E-02	-1.373

**3) Lines 180-181: “During the recovery stage (7d post -ISO), GLUT1 staining was high in the apex**
**but returned to baseline in the basal region, while an increase of GLUT4 staining was observed in**
**both regions”.** However, in Fig 3 there are no changes in GLUT4 levels in the basal region at any
time point. Moreover, measurements of total GLUT1 and GLUT4 protein levels provide no
information regarding their contributions to glucose uptake, as only membrane levels are of
relevance to glucose metabolism.

Indeed, there was no observable statistical difference in the expression level of GLUT4 at the different
pre- and post-ISO time points in the LV basal region. However, comparison with baseline suggested a
potential increase in the mean expression level of GLUT4 at 7 d post-ISO when (+107%, p=0.1262).
We have now corrected the revised manuscript appropriately (**page 7, lines 179-180**).

We completely agree with the reviewer that the translocation of GLUTs to the plasma membrane of
cardiomyocytes is the key indicator of the level of glucose entry into cells. Immunofluorescence studies
of cardiac sections allow to localize the distribution of GLUTs at the cellular level, which is more
meaningful than their expression level. Figure 3 shows that GLUTs are mainly membrane-based at 2h
post-ISO and at 7d post-ISO. At this time, we observe intense GLUT4 staining in all cellular
compartments, suggesting a saturation of its expression at the membrane surface in particular in the
apex. Moreover, proteomic analyses are consistent with these observations as they indicate the
significant activation of signaling pathways involved in GLUT4 translocation in the apex at 7d post-
ISO compared to control and at 2h post-ISO (**supplementary information, page 8**). Indeed, GLUT4
translocation is mainly insulin-dependent^{50,51} and proteomic analyses indicate a significant
overactivation of the insulin secretory pathway and also of its receptor pathway at 7d post-ISO
(**supplementary information, page 8, extended data 8A**). The PI3K/AKT pathway is also involved in
the metabolic effects of insulin and contributes to insulin-mediated translocation of GLUT4^{52,53}. This
pathway is significantly activated at 7d post-ISO compared to control and at 2h post-ISO
(**supplementary information, page 8, extended data 8A**). Activation of PI3K catalyzes the
phosphorylation of phosphoinositides, generating a signaling cascade including the phosphorylation of

AKT. The synthesis of phosphoinositides is also overactivated at 7d post-ISO. Another insulin-
independent pathway known to promote GLUT4 translocation in the heart is the AMPK (adenine
monophosphate-activated protein kinase) pathway^{54,55}. The AMPK pathway is also activated at 7 days
post-ISO in the apex (**supplementary information, page 8, extended data 8A**). Thus, there is both
direct (immunofluorescence staining of heart sections) and indirect (overactivation of the pathways
mentioned above) evidence for an increased translocation of GLUT4 to the cardiomyocytic plasma
membrane, and this evidence is corroborated by the increased entry of FDG in the apical cardiomyocytes
at 7d post-ISO. Taken together, these results demonstrate a significantly augmented glucose entry in the
cardiac apex at 7d post-ISO.

We have added these signaling pathways involved in the activation of GLUT4 translocation to the
membrane surface in **extended data 8A (supplementary information, page 8)**.

**4) Lines 193-194 “A high myocardial uptake of glucose increases collagen synthesis, myocardial**
**myofibroblast proliferation, and the expression of fibronectin and transforming growth factor**
**(TGF)- β 1”. This is a misinterpretation of the studies cited, which show that high extracellular**
**glucose levels stimulate fibrotic changes in cultured cardiac myoblasts. I am unaware of any**
**evidence that increased myocardial uptake of glucose induces fibrotic like changes.**

We thank the reviewer for his comment pointing out to an ambiguity due to text abridging. We meant
to say that (i) excess glucose entering the myocardium is diverted to alternative non-energetic pathways
and (ii) that a consequence of the overactivity of these pathways is extensive tissue and vascular
remodeling. As observed here in a Takotsubo model, several studies have demonstrated the
consequences of overactivation of the HBP and O-GlcNAcylation on myocardial function and structure.
Tran *et al.* showed that overactivation of the HBP promoted cardiac hypertrophy in vitro, with a
significant increase in the size of cardiac cells, protein synthesis and the expression of hypertrophy
markers. They showed that overactivation of the limiting enzyme GFAT1 of the HBP by a tetracycline-
induced promoter, led to an increase in the size of the heart and to cardiac fibrosis in transgenic mice³.
Their results and those of similar studies indicate that the overactivation of HBP is liable for significant
remodeling of the cytoskeleton and the ECM, as well as for vascular remodeling^{56,57}. The results from
these studies is similar to what we observed at 7-d post-ISO in the LV apex, namely hyperactivation of
the HBP coinciding with the presence of fibrosis and changes in the vascular network and cytoskeleton
of cardiomyocytes (see Figure 4, Figure 5A, Figure 6, extended data 8B and C). Since the HBP pathway
generates uridine diphosphate Nacetylglucosamine (UDP-GlcNAc), an essential substrate for O-
GlcNAcylation, a post-translational modification that plays an important regulatory role on O-linked β -
Nacetylglucosamination (O-GlcNAcylation) of cytoskeletal and ECM proteins⁵⁸⁻⁶¹. A significant
increase in O-GlcNAcylation has been observed in hearts with hypertension^{62,63}, diabetes^{4,64}, chronically
hypertrophied hearts and after heart failure⁶², and is thought to contribute to contractile and
mitochondrial dysfunction⁶⁵. Fülöp *et al.* also showed that O-GlcNAcylation contributed to impaired
function in diabetic cardiomyocytes⁴. In our study, the western blot quantification of O-Glycosylated

proteins indicated that they were significantly increased at 7d post-ISO in the apex (see Figure 4B),
reflecting a high level of O-GlcNAcylation, consistent with the overactivation of the HBP observed by
proteomic and western blot analyses. It has been shown in mice that overactivation of GFAT1 increases
O-GlcNAcylation and fibrosis⁶⁶. In rats, GFAT1 is the cardiomyocyte-specific isoform of GFAT, while
GFAT2 is the isoform specifically expressed in rat myofibroblasts⁶⁶. Both GFAT1 and GFAT2 were
significantly overexpressed at 2h and 7d post-ISO in the LV, indicating overactivation of the HBP and
O-GlcNAcylation in cardiomyocytes and myofibroblasts at 2h and 7d post-ISO. Excess post-
translational modifications of cardiac myofilaments by O-GlcNAcylation has been shown to induce
cardiomyocyte dysfunction^{67,68}.

All these studies support the link between overactivation of the HBP and functional, tissue, and
structural changes in the myocardium, in agreement with our findings. We have reworded this section
to better highlight this link in the revised version of the manuscript (**page 8, lines 204-208; pages 11-
12, lines 283-309**).

**5) Understanding the proteomic data presented in Fig. 4 is complicated by the fact that the legend**
**states that “green, overexpression higher than 1.3-fold, red: under-expression lower than -1.3-**
**fold”.** However, in the figure the green numbers are all preceded by a negative sign and the red
numbers have no sign. In the text, which is mostly focused on the HBP and Polyol pathways it is
stated that proteins in these pathways are overexpressed, so despite the legend I have assumed
that the red colors indicate overexpressed proteins.

We thank the reviewer for pointing out to this error and apologize for this mistake. Legend to **Figure 4**
has now been corrected accordingly (**page 30, lines 654-656**).

**The conclusion for this specific section is “notion that HBP hyperactivation induced by an acute**
**stress is a driver of cardiac fibrosis and angiogenesis”** which as stated in the Discussion this is due
**to increased O-GlcNAcylation. However, it is not possible to reach this conclusion based on the**
**data presented. It is true that key elements of the HBP and O-GlcNAc regulation are increased at**
**7 days post-ISO. While the authors have highlighted changes in GFAT expression they have**
**overlooked the 8-fold increase in O-GlcNAcase (OGA) expression, which removes O-GlcNAc from**
**proteins. Therefore, assuming that protein expression equates to activity, then OGA activity is**
**increased more 5-times that of OGT, which if true would result in reduced O-GlcNAc levels not**
**an increase. It is not possible to draw any conclusions regarding changes in O-GlcNAcylation,**
**without a measurement of O-GlcNAc levels at 0 and 7d post-ISO. Immunoblots of some of these**
**key proteins to support the changes observed in the proteomic studies would also be helpful.**

**6) Lines 304-306: “Finally, overactivation of O-GlcNAcylation increases synthesis of extracellular**
**matrix proteins, as observed here at 7d post-ISO in the apex.”** Apart from the fact that it cannot
be concluded that there are any changes in O-GlcNAc levels, the references cited to support this

**statement indicate that increased HBP flux in mesangial cells is linked to increased synthesis of**
**extracellular matrix proteins. Neither study includes measurements of O-GlcNAc levels.**

The level of O-GlcNAcylated proteins depends on OGT that adds GlcNAc to proteins, and on OGA
that removes it. We agree with the reviewer that estimating the overall assessment of O-GlcNAcylated
is not directly deducible from the levels of expression of the two enzymes, since the level of regulation
is mainly transcriptional for OGA and mainly post-transcriptional for OGT. To further complexify the
situation, there is a reciprocal regulation between these two enzymes^{69,70}. The mechanisms involved in
the regulation of OGA/OGT leading to homeostasis or conversely, to imbalance of O-GlcNAcylation
are poorly known, and it is tricky to draw conclusions from proteomic analyses measuring protein
expression. Therefore, as rightly suggested by the reviewer, we performed a western blot quantification
of O-GlcNAcylated proteins in the cardiac apex to explore the concurrent effects of both enzyme's
activities on HBP activation. Consistent with the high expression level of the rate-limiting enzyme of
the HBP and other enzymes involved in this pathway, we found a higher level at 7d post-ISO than at
baseline of O-Glycosylated proteins in the apex. We have added these new results in **Figure 4B** and
modified the manuscript in agreement (**page 8, line 207; page 11, line 285-286; page 12, lines 307-**
**309; pages 19-20, lines 508-519**).

REFERENCES

- 1. Scally, C. *et al.* Persistent Long-Term Structural, Functional, and Metabolic Changes After Stress-
Induced (Takotsubo) Cardiomyopathy. *Circulation* **137**, 1039–1048 (2018).
- 2. Schwarz, K. *et al.* Alterations in Cardiac Deformation, Timing of Contraction and Relaxation, and
Early Myocardial Fibrosis Accompany the Apparent Recovery of Acute Stress-Induced
(Takotsubo) Cardiomyopathy: An End to the Concept of Transience. *J. Am. Soc. Echocardiogr.* **30**,
745–755 (2017).
- 3. Tran, D. H. *et al.* Chronic activation of hexosamine biosynthesis in the heart triggers pathological
cardiac remodeling. *Nat. Commun.* **11**, 1771 (2020).
- 4. Fülöp, N. *et al.* Impact of Type 2 diabetes and aging on cardiomyocyte function and O-linked N-
acetylglucosamine levels in the heart. *Am. J. Physiol. Cell Physiol.* **292**, C1370-1378 (2007).
- 5. Prasad, A., Lerman, A. & Rihal, C. S. Apical ballooning syndrome (Tako-Tsubo or stress
cardiomyopathy): A mimic of acute myocardial infarction. *Am. Heart J.* **155**, 408–417 (2008).
- 6. Templin, C. *et al.* Clinical Features and Outcomes of Takotsubo (Stress) Cardiomyopathy. *N. Engl.*
*J. Med.* **373**, 929–938 (2015).
- 7. Pritchett, A. M. *et al.* Diastolic dysfunction and left atrial volume. *J. Am. Coll. Cardiol.* **45**, 87–92
(2005).
- 8. Aouar, L. M. M. E. *et al.* Relationship between left atrial volume and diastolic dysfunction in 500
Brazilian patients. *Arq. Bras. Cardiol.* **101**, 52–58 (2013).
- 9. McClain, D. A. Hexosamines as mediators of nutrient sensing and regulation in diabetes. *J.*
*Diabetes Complications* **16**, 72–80 (2002).
- 10. Trueblood, N. & Ramasamy, R. Aldose reductase inhibition improves altered glucose metabolism
of isolated diabetic rat hearts. *Am. J. Physiol.* **275**, H75-83 (1998).
- 11. Grewal, A. S., Bhardwaj, S., Pandita, D., Lather, V. & Sekhon, B. S. Updates on Aldose Reductase
Inhibitors for Management of Diabetic Complications and Non-diabetic Diseases. *Mini Rev. Med.*
*Chem.* **16**, 120–162 (2016).
- 12. Srivastava, S. K., Ramana, K. V. & Bhatnagar, A. Role of aldose reductase and oxidative damage
in diabetes and the consequent potential for therapeutic options. *Endocr. Rev.* **26**, 380–392 (2005).
- 13. Lipson, K. E., Wong, C., Teng, Y. & Spong, S. CTGF is a central mediator of tissue remodeling
and fibrosis and its inhibition can reverse the process of fibrosis. *Fibrogenesis Tissue Repair* **5**, S24
(2012).

- 14. Szabó, Z. *et al.* Connective tissue growth factor inhibition attenuates left ventricular remodeling
and dysfunction in pressure overload-induced heart failure. *Hypertens. Dallas Tex 1979* **63**, 1235–
1240 (2014).
- 15. Li, X., Tang, X., Lu, J. & Yuan, S. Therapeutic inhibition of galectin-3 improves cardiomyocyte
apoptosis and survival during heart failure. *Mol. Med. Rep.* **17**, 4106–4112 (2018).
- 16. Sharma, U. C. *et al.* Galectin-3 marks activated macrophages in failure-prone hypertrophied hearts
and contributes to cardiac dysfunction. *Circulation* **110**, 3121–3128 (2004).
- 17. Calvier, L. *et al.* Galectin-3 mediates aldosterone-induced vascular fibrosis. *Arterioscler. Thromb.*
*Vasc. Biol.* **33**, 67–75 (2013).
- 18. Suthahar, N. *et al.* Galectin-3 Activation and Inhibition in Heart Failure and Cardiovascular
Disease: An Update. *Theranostics* **8**, 593–609 (2018).
- 19. Mackinnon, A. C. *et al.* Regulation of transforming growth factor- β 1-driven lung fibrosis by
galectin-3. *Am. J. Respir. Crit. Care Med.* **185**, 537–546 (2012).
- 20. de Boer, R. A., Voors, A. A., Muntendam, P., van Gilst, W. H. & van Veldhuisen, D. J. Galectin-
3: a novel mediator of heart failure development and progression. *Eur. J. Heart Fail.* **11**, 811–817
(2009).
- 21. Yu, L. *et al.* Genetic and pharmacological inhibition of galectin-3 prevents cardiac remodeling by
interfering with myocardial fibrogenesis. *Circ. Heart Fail.* **6**, 107–117 (2013).
- 22. Thum, T. *et al.* MicroRNA-21 contributes to myocardial disease by stimulating MAP kinase
signalling in fibroblasts. *Nature* **456**, 980–984 (2008).
- 23. Lopez-de la Mora, D. A. *et al.* Role and New Insights of Pirfenidone in Fibrotic Diseases. *Int. J.*
*Med. Sci.* **12**, 840–847 (2015).
- 24. Edgley, A. J., Krum, H. & Kelly, D. J. Targeting fibrosis for the treatment of heart failure: a role
for transforming growth factor- β . *Cardiovasc. Ther.* **30**, e30-40 (2012).
- 25. Nguyen, D. T., Ding, C., Wilson, E., Marcus, G. M. & Olgin, J. E. Pirfenidone mitigates left
ventricular fibrosis and dysfunction after myocardial infarction and reduces arrhythmias. *Heart*
*Rhythm* **7**, 1438–1445 (2010).
- 26. Mirkovic, S. *et al.* Attenuation of cardiac fibrosis by pirfenidone and amiloride in DOCA-salt
hypertensive rats. *Br. J. Pharmacol.* **135**, 961–968 (2002).
- 27. Yamazaki, T. *et al.* The antifibrotic agent pirfenidone inhibits angiotensin II-induced cardiac
hypertrophy in mice. *Hypertens. Res. Off. J. Jpn. Soc. Hypertens.* **35**, 34–40 (2012).
- 28. Yamagami, K. *et al.* Pirfenidone exhibits cardioprotective effects by regulating myocardial fibrosis
and vascular permeability in pressure-overloaded hearts. *Am. J. Physiol. Heart Circ. Physiol.* **309**,
H512-522 (2015).
- 29. Wang, J. J.-C. *et al.* Genetic Dissection of Cardiac Remodeling in an Isoproterenol-Induced Heart
Failure Mouse Model. *PLOS Genet.* **12**, e1006038 (2016).
- 30. Samuel, C. S. *et al.* Relaxin remodels fibrotic healing following myocardial infarction. *Lab.*
*Investig. J. Tech. Methods Pathol.* **91**, 675–690 (2011).
- 31. Samuel, C. S., Hewitson, T. D., Zhang, Y. & Kelly, D. J. Relaxin ameliorates fibrosis in
experimental diabetic cardiomyopathy. *Endocrinology* **149**, 3286–3293 (2008).
- 32. Lekgabe, E. D. *et al.* Relaxin reverses cardiac and renal fibrosis in spontaneously hypertensive rats.
*Hypertens. Dallas Tex 1979* **46**, 412–418 (2005).
- 33. Henry, B. L. *et al.* Relaxin suppresses atrial fibrillation in aged rats by reversing fibrosis and
upregulating Na⁺ channels. *Heart Rhythm* **13**, 983–991 (2016).
- 34. Prunier, F. *et al.* Doppler echocardiographic estimation of left ventricular end-diastolic pressure
after MI in rats. *Am. J. Physiol.-Heart Circ. Physiol.* **283**, H346–H352 (2002).
- 35. Andreollo, N. A., Santos, E. F. dos, Araújo, M. R. & Lopes, L. R. Rat's age versus human's age:
what is the relationship? *ABCD Arq. Bras. Cir. Dig. São Paulo* **25**, 49–51 (2012).
- 36. Ahmed, K. A., Madhavan, M. & Prasad, A. Brain natriuretic peptide in apical ballooning syndrome
(Takotsubo/stress cardiomyopathy): comparison with acute myocardial infarction. *Coron. Artery*
*Dis.* **23**, 259–264 (2012).
- 37. Ghadri, J.-R. *et al.* International Expert Consensus Document on Takotsubo Syndrome (Part I):
Clinical Characteristics, Diagnostic Criteria, and Pathophysiology. *Eur. Heart J.* **39**, 2032–2046
(2018).
- 38. Doyen, D. *et al.* Cardiac biomarkers in Takotsubo cardiomyopathy. *Int. J. Cardiol.* **174**, 798–801
(2014).
- 39. Budnik, M. *et al.* Simple markers can distinguish Takotsubo cardiomyopathy from ST segment
elevation myocardial infarction. *Int. J. Cardiol.* **219**, 417–420 (2016).

- 40. Khan, H. *et al.* A systematic review of biomarkers in Takotsubo syndrome: A focus on better
understanding the pathophysiology. *IJC Heart Vasc.* **34**, 100795 (2021).
- 41. Madias, J. E. Takotsubo Cardiomyopathy: Current Treatment. *J. Clin. Med.* **10**, 3440 (2021).
- 42. Khwaja, Y. H. & Tai, J. M. Takotsubo cardiomyopathy with use of salbutamol nebulisation and
aminophylline infusion in a patient with acute asthma exacerbation. *Case Rep.* **2016**,
bcr2016217364 (2016).
- 43. Di Vece, D. *et al.* Outcomes Associated With Cardiogenic Shock in Takotsubo Syndrome.
*Circulation* **139**, 413–415 (2019).
- 44. Ghadri, J. R. *et al.* Long-Term Prognosis of Patients With Takotsubo Syndrome. *J. Am. Coll.*
*Cardiol.* **72**, 874–882 (2018).
- 45. Citro, R. *et al.* Long-term outcome in patients with Takotsubo syndrome presenting with severely
reduced left ventricular ejection fraction. *Eur. J. Heart Fail.* **21**, 781–789 (2019).
- 46. Kolwicz, S. C., Purohit, S. & Tian, R. Cardiac Metabolism and its Interactions With Contraction,
Growth, and Survival of Cardiomyocytes. *Circ. Res.* **113**, 603–616 (2013).
- 47. Stratmann, B. & Tschoepe, D. The diabetic heart: sweet, fatty and stressed. *Expert Rev. Cardiovasc.*
*Ther.* **9**, 1093–1096 (2011).
- 48. Barth, E. *et al.* Glucose metabolism and catecholamines. *Crit. Care Med.* **35**, S508 (2007).
- 49. Szablewski, L. Glucose transporters in healthy heart and in cardiac disease. *Int. J. Cardiol.* **230**,
70–75 (2017).
- 50. Bertrand, L., Horman, S., Beauloye, C. & Vanoverschelde, J.-L. Insulin signalling in the heart.
*Cardiovasc. Res.* **79**, 238–248 (2008).
- 51. Bevan, P. Insulin signalling. *J. Cell Sci.* **114**, 1429–1430 (2001).
- 52. Kohn, A. D., Summers, S. A., Birnbaum, M. J. & Roth, R. A. Expression of a Constitutively Active
Akt Ser/Thr Kinase in 3T3-L1 Adipocytes Stimulates Glucose Uptake and Glucose Transporter 4
Translocation *. *J. Biol. Chem.* **271**, 31372–31378 (1996).
- 53. Standaert, M. L. *et al.* Insulin Activates Protein Kinases C- ζ and C- λ by an Autophosphorylation-
dependent Mechanism and Stimulates Their Translocation to GLUT4 Vesicles and Other
Membrane Fractions in Rat Adipocytes *. *J. Biol. Chem.* **274**, 25308–25316 (1999).
- 54. Young, L. H., Coven, D. L. & Russell, R. R. Cellular and molecular regulation of cardiac glucose
transport. *J. Nucl. Cardiol.* **7**, 267–276 (2000).
- 55. Russell, R. R., Bergeron, R., Shulman, G. I. & Young, L. H. Translocation of myocardial GLUT-4
and increased glucose uptake through activation of AMPK by AICAR. *Am. J. Physiol.-Heart Circ.*
*Physiol.* **277**, H643–H649 (1999).
- 56. Qin, C. X. *et al.* Insights into the role of maladaptive hexosamine biosynthesis and O-
GlcNAcylation in development of diabetic cardiac complications. *Pharmacol. Res.* **116**, 45–56
(2017).
- 57. Hebert, L. F. *et al.* Overexpression of glutamine:fructose-6-phosphate amidotransferase in
transgenic mice leads to insulin resistance. *J. Clin. Invest.* **98**, 930–936 (1996).
- 58. Akella, N. M., Ciraku, L. & Reginato, M. J. Fueling the fire: emerging role of the hexosamine
biosynthetic pathway in cancer. *BMC Biol.* **17**, 52 (2019).
- 59. Fisi, V., Miseta, A. & Nagy, T. The Role of Stress-Induced O-GlcNAc Protein Modification in the
Regulation of Membrane Transport. *Oxid. Med. Cell. Longev.* **2017**, 1308692 (2017).
- 60. Yang, X. & Qian, K. Protein O-GlcNAcylation: emerging mechanisms and functions. *Nat. Rev.*
*Mol. Cell Biol.* **18**, 452–465 (2017).
- 61. Biwi, J., Biot, C., Guerardel, Y., Vercoutter-Edouart, A.-S. & Lefebvre, T. The Many Ways by
Which O-GlcNAcylation May Orchestrate the Diversity of Complex Glycosylations. *Mol. Basel*
*Switz.* **23**, E2858 (2018).
- 62. Lunde, I. G. *et al.* Cardiac O-GlcNAc signaling is increased in hypertrophy and heart failure.
*Physiol. Genomics* **44**, 162–172 (2012).
- 63. Lima, V. V., Rigsby, C. S., Hardy, D. M., Webb, R. C. & Tostes, R. C. O-GlcNAcylation: a novel
post-translational mechanism to alter vascular cellular signaling in health and disease: focus on
hypertension. *J. Am. Soc. Hypertens. JASH* **3**, 374–387 (2009).
- 64. Clark, R. J. *et al.* Diabetes and the accompanying hyperglycemia impairs cardiomyocyte calcium
cycling through increased nuclear O-GlcNAcylation. *J. Biol. Chem.* **278**, 44230–44237 (2003).
- 65. Mailleux, F., Gélinas, R., Beauloye, C., Horman, S. & Bertrand, L. O-GlcNAcylation, enemy or
ally during cardiac hypertrophy development? *Biochim. Biophys. Acta* **1862**, 2232–2243 (2016).
- 66. Nabeebaccus, A. A. *et al.* Cardiomyocyte protein O-GlcNAcylation is regulated by GFAT1 not
GFAT2. *Biochem. Biophys. Res. Commun.* **583**, 121–127 (2021).

- 67. Ramirez-Correa, G. A. *et al.* Removal of Abnormal Myofilament O-GlcNAcylation Restores Ca²⁺
Sensitivity in Diabetic Cardiac Muscle. *Diabetes* **64**, 3573–3587 (2015).
- 68. Ramirez-Correa, G. A. *et al.* O-linked GlcNAc Modification of Cardiac Myofilament Proteins: A
Novel Regulator of Myocardial Contractile Function. *Circ. Res.* **103**, 1354–1358 (2008).
- 69. Decourcelle, A., Loison, I., Baldini, S., Leprince, D. & Dehennaut, V. Evidence of a compensatory
regulation of colonic O-GlcNAc transferase and O-GlcNAcase expression in response to disruption
of O-GlcNAc homeostasis. *Biochem. Biophys. Res. Commun.* **521**, 125–130 (2020).
- 70. Qian, K. *et al.* Transcriptional regulation of O-GlcNAc homeostasis is disrupted in pancreatic
cancer. *J. Biol. Chem.* **293**, 13989–14000 (2018).

REVIEWER COMMENTS

Reviewer #1 (Remarks to the Author):

The additional analysis of the proteomic data is very informative, although representation of the data by PCA and hierarchical clustering does highlight the limitation of using low numbers of animals. As the authors point out, statistically significant proteins could only be identified using raw p-value, and the reason for this is evident from the PCA plot and cluster analysis.

In particular, the day-7 LV apex samples, while clearly separated from the 0 and 2hr samples in Principal Component 2 (which accounts for 11% of the variation), split into two subgroups in Principal Component 1 (which accounts for ~43% of the variation). These sub-groups are evident on the HM_apex figure, where R402, R403 and R404 are clearly distinct from R405 and R406.

I am therefore left wondering which of these subgroups is the 'representative' one, and how well the results can be extrapolated to a larger number of animals were the experiment repeated?

The authors need to explicitly comment in the manuscript on the level of biological/experimental variation revealed by this analysis and acknowledge the limitation of using n=5 animals where potentially two are 'outliers'.

Apart from these issues The authors should be commended for a very thorough revision of the manuscript and the provision of new data.

Reviewer #2 (Remarks to the Author):

Dear Authors,

In this very interesting study the authors highlighted the long term structural and metabolic changes in patients with TTS. As it is increasingly recognized that TTS involves long-term functional changes, this study contributes to the understanding of the possible changes and their origin. The evidence of fibrosis (even though never histologically demonstrated in patients with TTS), mid to long-term change in glucose metabolism and also engagement in the HBP signaling are interesting findings, especially when we consider the long term functional and structural changes that have been reported by Scally and Colleagues in their 2018 and 2019 published papers in Circulation.

While their findings and conclusions are sound, and a great deal of effort can be seen, the N is a little disappointing in some of the analyses (probably due to the number of experiments, but still).

Comments:

1. Strain analysis:

The authors did great effort to quantify LV Dysfunction due to Isoproterenol, and instead of being satisfied with the subjective diagnose of "apical ballooning" they analyzed the strain parameters, which is a very elegant proof, however, it should be mentioned (if already happened, please indicate where) if TTS like changes could be observed in all rodents, and if not, if the rodents without changes in their strain parameters have been excluded. And since the numbers are differing: why could you obtain different numbers of measurements at equal timepoints? (Figure 1 A,B,C at 3 months: n=4, but n=5 in the circumferential strain analysis). And in the atriums we have n=6?

2. Animal count:

Please indicate how many mice were treated?

In how many rodents was the ventricular function quantified with strain (if not all: how many and why not in all)?

- In figure 2 we have 9 measurements at every timepoints, are those the same animals was stress-cardiomyopathy even induced in all of them (as defined by strain?)
- In figure 3 we have different numbers of stainings for each antigen, why is that? the same amount of tissue sections of different rats should be available?

3. Western Blots:

The first Blot has a substantially different look. Please show a picture of the uncropped Western Blot (GAPDH missing?). Why is there a different number of samples in every Plot? (First plot says $n=4, n=5, n=5$) and there is a picture of 2/3 bands?

In the second Blot we see 4 different samples at 3 timepoints but shouldn't there be $3+3+3+3+3$ since $n=5$?

In the third Blot the full n is depicted, shouldn't the other pictures look alike?

4. Histology:

Figure 5. Why is there a substantial bigger number that was stained for Plot G & H compared to E & F, why were the other rodents excluded?

Figure 6. Why did you only stain 4 samples here?

So while the findings are very interesting and a great addition in the field, these substantial comments should be addressed.

Reviewer #3 (Remarks to the Author):

My comments have been well addressed.

Reviewer #4 (Remarks to the Author):

This revision is improved over the initial submission; however, there remain concerns regarding how the authors over interpret their metabolic data. While not uncommon in studies involving proteomics the authors frequently extrapolate changes in the abundance of proteins with changes in activity of metabolic pathways and then link the changes in the activity of these pathways to pathology. However, the measurement of an abundance of one or more proteins in a pathway contains no inherent information regarding the activity of that pathway. Specific examples are given below.

Another concern is that the results section contains discussion of interpretation and speculation of the meaning of much of the metabolism data that should be limited to the Discussion or Introduction. For example, the information contained in lines 156-166 is much more suited to the Introduction, whereas lines 173-175 should be moved to the Discussion. Such changes would greatly streamline the Results section, while at the same time separating and thereby clarifying descriptions of the data from interpretation of these data.

Overall, the metabolic data broadly demonstrates metabolic remodeling in TTC, which could be a contributing factor to the observed cardiac dysfunction. However, attributing changes in activity of any specific metabolic pathway(s) to such remodeling and dysfunction is beyond the scope of the data presented in the manuscript.

1) Lines 173-175: In the Results at the end of the first section describing the FDG PET data shortly following ISO administration it is stated that "An increase of the entry of glucose into myocardial cells combined to a decrease in its rate of phosphorylation and lipidotoxicity observed in oil-red staining correspond to "glucose metabolic stunning".

The FDG data certainly indicate increased glucose transport and decreased glucose phosphorylation.

The conclusion regarding lipotoxicity is based on Oil Red O staining shown in Extended Data Fig 5. An increase in Oil Red staining suggests an increase in neutral triglycerides and lipids in the heart at the 2hr time point, which could occur for a variety of reasons. However, these data are insufficient on their own to substantiate the development of lipotoxicity. In the pathway analysis in Extended data 8 there does not appear to be indications of impaired fatty acid oxidation at the 2hr time point, which could be one factor contributing to increased lipid accumulation. It is noted that Shao et al. (Eur J Heart Fail 2013 Jan;15(1):9-22. doi: 10.1093/eurjhf/hfs161), surprisingly not cited here, reported very similar results following acute ISO treatment, which they also described as lipotoxicity, although in that study a much more detailed analysis of lipids and lipid metabolic pathways was provided.

Nevertheless, it is unclear how such discordant metabolic changes can be used to describe a phenomenon of “glucose metabolic stunning”. Metabolic stunning has more typically been used in PET studies to describe delayed recovery of metabolism compared to recovery of cardiac function or perfusion. Here there is evidence of both cardiac stress/dysfunction and metabolic dysfunction at 2hrs following ISO, consequently it is difficult to understand why this is described as “metabolic stunning”.

2) In Section 4 of the results it is concluded that the proteomic data “indicate inactivation of glycolysis and oxidative phosphorylation and over-activation of glucose alternative pathways, namely the hexosamines biosynthetic (HBP) and polyol pathways” (Lines 191-193). It is a fundamental characteristic of proteomic data that it cannot alone indicate changes in the activation of specific metabolic pathways. There is no doubt that the pathway analysis data in Extended Data 8 is valuable and can provide supporting results, but it should be clearly stated that such data are “consistent with” or “supportive of changes” in pathway activities. The statement on line 195 that “large amounts of myocardial G6P were diverted to alternative pathways” cannot be supported by the pathway analyses.

3) Figure 4 and lines 195-204: Since a major theme is a decrease in glucose metabolism via glycolysis, immunoblots confirming the proteomic analysis would help support this conclusion. The same is true for OGT, OGA, aldose reductase and sorbitol dehydrogenase, since increased glucose metabolism via these pathways are proposed as contributing to the adverse remodeling observed in the TTC model.

As GAPDH changes in response to ISO treatment it cannot be used as a loading control for GFAT2 immunoblots. Also loading controls should be provided for GFAT1 and O-GlcNAc immunoblots.

It is surprising given the proposed importance of the HBP and PPP in TTC remodeling that neither pathway is identified in the pathway analyses.

4) Line 264: it is stated that “myocardial glycolysis was reduced or remained unchanged”. Based on the methods used and results presented, it is unclear how the rate of glycolysis was determined. If glycolytic rate was not measured directly this needs to be rephrased to reflect the results more accurately.

5) Line 266-267: As discussed earlier the concept of glucose metabolic stunning is inherently flawed as there does not appear to be a mismatch between function and metabolism. In addition no data are provided to indicate that G6P is diverted into alternative pathways of glycolysis. This needs to be rephrased to reflect the results more accurately.

6) Lines 328-343: Please change “hyperactivation” or “overactivation” to reflect the results in this study, and those cited, more accurately. For example, “significant increase in protein expression” would be one approach. It is unfortunately a limitation in studies of the HBP and protein O-GlcNAcylation that enzyme activities and metabolic fluxes through these pathways are rarely measured. As a result, steady state changes in protein expression or metabolite levels are often, misleadingly, characterized as changes in activity or flux.

Revisions – Comments to the questions

We thank all reviewers for their comments that have helped to greatly improve the manuscript. Please find attached a point-by-point answers to the reviewers' comments. The corresponding changes made in the main text are color-coded as follows:

- Comments/Answers to reviewer #1.
- Comments/Answers to reviewer #2.
- Comments/Answers to reviewer #4.

REVIEWER COMMENTS

Reviewer #1 (Remarks to the Author):

The additional analysis of the proteomic data is very informative, although representation of the data by PCA and hierarchical clustering does highlight the limitation of using low numbers of animals. As the authors point out, statistically significant proteins could only be identified using raw p-value, and the reason for this is evident from the PCA plot and cluster analysis.

In particular, the day-7 LV apex samples, while clearly separated from the 0 and 2hr samples in Principal Component 2 (which accounts for 11% of the variation), split into two subgroups in Principal Component 1 (which accounts for ~43% of the variation. These sub-groups are evident on the HM_apex figure, where R402, R403 and R404 are clearly distinct from R405 and R406.

I am therefore left wondering which of these subgroups is the 'representative' one, and how well the results can be extrapolated to a larger number of animals were the experiment repeated?

The authors need to explicitly comment in the manuscript on the level of biological/experimental variation
revealed by this analysis and acknowledge the limitation of using n=5 animals where potentially two are
'outliers'.

Apart from these issues the authors should be commended for a very thorough revision of the manuscript
and the provision of new data.

We thank you for your pertinent comments. The number of samples used in proteomic mass spectrometry
analyses is usually small (n=3-5), and in our study, we have endeavored to enhance statistically the
sensitivity of peptide ions detection, peptide abundance and functional quantification, and the sensitivity of
the level of activity of associated signaling pathways, through (i) sample replication (n=5 per group), (ii)
randomization of samples, and (iii) separate analysis of three conditions (0, 2h, and 7d post-ISO), and of
regional heart samples (apex vs. base of LV), following the recommendations of Clough et al. (**Clough, T.,
Thaminy, S., Ragg, S., Aebersold, R., & Vitek, O. (2012). Statistical protein quantification and
significance analysis in label-free LC-MS experiments with complex designs. BMC bioinformatics,
13(16), 1-17.**). We agree with the Reviewer that there are two subgroups among the 7h post-ISO timepoint
analysis of the apex. To define which subgroup is representative of the phenotype we opted for a 1D
principal component analysis, i.e., with only component 1 (see figure hereunder). The evolution of the
phenotypes clearly shows a majority of the baseline samples under 0 and a majority of the post-ISO samples
above 0. In this PCA, the 2 outliers are R405 and R406, while the 3 others show a post-ISO phenotype.
Therefore, the statistics are dominated by the 3 post-ISO samples, while the 2 outliers decrease the power
of the statistical test by excluding proteins of interest. We would certainly benefit from validating our
hypotheses on a larger sample size; however, the proteomic results were confirmed by other experimental
approaches including western blots, histology, and FDG PET to allow an accurate and fine-grained
interpretation of the mechanisms involved in the myocardium at different post-ISO time points.

**PCA: 1D representation of the Component 1 of PCA related to proteins with 100% of valid values**
**(proteins that have LFQ intensity value for all conditions) in the LV apex. The two apical 7 days**
**“outliers” are located in the control side.**

This can be explained by sample variability in 2 processed mouse hearts. Indeed, a close look at the
performance identification rate and/or missing value occurrence shows that the samples have clearly
underperformed in this analysis. The following chart reporting the number of identified proteins per
injection shows that the hearts rats R405 and R406, which correspond to injections A14 and A15,
respectively, had a lower protein identification yield. The same is true for rat R411 that corresponds to the
sample injection A9, that accordingly also segregated in the PCA 2 figure of the previous point-by-point
answers to reviewers' comments.

Thus, the two replicates that segregated separately in the 1st PCA scale also segregated separately in the
2nd scale, while 3 replicates out of 5 are comparable to the whole dataset. Downsizing the replicate number
to only these 3 replicates would have eliminated candidate proteins, therefore the advantage of n=5 in terms
of statistical power compared to n= 3. To state it simply: the 1st PCA dimension may translate differences
in sample identification performances due to sample processing issues

**Reviewer #2 (Remarks to the Author):**

Dear Authors,

In this very interesting study the authors highlighted the long term structural and metabolic changes in
patients with TTS. As it is increasingly recognized that TTS involves long-term functional changes, this
study contributes to the understanding of the possible changes and their origin. The evidence of fibrosis

(even though never histologically demonstrated in patients with TTS), mid to long-term change in glucose
metabolism and also engagement in the HBP signaling are interesting findings, especially when we consider
the long term functional and structural changes that have been reported by Scally and Colleagues in their
2018 and 2019 published papers in Circulation.

We thank the Reviewer for his encouraging comments.

While their findings and conclusions are sound, and a great deal of effort can be seen, the N is a little
disappointing in some of the analyses (probably due to the number of experiments, but still).

Comments:

1. Strain analysis:

The authors did great effort to quantify LV Dysfunction due to Isoproterenol, and instead of being satisfied
with the subjective diagnose of "apical ballooning" they analyzed the strain parameters, which is a very
elegant proof, however, it should be mentioned (if already happened, please indicate where) if TTS like
changes could be observed in all rodents, and if not, if the rodents without changes in their strain parameters
have been excluded. And since the numbers are differing: why could you obtain different numbers of
measurements at equal timepoints? (Figure 1 A, B, C at 3 months: n=4, but n=5 in the circumferential strain
analysis). And in the atriums, we have n=6?

Thank you for this comment, we reckon it was not explicit in the Methods section of our manuscript, please
find here a thorough description of the *Ns* and the reasons for their variation along time points and cardiac
wall chambers. For strain measurements, we followed longitudinally the same exactly 6 rats at 0 (baseline),
2h, and 7d post-ISO. Each animal was its own control throughout the successive time points, which allows
a better statistical robustness of the variability of quantitative parameters. All 6 rats showed abnormalities
of myocardial wall strain but, unfortunately, we could not obtain all directional strain measurements in all
wall chambers at all time points in each rat. This is because presently, the acquisition of cardiac strain
parameters using MRI in rodents still carries technical limitations, due to low temporal resolution (16
frames/heart cycle for a heart rate beating at 400 beats/min), and to random loss of some orientation planes
using the four-chamber-view with some missing data. Please note that technical failure was the only reason
for the variations of *n* between the three pre- and post-ISO conditions and between the types of strain
measured. Missing data corresponds to the incalculable values, and no data was removed to ameliorate the
statistical significance (see revised manuscript, page 19, lines 488-490).

2. Animal count:

Please indicate how many mice were treated?

As mentioned in the revised manuscript (see page 15, line 398), a total of 120 rats were used in this study,
this total number comprising the setup of the ISO model.

In how many rodents was the ventricular function quantified with strain (if not all: how many and why not
in all)?

As mentioned above, strain measurements were obtained in 6 rats at 0, 2h, 7d, 1mo and 3mo post-ISO. In
each of these animals the physiological parameters were monitored at 2h post-ISO (acute stress phase) and
compared with baseline to control the presence of a Takotsubo-type response to isoprenaline stress, notably
hyperglycemia and increased heart rate, as shown in the graphs below. Please also note that each animal
being his own control.

- In figure 2 we have 9 measurements at every timepoints, are those the same animals was stress-
 cardiomyopathy even induced in all of them (as defined by strain?)?

Nine rats were monitored using FDG PET at 0, 2h, 7d, 1mo and 3mo post-ISO, each animal serving as its
 own control, which allows for statistical robustness of quantitative parameters. Physiological parameters
 were monitored in these rats to validate the catecholamine stress induction of Takotsubo type
 cardiomyopathy (see graphs).

- In figure 3 we have different numbers of stainings for each antigen, why is that? the same amount of tissue
sections of different rats should be available?

Thank you for this question. Not all heart sections were triple stained simultaneously, which explains the
difference in *n* between the stains. The overall performance of the Opal™ Multiplex immunohistochemistry
method was a little under 100%, which also explains the difference in *n* between the different staining
results.

3. Western Blots:

The first Blot has a substantially different look. Please show a picture of the uncropped Western Blot
(GAPDH missing?). Why is there a different number of samples in every Plot? (First plot says n=4, n=5,
n=5) and there is a picture of 2/3 bands?

The GFAT1 western blots were done on the same set of animals as the mass spectrometry, and the figure
is representative of GFAT1 expression levels. The other western blots (GFAT2, O-GlcNAc) were done at
the request of Reviewer 4 on a different set of animals for the revision of the manuscript. Please find the
complete sets of blots in the accompanying material. We have also changed the representation of the
GFAT1 blots in figure 4 of the manuscript (figure 4, page 28 in revised manuscript). Below are shown all
blots with their GAPDH loading controls.

 In the second Blot we see 4 different samples at 3 timepoints but shouldn't there be 3+3+3+3+3 since n=5?
 As with the first blot, this one is a representative illustration of the relative quantification of GFAT2
 expression level in the apex. Please find here below the full series.

 In the third Blot the full n is depicted, shouldn't the other pictures look alike?
 All samples of the third blot are shown, in contrast to the other blots that included both apical and basal
 samples of the LV that we did not show to maintain coherence with the quantification graphs.

 4. Histology:

Figure 5. Why is there a substantial bigger number that was stained for Plot G & H compared to E & F,
 why were the other rodents excluded?

The reason for the difference in the number of LV (E and F) and atria (G and H) is that we initially explored
 fibrosis in the LV only, not in the atria in which we did not expect to find changes. However, after MRI
 strain measurements evidenced a pathology of the left atrial strains, we quantified fibrosis in a new series
 of animals in the left atrium and in the LV. Therefore, histological analyses correspond to two series of

animals for the LV and to one series for the atria, hence a larger n for E and F than for G and H panels. No
animals were excluded from the histological analyses.

Figure 6. Why did you only stain 4 samples here?

Although co-immuno-staining of CD31 and alpha-SMA used 4 rats, each heart was sectioned in 8 sections
(5 apical sections and 3 basal sections of the LV), for replications allowing a better robustness and reliability
of histological observations and quantifications. Thus, in order to avoid initiating a new series of
experiments and to reduce the number of animals, we stained all 8 sections of four hearts.

**Reviewer #3 (Remarks to the Author):**

My comments have been well addressed.

We thank the Reviewer for this remark.

**Reviewer #4 (Remarks to the Author):**

This revision is improved over the initial submission; however, there remain concerns regarding how the
authors over interpret their metabolic data. While not uncommon in studies involving proteomics the
authors frequently extrapolate changes in the abundance of proteins with changes in activity of metabolic
pathways and then link the changes in the activity of these pathways to pathology. However, the
measurement of an abundance of one or more proteins in a pathway contains no inherent information
regarding the activity of that pathway. Specific examples are given below.

Another concern is that the results section contains discussion of interpretation and speculation of the
meaning of much of the metabolism data that should be limited to the Discussion or Introduction. For
example, the information contained in lines 156-166 is much more suited to the Introduction, whereas lines
173-175 should be moved to the Discussion. Such changes would greatly streamline the Results section,
while at the same time separating and thereby clarifying descriptions of the data from interpretation of these
data.

As suggested by the Reviewer, we have moved some parts of the Results section into the Introduction and
Discussion sections (see revised manuscript, page 3, lines 62-73).

Overall, the metabolic data broadly demonstrates metabolic remodeling in TTC, which could be a
contributing factor to the observed cardiac dysfunction. However, attributing changes in activity of any
specific metabolic pathway(s) to such remodeling and dysfunction is beyond the scope of the data presented
in the manuscript.

We agree with the reviewer: metabolic remodeling is a contributing factor to cardiac dysfunction.
Considering the complexity of cardiac metabolism, it is most likely that a combination of mechanisms
concurs to long-term tissue remodeling and cardiac malfunction. Our intention was to pinpoint glucose
alternative metabolism as an important contributor, not to eliminate other possibilities such as acute
inflammatory signaling, calcium signaling, or others.

1) Lines 173-175: In the Results at the end of the first section describing the FDG PET data shortly
following ISO administration it is stated that “An increase of the entry of glucose into myocardial cells
combined to a decrease in its rate of phosphorylation and lipidototoxicity observed in oil-red staining
correspond to “glucose metabolic stunning”.

The FDG data certainly indicate increased glucose transport and decreased glucose phosphorylation. The
conclusion regarding lipotoxicity is based on Oil Red O staining shown in Extended Data Fig 5. An increase
in Oil Red staining suggests an increase in neutral triglycerides and lipids in the heart at the 2hr time point,
which could occur for a variety of reasons. However, these data are insufficient on their own to substantiate
the development of lipotoxicity. In the pathway analysis in Extended data 8 there does not appear to be
indications of impaired fatty acid oxidation at the 2hr time point, which could be one factor contributing to
increased lipid accumulation. It is noted that Shao et al. (Eur J Heart Fail 2013 Jan;15(1):9-22. doi:
10.1093/eurjhf/hfs161), surprisingly not cited here, reported very similar results following acute ISO
treatment, which they also described as lipotoxicity, although in that study a much more detailed analysis of
lipids and lipid metabolic pathways was provided.

Lipidototoxicity or lipotoxicity or in other words abnormal and transient accumulation of lipids in the
myocardium is a characteristic clinical feature of Takotsubo cardiomyopathy, especially in its acute phase
(Nef, H. M., Möllmann, H., Kostin, S., Troidl, C., Voss, S., Weber, M., ... & Elsässer, A. (2007). Tako-
Tsubo cardiomyopathy: intraindividual structural analysis in the acute phase and after functional
recovery. *European heart journal*, 28(20), 2456-2464.). Lipid droplets are tiny cellular organelles that
control the storage and metabolism of neutral lipids. In non-adipocytes, lipid droplets protect against an
excess of fatty acids (FAs) because they store surplus FAs in form of neutral triacylglycerol, as extensive

lipid droplet accumulation can lead to an inflammatory response and so called lipidotoxicity³ (**Rawish, E.,**
**Nickel, L., Schuster, F., Stölting, I., Frydrychowicz, A., Saar, K., ... & Raasch, W. (2020). Telmisartan**
**prevents development of obesity and normalizes hypothalamic lipid droplets. Journal of**
**Endocrinology, 244(1), 95-110.**). As motioned by the Reviewer, Shao et al. have indeed demonstrated
intramyocardial lipid droplet accumulation in rodent cardiomyocytes in response to catecholamine surge⁴
(**Shao, Y., Redfors, B., Ståhlman, M., Täng, M. S., Miljanovic, A., Möllmann, H., ... & Omerovic, E.**
**(2013). A mouse model reveals an important role for catecholamine-induced lipotoxicity in the**
**pathogenesis of stress-induced cardiomyopathy. European journal of heart failure, 15(1), 9-22.**). This
had been confirmed in myocardial biopsies of patients suffering TTC during the acute phase but not after
recovery (**Nef, H. M., Möllmann, H., Kostin, S., Troidl, C., Voss, S., Weber, M., ... & Elsässer, A.**
**(2007). Tako-Tsubo cardiomyopathy: intraindividual structural analysis in the acute phase and after**
**functional recovery. European heart journal, 28(20), 2456-2464.**; **Rawish, E., Stiermaier, T., Santoro,**
**F., Brunetti, N. D., & Eitel, I. (2021). Current knowledge and future challenges in Takotsubo**
**syndrome: part 1-pathophysiology and diagnosis. Journal of clinical medicine, 10(3), 479.**). In our
study, it was only a question of verifying these clinical and preclinical descriptions of transient
lipidotoxicity by oil-red staining in order to validate our Takotsubo-like animal model (see revised
manuscript, page 4, lines 107-108 and page 7, lines 175-178).

Nevertheless, it is unclear how such discordant metabolic changes can be used to describe a phenomenon
of “glucose metabolic stunning”. Metabolic stunning has more typically been used in PET studies to
describe delayed recovery of metabolism compared to recovery of cardiac function or perfusion. Here there
is evidence of both cardiac stress/dysfunction and metabolic dysfunction at 2hrs following ISO,
consequently it is difficult to understand why this is described as “metabolic stunning”.

We, as others (**Rawish, E., Stiermaier, T., Santoro, F., Brunetti, N. D., & Eitel, I. (2021). Current**
**knowledge and future challenges in Takotsubo syndrome: part 1-pathophysiology and diagnosis.**
**Journal of clinical medicine, 10(3), 479;** **Shao, Y., Redfors, B., Ståhlman, M., Täng, M. S., Miljanovic,**
**A., Möllmann, H., ... & Omerovic, E. (2013). A mouse model reveals an important role for**
**catecholamine-induced lipotoxicity in the pathogenesis of stress-induced cardiomyopathy. European**
**journal of heart failure, 15(1), 9-22.**) used the term "metabolic stunning" to qualify the mismatch between
the energetic needs of the heart to counter ventricular dysfunction and the availability of energy metabolites
and their dysregulation in the acute phase following stress. The term has previously been used in the context

of perfusion/metabolism, while in our sense and others, it depicts an unexpected metabolic switch.
Therefore, in order to avoid misinterpretation, we have removed this term from the revised manuscript.

2) In Section 4 of the results it is concluded that the proteomic data “indicate inactivation of glycolysis and
oxidative phosphorylation and over-activation of glucose alternative pathways, namely the hexosamines
biosynthetic (HBP) and polyol pathways” (Lines 191-193). It is a fundamental characteristic of proteomic
data that it cannot alone indicate changes in the activation of specific metabolic pathways. There is no doubt
that the pathway analysis data in Extended Data 8 is valuable and can provide supporting results, but it
should be clearly stated that such data are “consistent with” or “supportive of changes” in pathway
activities. The statement on line 195 that “large amounts of myocardial G6P were diverted to alternative
pathways” cannot be supported by the pathway analyses.

The reviewer rightly points to the fact that measuring the abundance of a protein is different from measuring
its activity. However, the conclusions of our study were obtained by collecting data from multiple sources,
i.e., *in vivo* imaging using three different modalities, histology, physiology, blood tests, in addition to
protein levels using western blots and proteomics. Regarding the latter, the analysis of our proteomics data
was performed with the reference software QIAGEN IPA that calculates z-scores in order to predict the
activation status of a pathway based on the fold-change of its actors and literature-based effects on
pathways. As regarded, the use of terms such as “inactivation”, “over-activation” and “large amounts of
myocardial G6P were diverted to alternative pathways” are relevant here, please refer to 3), 5) and 6).

3) Figure 4 and lines 195-204: Since a major theme is a decrease in glucose metabolism via glycolysis,
immunoblots confirming the proteomic analysis would help support this conclusion. The same is true for
OGT, OGA, aldose reductase and sorbitol dehydrogenase, since increased glucose metabolism via these
pathways are proposed as contributing to the adverse remodeling observed in the TTC model.

The recent work of Godsman et al. confirms an inactivation of glycolysis and over-uptake of glucose in the
Takotsubo myocardium (Godsman, N., Kohlhaas, M., Nickel, A., Cheyne, L., Mingarelli, M.,
Schweiger, L., ... & Dawson, D. K. (2022). **Metabolic alterations in a rat model of takotsubo syndrome.**
**Cardiovascular Research, 118(8), 1932-1946.**). Our own analyses of several enzymes of the cardiac
glucose metabolism pathways confirm the overactivation of polyol and hexosamines biosynthetic pathways
shown by proteomic analysis.

As GAPDH changes in response to ISO treatment it cannot be used as a loading control for GFAT2
immunoblots. Also loading controls should be provided for GFAT1 and O-GlcNAc immunoblots.

The anti-GAPDH antibody that we used for the western blots detects mainly its monomeric form (molecular
weight 36 kDa), a minor portion of its dimeric form, and not the tetrameric form (146 kDa) of GAPDH
which is its active form (<https://www.abcam.com/gapdh-antibody-6c5-loading-control-ab8245.html>).

This antibody detects GAPDH from the cytoplasm, cytosol, perinuclear region, membrane and from the
cell nucleus, and corresponds to the inactive form of GAPDH present in different cellular compartments.

Therefore, its constant expression level here cannot reflect its enzymatic activity level, especially in
glycolysis. Thus, it is ideal as loading control for western blots of cardiac proteins (please see figure
hereunder).

**Figure showing Rouge Ponceau (left) and GAPDH immunoblot stains (right) of four western blotting**
**membranes used for Western Blots in the study. All samples are cardiac tissue extracts from the**
**takotsubo rat model except lanes with an asterisk (*) that correspond to our western blotting internal**
**reference. Numbers [1,2,3,4] correspond to the same Rouge Ponceau and GAPDH immunoblots.**

It is surprising given the proposed importance of the HBP and PPP in TTC remodeling that neither pathway
is identified in the pathway analyses.

For over-representation analyses of proteomic data, we used Ingenuity® Pathway Analysis software
(QIAGEN Inc, version 60467501) to iteratively build networks capable of representing biological systems.

The software allows for rapid visualization and understanding of complex omics data and insightful data

analysis and interpretation by placing experimental omics data in the context of biological systems. These
biological systems depend on the history of signaling pathways and involve proteins highlighted in the
scientific literature. The alternative anabolic pathways of glucose in the myocardium have been little
explored and are not yet highlighted in the scientific literature, which explains their absence from
Ingenuity® analyses.

4) Line 264: it is stated that “myocardial glycolysis was reduced or remained unchanged”. Based on the
methods used and results presented, it is unclear how the rate of glycolysis was determined. If glycolytic
rate was not measure directly this needs to be rephrased to reflect the results more accurately.

We agree with the Reviewer that we cannot deduce rates of glycolytic activity and should only refer to
activation and inactivation of the pathway and have corrected revised manuscript accordingly (see page 10,
line 267).

5) Line 266-267: As discussed earlier the concept of glucose metabolic stunning is inherently flawed as
there does not appear to be a mismatch between function and metabolism. In addition, no data are provided
to indicate that G6P is diverted into alternative pathways of glycolysis. This needs to be rephrased to reflect
the results more accurately.

The mention to metabolic stunning has now been removed, please refer to 1).

We quantified the expression level of Hexokinase2 in the LV apex by western blot, which showed, as did
the proteomic analyses, a high expression/abundance level at 7d post-ISO compared to the control group
(see graph below) and suggesting increased phosphorylation of glucose into G6P at 7d post-ISO compared
to pre-ISO baseline. The other enzymes of this pathway being in low abundance at 7d post-ISO, according
to statistical analyses of over-representation of proteomic data, it infers that glycolysis was inactivated. This
is highly suggestive of the fact that the excess incoming glucose and excess G6P at 7d post-ISO are derived
to glucose alternative pathways.

6) Lines 328-343: Please change “hyperactivation” or “overactivation” to reflect the results in this study, and those cited, more accurately. For example, “significant increase in protein expression” would be one approach. It is unfortunately a limitation in studies of the HBP and protein O-GlcNAcylation that enzyme activities and metabolic fluxes through these pathways are rarely measured. As a result, steady state changes in protein expression or metabolite levels are often, misleadingly, characterized as changes in activity or flux.

The proteins involved and their level of abundance in these pathways are clearly shown by the proteomic analyses. Moreover, the western blots of O-GlcNAcylated proteins, products of HBP pathway activation, and those of the two isoforms of the rate-limiting enzyme of this pathway, GFAT1 and GFAT2, are direct indicators of the level of activity of the HBP pathway and, in this case, of its overactivation at 7d post-ISO. Therefore, we believe that the term "overactivation" is legitimate here.

REVIEWER COMMENTS

Reviewer #1 (Remarks to the Author):

I thank the authors for providing a more in-depth presentation of their proteomic data, and I accept that the salient findings have been verified using orthogonal methods.

The authors response did not include any acknowledgement of the limitation of using n=5 animals with two d7 post-ISO samples that were clearly affected due to sample processing issues. I suggest that this is included in the manuscript.

The bar graph indicates a couple of further concerns that should be addressed. In the methods section there is no detail provided for inclusion/exclusion of a protein according to the number of valid values (eg what was the minimum number of samples required for a protein to be included?). Neither is there any mention of how remaining missing values were imputed.

The graph also indicates that the maximum number of protein ID's in any sample was ~2,500, yet in the manuscript it is claimed that 3,203 proteins were quantified across all the samples. This suggests that for any given sample many of these proteins were not detected.

It is unclear to me how sample injections A14, A15 and A9 with anywhere between ~1,200 and 2,000 protein ID's be included in this quantitative analysis given that this would involve a very high proportion of missing values.

Finally, can the authors also confirm that the sample injections were not done in the sequence A1-A15, as this would contradict their earlier statement about sample randomization?

Reviewer #2 (Remarks to the Author):

All questions have been answered.

Reviewer #4 (Remarks to the Author):

No further comments.

Revisions – Comments to the questions

We thank the reviewers for their comments that helped to improve the manuscript. Please find attached a point-by-point answer to Reviewer #1's remarks. The corresponding changes made in the main text are highlighted in **yellow**.

REVIEWER COMMENTS

Reviewer #1 (Remarks to the Author):

I thank the authors for providing a more in-depth presentation of their proteomic data, and I accept that the salient findings have been verified using orthogonal methods.

The authors response did not include any acknowledgement of the limitation of using n=5 animals with two d7 post-ISO samples that were clearly affected due to sample processing issues. I suggest that this is included in the manuscript.

We thank the reviewer for this remark. We have now reported this limitation in the Study limitations and the Methods sections of the revised manuscript (**lines 389-395, page 15 and lines 635-638, pages 24**). We have also added the figure of 1D PCA (see below **figure 1**), shown in the previous revision, in Supplementary information in order to allow a better transparency of our data and proteomic analysis (**extended data 9B, pages 10 and 13-14**).

Figure 1: 1D representation of the component 1 of the principal component analysis (PCA) related to proteins with 100% of valid values (proteins that have LFQ intensity value for all conditions) in the LV apex. The two apical 7d post-ISO “outliers” (R405 and R406 in red) are located in the control side (in black) while the three apical samples (in red) of the five 7d post-ISO samples distinguished from the control and 2h post-ISO groups, indicating a real post-ISO phenotype.

The bar graph indicates a couple of further concerns that should be addressed. In the methods section there is no detail provided for inclusion/exclusion of a protein according to the number of valid values (eg what was the minimum number of samples required for a protein to be included?). Neither is there any mention of how remaining missing values were imputed.

The graph also indicates that the maximum number of protein ID's in any sample was ~2,500, yet in the manuscript it is claimed that 3,203 proteins were quantified across all the samples. This suggests that for any given sample many of these proteins were not detected.

It is unclear to me how sample injections A14, A15 and A9 with anywhere between ~1,200 and 2,000 protein ID's be included in this quantitative analysis given that this would involve a very high proportion of missing values.

We thank the reviewer for these pertinent observations. We have added the inclusion and exclusion criteria for a given protein in the Methods section of the revised manuscript (lines 646–645, pages 24-25). We have also added Table 1 shown hereunder in the Supplementary information (extended data 9A, pages 10, 13 and 14) and mentioned it in the Methods section of the revised manuscript (lines 636-648, page 25). This table summarizes the number of proteins identified by MS proteomics and the number of proteins used for statistical comparisons of the 3 groups (control, 2h and 7d post-ISO). Statistical comparisons between groups were made using the (C) dataset of Table 1. Proteins with missing values in all samples of one group (0/5) and at least 3 values in another group (3/5) were considered as appearance / disappearance and included in the (C) dataset of 2618 proteins.

Number of proteins identified by MS proteomics and of proteins used for group comparisons		
(A)	Total number of identified proteins across all samples	4257
(B)	Among (A), proteins with at least one LFQ	3203
(C)	Among (B), proteins with at least 3 out of 5 LFQ in at least one group	2618
(D)	Proteins with a LFQ in each sample	971

Table 1: Number of proteins identified by MS proteomics

Additionally, the values reported in the sample-by-sample bar graph, shown in the previous revision (see figure blow), reflect only the identified proteins: without MBR and only quantifiable (= LFQ intensities columns). This restriction logically widens the gap between the 4257 total identified proteins and the 3203

proteins quantified in all samples, i.e., including MBR and selective imputations (reported **pages 24-25** in the revised manuscript).

Finally, can the authors also confirm that the sample injections were not done in the sequence A1-A15, as this would contradict their earlier statement about sample randomization?

We sincerely apologize for the discrepancy between the data presented on the bar graph (see above) and the answer given to the reviewer in the previous revision that had escaped our attention. Going back to the lab book we realized that, at the time the experiments were performed and in contrast to our present protocols, we had not randomized the samples and thus had not exactly followed the recommendations of Clough et al. 2012¹.

The lab books show that the samples were in fact injected in the following order: A1, B1, A2, B2 ...up to A9 B9, followed by A14, B14, B15, C15, and finally A10, B10, ... up to A13 B13. For each sample, the apical region (A) was injected first and the basal region (B) immediately after.

In favor of the reliability and confidence in the proteomic data (i) all the samples were prepared on the same day, (ii) a Coomassie blue staining showed no differences between samples in protein abundance after homogenization prior sample digestion, shown below are the Coomassie blue-stained SDS- PAGE and corresponding table, (iii) two non-successive samples from different injection series underperformed similarly, (iv) while A14 and B14 from the same heart behaved differently in terms of relative performance between apical and basal regions, and (v) A9-B9 and A15-B15 pairs, although not contiguous in injection

order, underperformed similarly, as if the hearts had yielded less material in spite of SDS PAGE and Coomassie staining showing similar profiles.

LV samples of Wistar female rats

	Groupe A (control)	Groupe B (2h post-ISO)	Groupe C (7d post-ISO)
Apical region	A_A1: R398 A_A2: R399 A_A3: R400 A_A4: R409 A_A5: R410	B_A6: R401 B_A7: R407 B_A8: R408 B_A9: R411 B_A10: R412	C_A11: R402 C_A12: R403 C_A13: R404 C_A14: R405 C_A15: R406

~ 400µL/sample

Coomassie blue of LV Apical samples (n=5/group)

¹Clough, T., Thaminy, S., Ragg, S., Aebersold, R., & Vitek, O. (2012). Statistical protein quantification and significance analysis in label-free LC-MS experiments with complex designs. *BMC bioinformatics*, 13(16), 1-17.

Corresponding changes have been added in the revised manuscript and in the “**Data processing protocol**” section of the PRIDE dataset (**PXD032667**):

“Five samples/group post-ISO were performed. Although, two of the five 7d post-ISO LV apical samples were outliers (extended data 9B), in order to maintain homogeneous sample comparison groups of individuals, we chose not to eliminate the outliers as they showed no series effect nor were they detrimental to our observations of statistical evidence.

Overall, after excluding reverse and contaminant proteins from analysis, 4257 proteins were identified, among which 3203 proteins had at least one LFQ value across all samples. Before performing statistics, the data was transformed to a logarithmic scale (\log_2) and further filtered: 2618 proteins with at least 3 out of 5 valid LFQ values in at least one experimental group were used for comparison between groups (extended data 9A). Significant differential proteins were identified with a two-tailed Student's t-test. Only proteins with $p\text{-value} < 0.05$ and $|\text{fold change}| > 1.3$ were kept for over-representation analysis using Ingenuity Pathway Analysis (QIAGEN Inc, version 60467501). Proteins with missing values in all samples of one group (0/5) and at least 3 values in another group (3/5) were considered as appearance / disappearance and included in the dataset of 2618 proteins. A significant difference was observed in 1110 proteins, from the 6 comparison groups (in the apex: 0 vs. 2h, 7d vs. 0 and 7d vs. 2h; in the basal region: 0 vs. 2h, 7d vs. 0 and 7d vs. 2h)."

REVIEWERS' COMMENTS

Reviewer #1 (Remarks to the Author):

Thank you to the authors for addressing the further comments regarding the proteomic analysis.

1 - study limitations. This is fine as an explanation, although the grammar needs to be improved ("raw identification" should be plural; should be "reasonable use" - what does "interesting tracks" mean, exactly; "etc" is not appropriate)

2 - protein ID numbers. The table is helpful to a degree. However, it is not clear what the difference between category A and B is. What does it mean for a protein to have LESS than one LFQ (ie 0 values)? Does D mean proteins with an LFQ value in EVERY sample? Please clarify both these points.

I am actually quite puzzled about this table and the explanation on page 5 of the rebuttal. How can it be that "3203 proteins had at least one LFQ across all samples" (group B) yet only 2618 proteins were detected in a minimum of 3/5 samples (group C)?

Thank you for clarifying the issue regarding randomization of the samples.

Revisions – Comments to the questions

Please find attached a point-by-point answer to Reviewer #1's remarks. The corresponding changes made in the main text are highlighted in **yellow**.

REVIEWER COMMENTS

Reviewer #1 (Remarks to the Author):

Thank you to the authors for addressing the further comments regarding the proteomic analysis.

We thank the reviewer for his encouragement to present the proteomic analysis as rigorously as possible.

1 - study limitations. This is fine as an explanation, although the grammar needs to be improved ("raw identification" should be plural; should be "reasonable use" - what does "interesting tracks" mean, exactly; "etc" is not appropriate)

We have reworded **lines 388-394** of the “study limitations” paragraph (**page 15**) in the main text as follows: “The proteomic analysis at 2 hours and 7 days were based on 5 replicates per condition, which may lead some samples to yield less protein raw identifications than the average across all samples. However, in the present study we used the proteomics data to build hypotheses regarding energetic pathway changes and interpreted these in the light of the *in vivo* and *ex vivo* imaging, and of the physiological and biochemical analysis. This reasonable use of proteomics-derived data does suggest clues/leads about pathophysiological mechanisms in TTC”.

2 - protein ID numbers. The table is helpful to a degree. However, it is not clear what the difference between category A and B is. What does it mean for a protein to have LESS than one LFQ (ie 0 values)? Does D mean proteins with an LFQ value in EVERY sample? Please clarify both these points. I am actually quite puzzled about this table and the explanation on page 5 of the rebuttal.

How can it be that "3203 proteins had at least one LFQ across all samples" (group B) yet only 2618 proteins were detected in a minimum of 3/5 samples (group C)?

We thank the reviewer for pointing out to the fact that the legend of **supplementary figure 8 (pages 15-16)** was not as explicit as expected. We have slightly modified the Table to avoid confusion and better showcase the stringency levels, i.e., L1-L4, and rewritten and completed the legend of extended data 9A in order to better fit with MaxQuant software description [Tyanova S, Temu T, Cox J. *The MaxQuant computational platform for mass spectrometry-based shotgun proteomics. Nat Protoc. 2016 Dec;11(12):2301-2319. doi: 10.1038/nprot.2016.136. Epub 2016 Oct 27. PMID: 27809316*], as follows: “(A) Number of proteins identified by MS proteomics used for group comparisons, indicating the levels (L) of stringency, L1 < L2 < L3 < L4. L1: Unfiltered data from the LFQ intensity columns of the *proteingroup.txt* file of MaxQuant output, corresponding to proteins identified with an intensity value in at least one sample, yet without necessarily a sufficient level of confidence and correlation across the proteins' distinct peptide intensities to qualify for quantification. L2: Proteins bearing at least one value in the LFQ intensity columns of the *proteingroup.txt* file of MaxQuant output. L3: Proteins filtered to show enough valid values for statistical analysis (3/5 samples in at least one group). These are the proteins listed in the table of Extended data 8. L4: Proteins with an LFQ value in every sample.”

Number of proteins identified by MS proteomics and of proteins used for group comparisons	
(L1) Total number of identified proteins across all samples	4257
(L2) Among (A), proteins with at least one LFQ	3203
(L3) Among (B), proteins with at least 3 out of 5 LFQ in at least one group	2618
(L4) Proteins with a LFQ in each sample	971

Table 1: Number of proteins identified by MS proteomics.

In addition, we now refer to the legend to supplementary figure 8b in the Material and Methods paragraph in **lines 602-605 (page 23)** of the main text:

“Five samples/group post-ISO were performed. In the five 7d post-ISO LV apical group, two samples were outliers. However, in order to maintain homogeneous sample comparison between groups, we did not eliminate these 2 outlier samples since (i) they showed no series effect and (ii) their inclusion did not flaw the statistical confidence of this group's comparisons (supplementary fig.8b).”

Thank you for clarifying the issue regarding randomization of the samples.

Again, we thank the reviewer for inciting us to revisit and correct the randomization issue.